# Characteristics of hail hazard in South Africa based on satellite detection of convective storms

Heinz Jürgen Punge[1], Kristopher M. Bedka[2], Michael Kunz[1], Sarah D. Bang[3], and Kyle F. Itterly[4]

[1]Karlsruhe Institute of Technology (KIT), Institute of Meteorology and Climate Research, Karlsruhe, Germany
[2]NASA Langley Research Center, Science Directorate, Climate Science Branch, Hampton, VA, USA
[3]NASA Marshall Space Flight Center (ST-11), Huntsville, AL, USA
[4]Science Systems and Applications Inc., Hampton, Virginia USA

**Correspondence:** Michael Kunz (michael.kunz@kit.edu)

**Abstract.** Accurate estimates of hail risk to exposed assets, such as crops, infrastructure and vehicles, are required for both insurance pricing and preventive measures. Here we present an event catalog to describe the hail hazard in South Africa guided by 14 years of geostationary satellite observations of convective storms. Overshooting cloud tops have been detected, grouped and tracked to describe the spatio-temporal extent of potential hail events. It is found that hail events concentrate mainly in the southeast of the country, along the Highveld, and around the eastern slopes. Events are most frequent from mid-November through February and peak in the afternoon, between 13 and 17 UTC. Multivariate stochastic modeling of event properties yields an event catalog spanning 25 000 years, aiming to estimate, in combination with vulnerability and exposure data, hail risk for return periods of 200 years.

## 1 Introduction

Damage from large hail is a significant contribution to natural hazards losses in many parts of the world (Punge and Kunz, 2016; Púčik et al., 2019; Allen et al., 2020), including South Africa, and growing research activity has opened up opportunities to estimate risk for the insurance sector (Punge et al., 2014; Rädler et al.). In South Africa, hail has long been known to generate large amounts of damage to agriculture – around two percent of the value of products (Carte, 1977) – and forestry (Smith et al., 2002; Wingfield and Swart, 1994). Events with severe hail damage to buildings, vehicles and infrastructure like the one on 28 November 2013 (total loss 1.4 bn South African Rand, around 140 mn US dollars, Powell and Burger, 2014; Visser, 2014) are numerous (e.g., Perry, 1995). Still, in comparison to other natural hazards, the sporadic occurrence and highly localized effects of hail pose a particular challenge to hazard quantification, which forms the basis for any risk modeling. Such modeling is required for the insurance sector to estimate the financial risks related to a hazard, as required, for example, by the Insurance Act 18 of 2017, and can guide measures to improve resilience.

Across the world, reliable records of hailfall including size information are limited to reports by volunteer observer networks (e.g., Held, 1974) and other sources, sometimes collected in databases of hail reports (e.g., Dotzek et al., 2009; Allen et al., 2015) or hailpad networks (e.g., Palencia et al., 2009). Leigh and Kuhnel (2001), for example, constructed a regional risk model based on such reports and loss data alone. Grieser and Hill (2019) used volunteer-collected hail observations in the

United States (Reges et al., 2016) to model the rate of hailstones hitting the ground per unit area, time, and hailstone size bin during the passage of a hailstorm. Based on that data, they set up a model to calculate the vulnerability of subjects at risk as a function of the diameter of the largest hailstone, which can be transferred to other regions.

For South Africa, Admirat et al. (1985) evaluated hailpad and hail reports from a network of voluntary hail observers (mainly farmers equipped with hail cards) to quantify hail properties in an area of 2800 $km^2$ in the "Transvaal Highveld," nowadays a part of the Gauteng region (see Fig. 2a). Using reports from the same network over a 19-year period Smith et al. (1998) found on average 68.5 hail days per year, much more than in other regions of similar size, for example, northern Italy, which is the highest hail-exposed region in Europe (e.g., Giaiotti et al., 2003; Punge et al., 2017). Of these, 3.3 days had hail greater than 3 cm.

While distinction between hail and rain or sleet is often challenging, remote sensing data from either radar or satellite instruments is required to determine the spatial extent of hail events and to depict the geographic distribution of the hazard (Puskeiler et al., 2016; Bedka et al., 2017; Nisi et al., 2018; Allen et al., 2020). Alternatively, numerical models such as high-resolution reanalysis can be used to identify atmospheric conditions favorable for hailstorm formation (e.g., Rädler et al.; Kunz et al., 2020; Taszarek et al., 2020). In that case, climatologies over long time series can be generated (Dyson et al., 2020; Prein and Holland, 2018). These are, however, generally limited by model resolution and the inaccurate representation of convective initiation, since hailstorms often form by local and meso-scale processes related to, for example, orographic lifting and mountain winds, low-level convergence zones, or land use inhomogeneities (Allen et al., 2020). In addition, reanalyses or regional climate models use simplified microphysical parameterization schemes and not two- or even three-moment schemes required for more realistic hail size modeling (Seifert and Beheng, 2006; Loftus et al., 2014; Wellmann et al., 2020). Several studies have used hail signals or hail detection algorithms for hail frequency assessments (Cintineo et al., 2012; Junghänel et al., 2016; Fluck et al., 2021) and risk modeling (Puskeiler et al., 2016; Nisi et al., 2018; Schmidberger, 2018). However, radar data is usually only available on country scales due to availability and inter-radar calibration issues. In South Africa, the use of radar data for nowcasting of hail has been studied for the Highveld region (Ayob, 2019). However, the South African radar network does not cover the entire country.

Even though satellite data are a less accurate proxy for hailstorm detection compared to radar, the big advantage is that these data cover comparatively larger areas almost homogeneously. The detection of hail via scattering of upwelling Earth-emitted microwave radiation is currently limited to satellites in low-earth orbit (Mroz et al., 2017; Ni et al., 2017; Bang and Cecil, 2019). Such data can be exploited for global analysis of hail occurrence as well as for identification of atmospheric conditions prevailing during individual hailstorms. The drawback, however, is the lack of temporal coverage required to examine the evolution of hailstorms. In contrast, indirect indicators have been designed to extract severe weather and hail signals from much more frequent and spatially detailed geostationary satellite imagery (Bedka et al., 2010; Melcón et al., 2016).

An overshooting cloud top (OT) indicates an intense updraft capable of generating hail. The OTs can be detected in both visible and infrared data (Bedka and Khlopenkov, 2016; Khlopenkov et al., 2021). In particular, the most severe hailstorms show a clear OT signature (e.g., Kunz et al., 2018; Wilhelm et al., 2021). The OT detection algorithm has been extensively calibrated and tested against severe weather reports and radar data (Bedka and Khlopenkov, 2016; Sandmæl et al., 2019;

Cooney et al., 2021). Still, in some cases, OT features may have been misdetected or may not have produced hail on the ground, for example, due to melting of hailstones during fall through a deep column of warm air. This is acknowledged in the studies of Punge et al. (2014, 2017) and by Bedka et al. (2018a), who noted the large percentages of OTs without hail on the ground. However, in addition to the hazard modeling purpose, the focus of our study is on the identification of larger spatial SCS clusters with an increased potential of hail production during the lifetime of the event, rather than detecting each individual storm with enhanced hail potential. These large-scale hail-producing outbreaks can cause by far the largest part of the damage registered by insurers, and can induce solvency issues when the risk was not properly estimated.

Punge et al. (2014, 2017) used the OT approach to estimate a hail event dataset for Europe which served as the physical basis for the Willis European Hail Model, the first fully randomized stochastic hail model to cover all of Europe. Since 2014, the model has been established as a standard tool in hail risk estimation and pricing among insurance and reinsurance companies in Europe. A similar approach was later applied to Australia (Bedka et al., 2018b). However, assessments of this kind are absent in many emerging insurance and markets around the world. Of these, South Africa is a prime example where hail is a common hazard and major risk driver. Therefore, we focus on South Africa in this article, refining the methodology of Punge et al. (2014, 2017) to describe hail hail events more accurately. In contrast to Bedka et al. (2010), the Khlopenkov et al. (2021) OT detection technique, which was applied here, provided a gridded probabilistic representation of an OT rather than a list of OT centroid pixel locations, accounting for both size and reliability of the updraft detections. The event definition procedure now tracks storm signatures over time, allowing to follow convective activity more closely. In the stochastic component of the model, rather than simply re-sampling historic events to describe possible future hazard, distributions of relevant event properties are modeled and sampled separately, conserving correlation among these properties. Improvements compared to the European and Australian hail models concern event definition, event parameter distributions, and detail of the stochastically generated footprints.

Section 2 presents the methodology and data sets used as input for the model, whereas Sect. 3 describes the derived hazard distribution and OT event sets. Stochastic event sampling and potential hail footprint generation are discussed in Sect. 4.

## 2 Methods and Data

The diagram in Fig. 1 illustrates how the different data sources, explained in the following, are combined in the model and processed to yield a set of event footprints representing 25 000 years of hail-generating convective storms based on the climate and weather of the period 2005-2018. Important steps in the development of the final stochastic modeling are: (i) filtering of OTs that are unreliable using both passive microwave hailstorm detections and insurance loss data combined with convective environments (convective available potential energy CAPE, wind shear, melting level) from ERA-5 reanalysis; (ii) cluster of OTs in space and time to attain single events; (iii) quantify histograms of most important event properties (length, width, duration, time of day, day of year) and their relations; consider a hailstone size spectrum from severe weather reports outside of the study area; (iv) adjust appropriate statistical distribution functions to the different properties; (v) stochastically generate (artificial) events that resemble the climatology of potential hail events and their characteristics; (vi) apply importance sampling

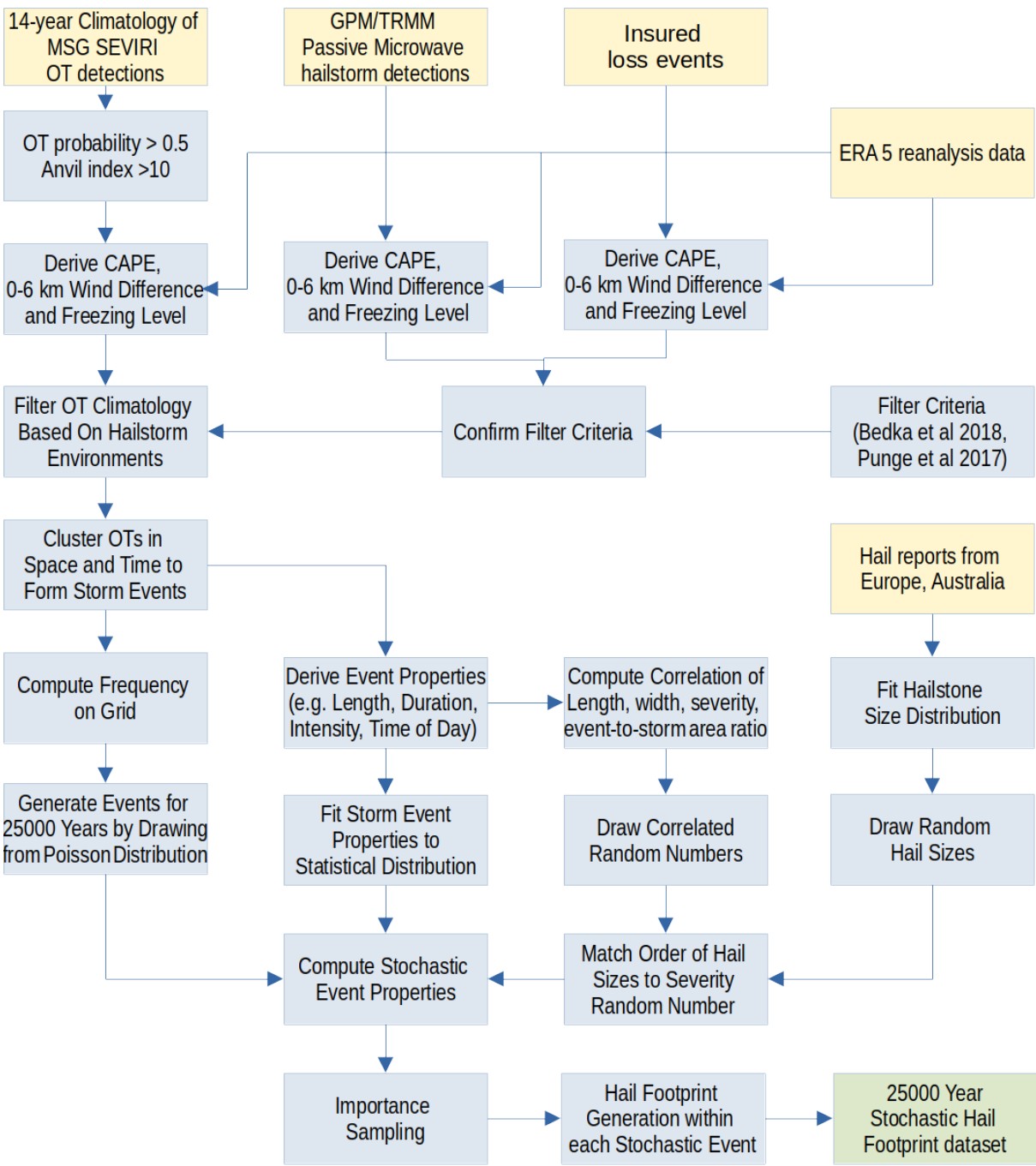

**Figure 1.** Diagram illustrating the functioning of the hail hazard model for South Africa.

to reduce the number of events; and, finally, (vii) compute single hail footprints for the stochastic event set. All the steps and procedures mentioned above will be explained in the following sections.

## 2.1 Study area

South Africa's topography consists of a large central plateau with extensive grasslands, a continuous escarpment of mountain ranges surrounding the plateau on the west, south and east, and a narrow strip of low-lying land near the coastline. The central plateau is bounded by the Great Escarpment (see Fig. 2a), a major topographical feature in Africa consisting of steep slopes that drop from the high central plateau toward the oceans that surround southern Africa on three sides. The eastern part of the Great Escarpment in the border region of South Africa and Lesotho is the Drakensberg Escarpment. It is the highest mountain range in southern Africa reaching an elevation of 3 482 above mean sea level (msl).

South Africa's climate is determined by the subtropical zone of the Southern Hemisphere, the location between the Atlantic and the Indian ocean, and its orographic characteristics. Climate conditions vary between subtropical and temperate, influenced by the ocean along the east and west coasts and the interior plateaus, warm, sub-tropical in the northeast, Mediterranean in the southwest and a warm dry desert environment in the central west and northwest.

## 2.2 Overshooting Top Detection

Intense thunderstorms are routinely observed by visible and infrared imagery from geostationary satellites for forecasting and warning purposes (Zinner et al., 2013; de Coning et al., 2015). In particular in the infrared channel, deep convective cloud tops atop updrafts appear as cold spots growing near to or above the tropopause level, surrounded by a warmer anvil (Adler et al., 1985). Cloudy air masses are propelled upwards in the storm's core before rebounding or dissolving again on time scales of a few minutes.

Detection of these OTs hereinafter referred to as OT detections has been automated by Bedka et al. (2010), revealing the climatological distribution in North America as well as in Europe (Bedka, 2011) and Australia (Bedka et al., 2018b). An advanced version of the OT detection algorithm described by Khlopenkov et al. (2021) delivers a 3-km gridded probabilistic estimate of OT likelihood based on a statistical combination of tropopause-relative infrared (IR) brightness temperature, prominence of an OT relative to the surrounding anvil, and the area and spatial uniformity of the anvil cloud surrounding an OT candidate region. OTs detected with a probability >50% and with a surrounding anvil cloud (green and yellow colors in Fig. 2b) are used in this work (Scarino et al., 2020; Khlopenkov et al., 2021). The method was validated by Cooney et al. (2021) using OT identifications from gridded weather radar observations and by Khlopenkov et al. (2021) using human OT identifications over the United States (US). A relation between hail size estimated from radar and OT intensity has been suggested in several studies (e.g., Bedka, 2011).

In the Khlopenkov et al. (2021) study, the human analysts identified OTs with two confidence levels, resulting in a conservative mask with only the most confident OTs and a liberal mask that also included less confident OT identifications that did not appear as prominently in the imagery as those in the conservative mask. For Geostationary Operational Environmental Satellite GOES-16, probability of detection (POD) at an OT probability >0.5 ranged from 0.51 to 0.95 and false alarm

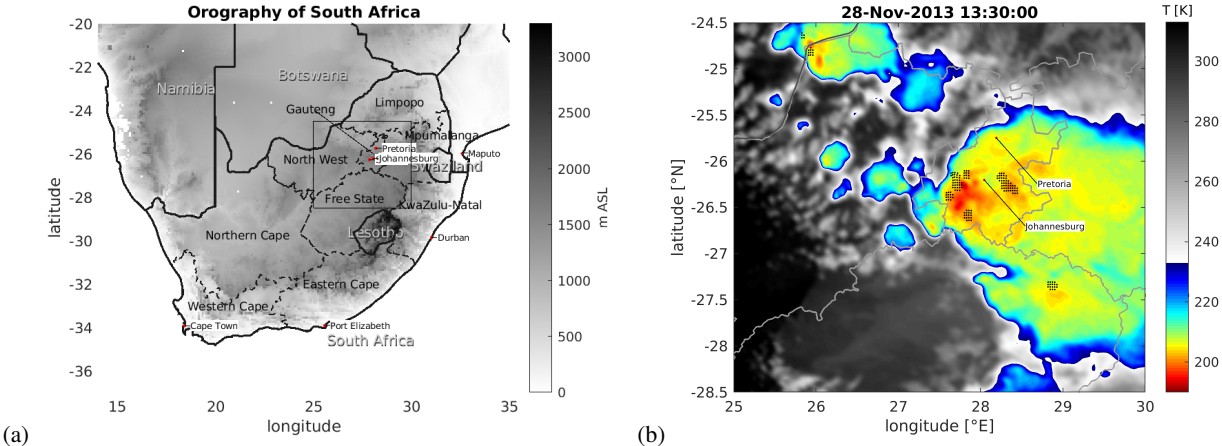

**Figure 2.** (a) Map showing the relief of South Africa from the Shuttle Radar Topography Mission (Farr et al., 2007), provinces and major cities. (b) Meteosat IR image showing convective storm activity on 28 Nov. 2013 at 1330 UTC. Colors represent 10.8 $\mu$m channel brightness temperature and dots indicating detected overshooting tops (OTs, black) for this image. OT detections are parallax corrected here but the underlying IR satellite image is not, leading to some slight displacement of a detection and a corresponding cold region. The anvil cloud in yellow and green colors is automatically detected using the IR anvil detection index greater than 10.

ratio (FAR) ranged from 0.04 to 0.24, with highest POD for the conservative mask and lowest FAR for the liberal mask. For GOES-13, POD decreased to 0.8 for the conservative mask but remained nearly the same as GOES-16 for the liberal mask. FAR increased by 0.10 for the liberal mask. In Cooney et al. (2021), an OT probability of 0.5 corresponded to a median 20 dBZ precipitation echo top near the tropopause and a FAR ranging from 0.1 to 0.5 depending on the reflectivity level used to define the precipitation echo top (e.g., 10 or 20 dBZ), the height of the echo, and the satellite data used as input (e.g. GOES-13 vs GOES-16). POD based on these comparisons with echo tops ranged from 0.35 to 0.75. In summary, even the most prominent OTs are less evident and harder to detect in the GOES-13 data. Given the reduced prominence from coarser resolution, OT detection algorithm sensitivity settings must allow detection of smaller temperature gradients within anvils, which results in increased FAR.

False OT detections in very cold outflow near to actual OT regions is the most common source of error. Despite these false alarms, which in our opinion are impossible to completely eliminate, the Khlopenkov et al. (2021) OT detection method improves upon the Bedka et al. (2010) version used in previous hailstorm climatology studies, and will represent the convection climatology across South Africa quite well.

Imagery of the Meteosat Second Generation (MSG) Spinning Enhanced Visible and Infrared Imager (SEVIRI) instrument (Schmetz et al., 2002) between January 2005 and December 2018 is scanned for OTs at a temporal resolution of 15 minutes. Since South Africa is not continuously covered by the high resolution visible imagery product of MSG, only the IR channel data (given as 10.8 $\mu$m brightness temperature) is provided to the detection algorithm. An example is shown in Fig. 2b, where brightness temperature during a very strong hail event on 28 November 2013 and detected OTs are displayed.

The spatial distribution of OT pixel detections across the entire study domain and 14-year study duration is depicted in Fig. 3a. Clearly, convective storms are most common in the prevailing moist subtropical climate in east South Africa, along

the Great Escarpment, including the south-eastern flanks of the Drakensberg and stretching north through the Mpumalanga province (see Fig. 2a and Sect. 2.1). Here the complex terrain with a height of more than 2 300 m asl induces uplift to serve as a trigger for convection initiation. By contrast, OT frequency decreases towards the west and towards the coast of the Indian Ocean, where the climate is mainly semi-arid to desert.

Compared to Dyson et al. (2020), we note the absence of an OT frequency maximum over the country of Lesotho, even in the
150 unfiltered OT data, and higher values to the north and southeast. We attribute these differences to the coarse spatial resolution of the ERA-Interim reanalysis used in the above mentioned study, which is likely insufficient to resolve local orography. The high altitudes in southern Lesotho (mostly 2500–3500 m asl, cf. Fig.2a) seem to suppress deep convection to some degree, similar to the situation in the interior of the Alps in Europe (Punge and Kunz, 2016; Nisi et al., 2018).

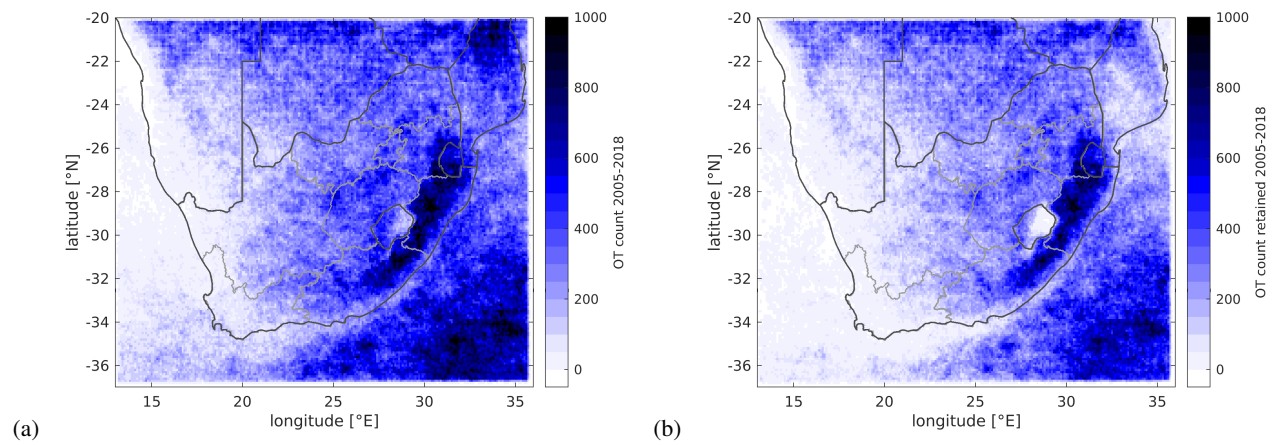

(a)                                            (b)

**Figure 3.** Map of (a) all OT pixel detections in the period 2005–2018 over continental South Africa and neighboring nations and seas (11 143 479 OTs) and (b) OT pixels retained after filtering for atmospheric conditions (8 272 509 OTs; see discussion in Sect. 2.6)

.

### 2.3   Hail reports and insurance claims

Reports of hail observations including estimates of hail sizes are registered in several continental-scale, centralized databases for North America, Europe and Australia. These reports are very helpful for validating the severity of the storms with detected OTs. In addition, derived hail size spectra are required as as a measure of intensity in hail risk models. For South Africa no such database of comparable extent is available. However, comparing studies that estimated hail size spectra (e.g., Sánchez et al., 2009; Eccel et al., 2012; Dessens et al., 2015; Grieser and Hill, 2019), one can conclude that hail size spectra do not tend
to vary greatly among regions and even continents. For this reason, we computed hail size spectra for the stochastic modeling component of this work from 26 884 hail reports archived by the European Severe Weather Database (ESWD; Dotzek et al.,

2009) for the period 2005–2019, and from 3 764 reports provided by the Severe Storms Archive of Australia's Bureau of Meteorology for the period 1950–2019. Reporting policies meant that events of hail diameter of 2 cm or more are covered, but in some cases reports with smaller stones accumulating to thick layers are included. A uniformly distributed random value

between -0.5 and +0.5 cm was added to each reported hail diameter to compensate for rounding in the hailstone measurement process and to obtain a smooth distribution.

In addition, 1 423 hail damage claims between 1984 and 2017 including a reference to a location were obtained from several insurance companies of South Africa (see Fig. 4). Data did not include information on the size of hail, hour of occurrence, or type of asset affected. Claims data tends to be biased towards population centers and – in this case – towards major hail days,

but still has the advantage to provide direct evidence for hail occurrences.

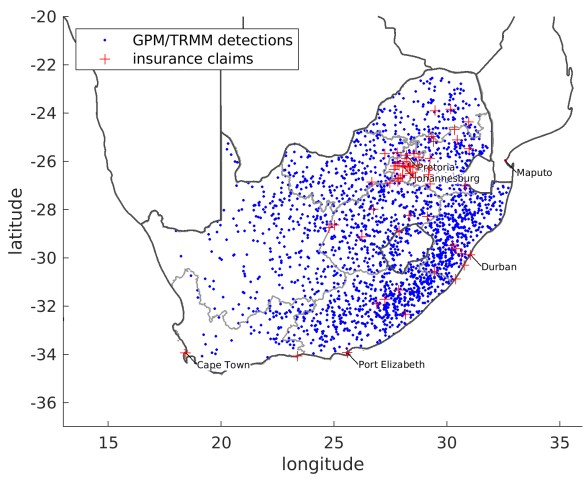

**Figure 4.** Claims locations of hail damage in South Africa (1984–2017) and passive microwave detections (1998–2018) in the model domain.

## 2.4 ERA-5 reanalysis data

ERA-5 (Hersbach et al., 2020) is the 5th generation reanalysis of the European Centre for Medium-Range Weather Forecasting (ECMWF). It is a global observation-guided model representation of past weather. Data were obtained for the period 2005–2018 at a spatial resolution of 0.25°and hourly resolution. While CAPE and the height of the freezing level are provided as

output variables, bulk wind difference between the near-surface (10 m) and 6 km above the ground (0–6 km wind difference) was computed based on pressure level data using linear interpolation.

## 2.5 Passive Microwave hail retrievals

Scattering of surface emitted microwave radiation by hailstones is an alternative method of hailstorm detection by satellites (Cecil, 2009). The measurement principle has the advantage that in contrast to OTs the signal is directly caused by hailstones,

but at the cost of spatial and temporal coverage, commonly limited to two overpasses per day, and at the risk of nonuniform beam filling of the field of view. Beam sizes of current generation sensors (e.g., Petty and Bennartz, 2017) are on the same order of magnitude as convective storm core diameters, meaning that sensor and storm need to be align for successful detection.

In this work, such detections from the Tropical Rainfall Measuring Mission (TRMM) and the Global Precipitation Measurement Mission (GPM) satellites (Bang and Cecil, 2019, 2021) over the South African domain were evaluated for the period 1998–2019. Cases with a hail probability greater than 10% over South Africa and Lesotho were retained. The threshold here is lower than in the studies cited above (threshold of 20%) because our intension is to consider also small hail diameters and not just significant hail. Along with the damage claims, they are used to compare hail-prone environments in South Africa to those in Europe and Australia (Punge and Kunz, 2016; Bedka et al., 2018b), and to constrain OT detections to a certain range of convection-related parameters from ERA-5 reanalysis (see next section).

## 2.6 OT filtering by conditions from reanalysis

Before clustering the OT detections, a filter is applied based on surrounding atmospheric conditions in terms of wind shear and CAPE obtained from ERA-5 for days and locations with microwave hail detections and damage reports and (Fig. 5). Since insurance data is only available for South Africa (see Fig. 4), the filter criteria are also only determined for the territory of South Africa and Lesotho. Note that the purpose of this filter is distinct from other studies aiming to identify hail-prone conditions from reananlysis (e..g., Taszarek et al., 2020; Dowdy et al., 2020; Prein and Holland, 2018), which tend to suggest much stricter criteria.

The filter design used for Europe (Punge et al., 2017) and Australia (Bedka et al., 2018b) was retained in this work. For the filtering, thresholds for the different ambient parameters were defined based on the OT distribution for insurance loss data and microwave hail detections. The thresholds are then determined from the 2th and 98th percentiles of the respective distribution functions. OT detections that occur outside that range are then removed to obtain the filtered OT dataset.

Ambient conditions near OT detections are interpolated spatially from the much higher resolved ERA-5 reanalysis instead of ERA-Interim (25 km rather than 80 km). In contrast to the latter, ERA-5 has hourly rather than 6-hourly fields, so values at the full hour are used for OT or microwave detections in the following 60 minutes. This reduces false filtering due to model uncertainty to resolve, for example, the rapidly evolving CAPE field with a strong diurnal cycle. For insurance loss data without time of occurrence, we have chosen 12 UTC in ERA-5 by default.

Bulk wind difference (near-surface to 6 km) and freezing level for both microwave hail detections and insurance claims in the vicinity of OT detections are shown in Fig. 5 for South Africa and Lesotho. OTs occur at somewhat lower 0–6 km wind difference and higher freezing level compared to microwave detections, confirming the filter choice in Punge et al. (2017). Note that microwave hail and OT detections occur most frequently at a 0–6 km wind difference between 10 and 20 $\mathrm{ms}^{-1}$, which represents the lower limit for organized convective storms such as multicells, supercells, or mesoscale convective storms (MCS) to occur (e.g., Markowski and Richardson, 2010). Using a higher threshold would exclude a relevant fraction of situations where hail is likely based on the microwave algorithm. Damage reports often occur at 0–6 km wind difference between 20 and 30 $\mathrm{ms}^{-1}$, indicating a bias towards the more organized storms producing more damaging hail. But given the high concentration

of these claims in populated regions, we refrain from using a more restrictive thresholds based on this data alone. The somewhat
odd distributions of damage reports shown in Figs. 5a and b are due to heavily population-biased sampling locations. As 9.5%
of the OTs, but only 3.5% of the microwave hail detections and 2.5% of the claims occur at a melting level of less than 2 400 m,
this altitude was used for the lower threshold with this parameter.

OT detections are thus retained if the surroundings of a given OT fulfill minimum conditions of convective instability (CAPE
>100 $\mathrm{J\,kg^{-1}}$), 0–6 km wind difference (>1.5 $\mathrm{m\,s^{-1}}$) and the height of the melting level (>2 400 m, <4 845 m agl). As can
be seen in the spatial distribution of the filtered OTs in Fig. 3b, the filter removes OT detections in particular over the ocean,
central Moçambique and the Lesotho Highlands. The latter feature is due to the minimum freezing level condition and remains
to be confirmed by independent observations, for example, by hailpad or hail sensor data from the region.

A similar pattern of hail events shown in Fig. 3b is found in the global hail study by Prein and Holland (2018, Fig. 11) as
well as in passive microwave data by Bang and Cecil (2019, Fig. 7; see also the discussion in the Appendix). A final judgment
on the actual occurrence of significant hail on the ground would require surface observation data, such as hailpads or sensors
covering multiple regions of South Africa. Such direct observation data, however, are not sufficiently available for the entire of
South Africa.

All subsequent analyses presented in the next sections are based on the filtered OT detections.

## 2.7 Event definition: Grouping hail activity

Figure 6 illustrates the definition of historic potential events based on observed and filtered OT detections. These events are
formed by computing the spatial and temporal distance between individual OT detections. OTs are assigned to the same event
if they are separated by less than 1 hour and less than 30 km. This simple approach can detect both single cells and other more
organized forms of convection including MCSs or squall lines. Event centroids are defined as the mean latitude and longitude
of the event, or grouped, OTs. An event is approximated by an ellipse and characterized by its length, width, orientation relative
to the meridian, as well as the fraction filled with OTs. In addition, we also considered the highest OT-anvil mean temperature
difference among the OTs assigned to an event as a criterion for storm severity (see Sect. 3.2). Lifetime and propagation speed
are estimated based on the initial and final OT detections within an event. Events can overlap when several storms pass over the
same region on a given day. This actually happened in the example on 28 November 2013, shown in Fig. 6. The three events
overlap, but we can assume that the activity of the smaller ones is not related to the main event, since the convection occurred
several hours later when the main storm had already moved away, and also was much weaker.

The event definition criteria are more restrictive compared to Punge et al. (2014), so the events are better constrained to zones
of possible hail activity. Events made up by OT detections only at a single time step, hence lasting for less than 30 minutes, are
neglected. In total, 33 820 events were identified from the total of 8 272 509 filtered OT detections for the entire 14-year study
period. This means, one event on average contains 245 individual OT detections.

Histograms of duration and propagation speed of SCS events – not considered in Punge et al. (2014) – are shown in Fig. 7.
The distribution of duration is exponential in shape, in line with radar-based studies (Schmidberger, 2018; Fluck et al., 2021).
While duration is roughly proportional to length, it becomes clear that propagation speed varies over a wide range. Most fre-

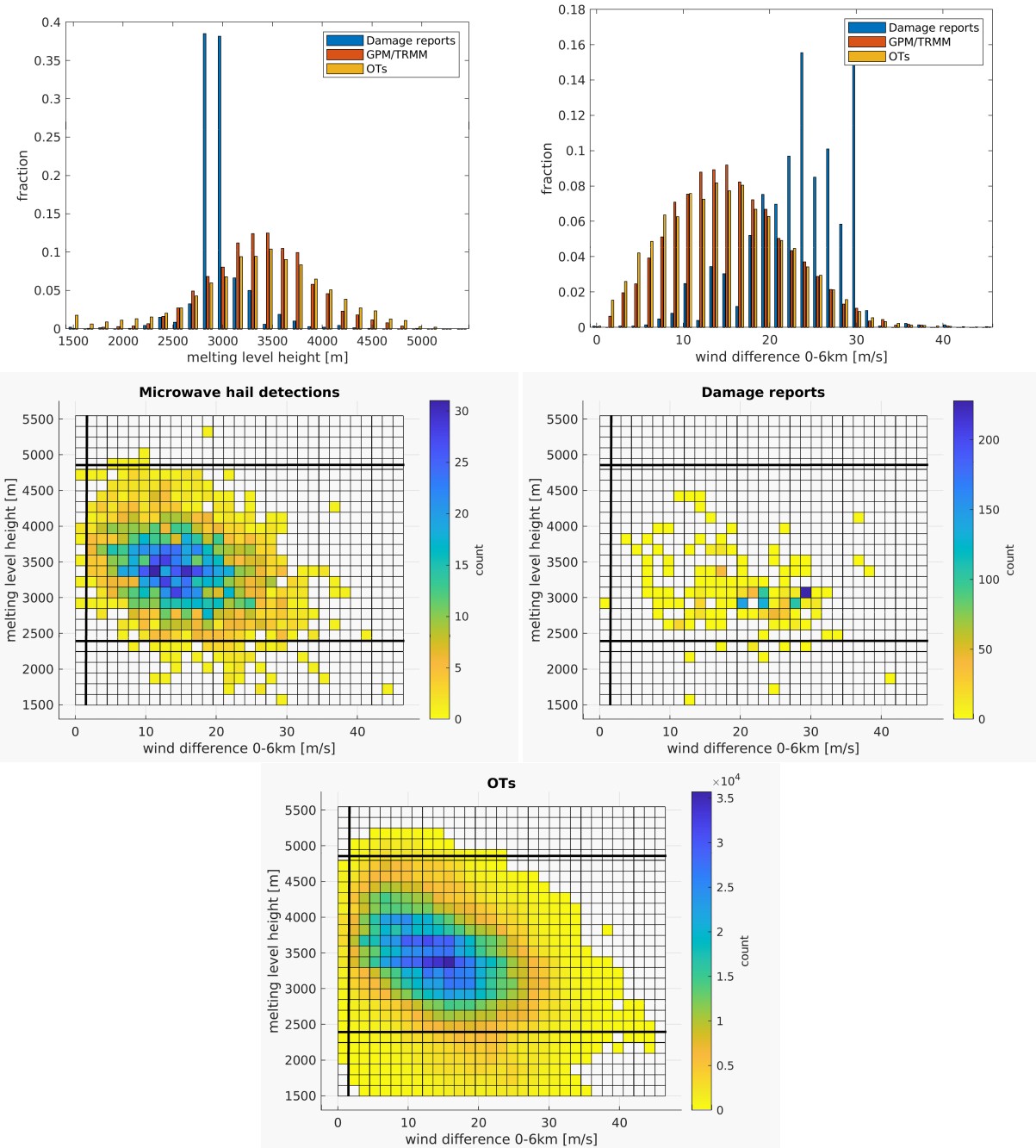

**Figure 5.** Distribution of ERA-5 0–6 km wind difference and melting level height in hail claims (1984–2017), passive microwave detections (1998–2019), and OTs (2005–2018) for South Africa and Lesotho. Top row: probability distribution; Bottom row: bi-variate histogram for 0–6 km wind difference and freezing level (bold lines indicate the selected filter thresholds; see text for further details)
. Only situations with positive CAPE in ERA-5 were used for these computations.

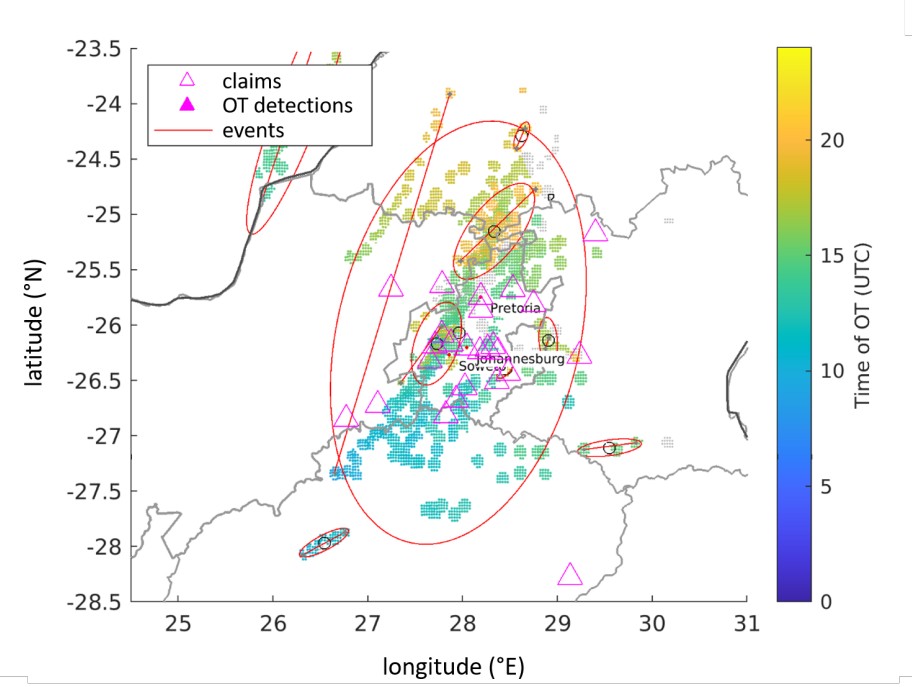

**Figure 6.** Convective storm activity in the Gauteng region of South Africa on 28 Nov. 2013. Filtered OTs (gray dots), retained OTs (color-coded by time), hail claims (pink), and event definition (ellipses). Hail claims (magenta) indicate the location of successive hailstorms, split by the algorithm into three events, for which ellipses show the spatial extent. A line connects the locations of the initial and final OT locations, determining the orientation of the ellipse. Two smaller events overlap the main event centered over the Gauteng region, as the associated cells developed at a later time, separately from the main activity.

quently, the speed ranges around 30–35 $\mathrm{km\,h^{-1}}$ ($\approx$8–10 $\mathrm{m\,s^{-1}}$), slightly lower than the range of 10–30 kt ($\approx$ 18–55 $\mathrm{km\,h^{-1}}$) found by Carte (1966). Very high values beyond 100 $\mathrm{km\,h^{-1}}$ are explained by cases where convection is triggered simulta-
neously over a larger area and OTs from several storms are unintendedly grouped to an event. Such unrealistic events can be simply excluded in the event set.

## 3 Stochastic modeling and event properties

In this section we describe the spatial distribution of SCS detections for the hail hazard model as well as additional event properties. This section refers to events of grouped OTs according to the event definition in Sect. 2.7. Because we cannot be
assured that all historic events identified by satellite data and filtered through the ERA-5 reanalysis are associated with hail on the ground, these events are hereinafter referred to as potential hail events. Distribution functions are used to approximate the historic set of potential events presented in the previous section. Stochastic events are generated using these distribution functions for relevant event properties, and 14-year samples from the stochastic event set are compared to historic data such

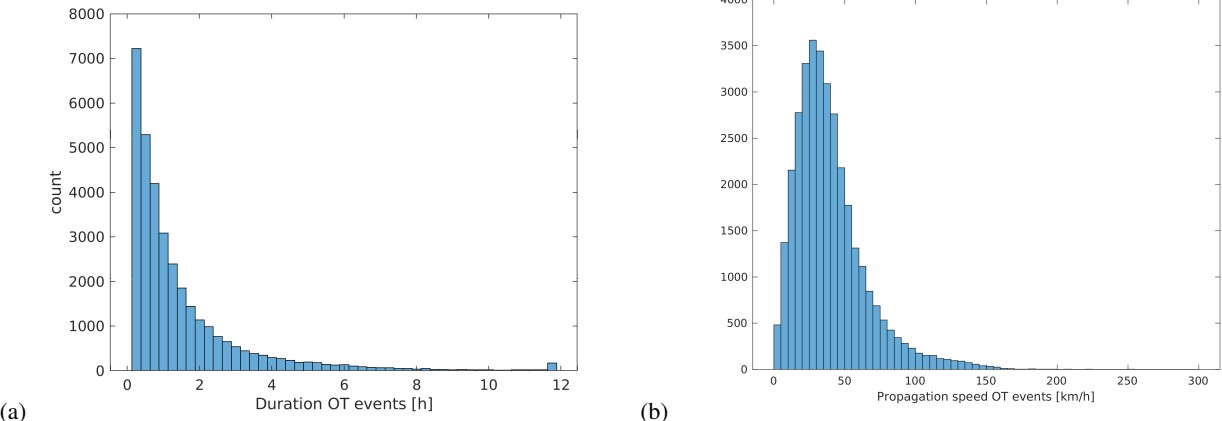

**Figure 7.** Histogram of hail-filtered OT event (a) duration and (b) propagation speed, 2005–2018.

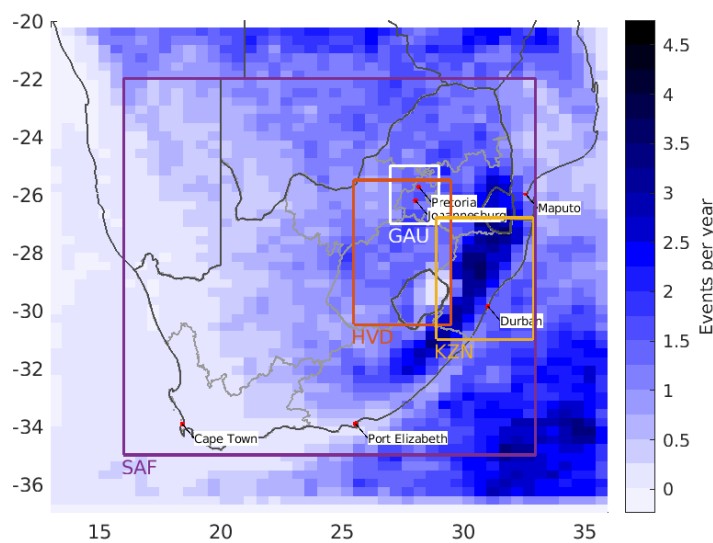

**Figure 8.** Event frequency distribution used for stochastic modeling. The boxes show different regions discussed in Sect. 3.6: The Greater South Africa domain (SAF, purple box; for simplicity defined as the area between -35 and -22° N, and 16 and 33° E), the Highveld (HVD, red box) and KwaZulu-Natal (KZN, yellow box)

as the most populated and most storm-affected regions, and the Gauteng (GAU) as the largest contiguous urban area with a significant concentration of assets.

as those available in the existing databases. The full stochastic event set covers 25 000 years with a total of 21 093 957 events 260    spanning 3 442 346 days. The key challenge in generating the stochastic event set is to ensure conservation of event properties,

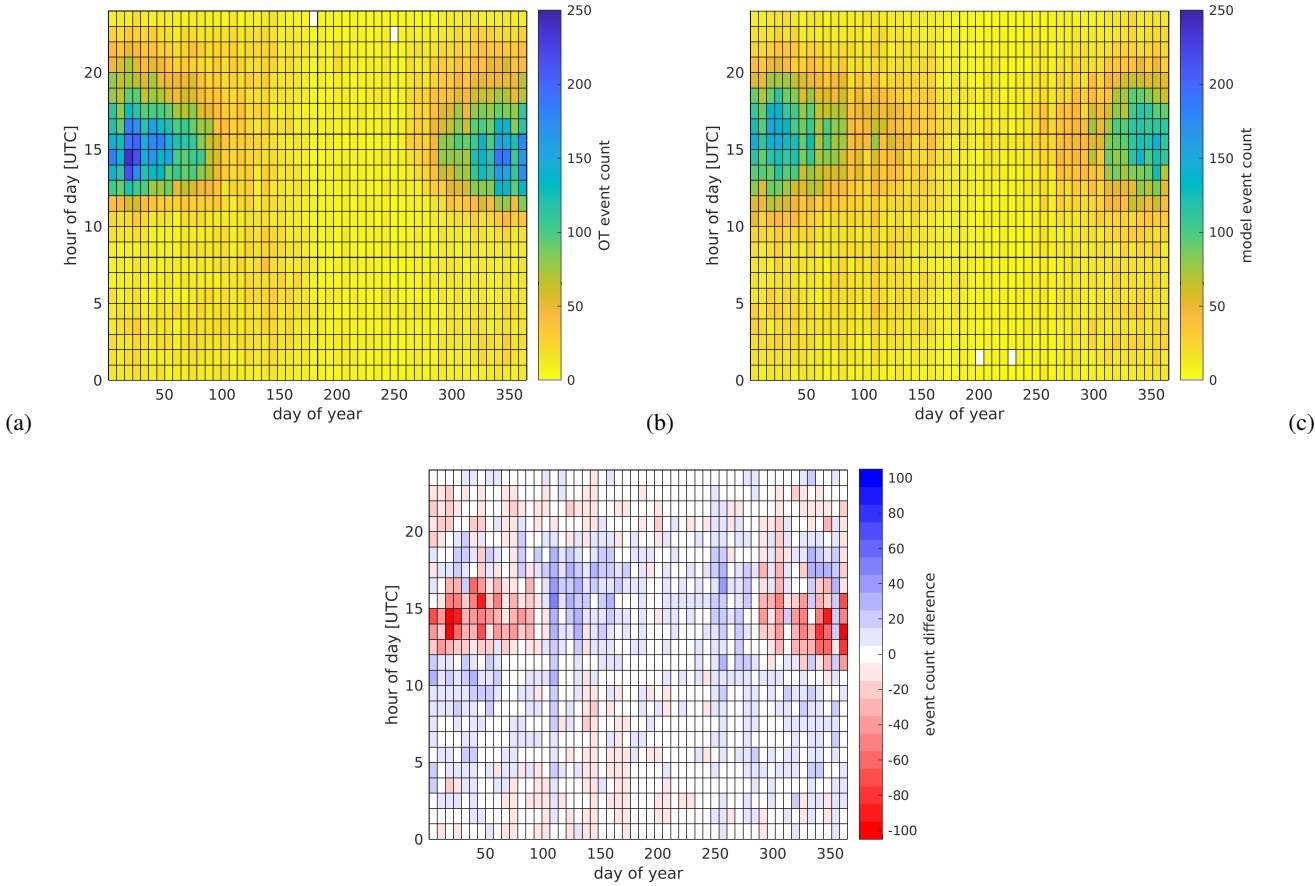

**Figure 9.** Distribution of potential hail events in time of day and day of year: (a) historic events 2005–2018; (b) stochastic model events; and (c) difference between the two for the events shown in Fig. 8. Days of the year are counted from January 1st. The grid shown in all Figs. represents 1 hour and 7 days.

their inter-relationships, and the spatial distribution of the historic potential hail events. This point is addressed by drawing from historic distributions using correlated random numbers where required.

The spatial distribution of potential hail events is obtained by normalizing the annual frequency of filtered OT detections, counted on a rectangular $0.3° \times 0.5°$ (lat/lon) grid with the average number of OTs per event (Fig. 8). This grid was chosen to retain spatial details of OT occurrence due to atmospheric processes and mechanisms, such as orographically induced lifting, but dampen local accumulations of OTs on individual pixels caused by occasional passage of multiple storms at the same location. The implicit assumption with this method that potential hail events are distributed in the same way as OTs implies a certain amount of additional smoothing that can be neglected at scales larger than the event length.

Following Punge et al. (2014), both annual and diurnal cycles are modeled with Gaussian distributions. Approximate normal distributions were found, for example, by Allen et al. (2015) for hail reports in the US, or by (Taszarek et al., 2018) using

ESWD reports across Europe. Even if in a few regions the annual and diurnal cycles deviate from a normal distribution due to climatological peculiarities, this distribution is plausible on average over a large area. To determine the day of year, larger grid cells of $3° \times 5°$ are considered, grouping $10 \times 10$ of the smaller grid cells used above such that the number of observations in each cell is sufficient to derive characteristics of the distribution. Depending on the mean number of OT events at a certain
location, either a mixture of two Gaussians – to accomodate for two peaks in spring and autumn – or a simple Gaussian distribution is fitted to the data. The relatively large grid was chosen to ensure a stable solution when fitting this complex distribution function, also in regions with few OT events. For each batch of stochastic events representing 250 years, the following procedure is applied: Days are drawn from each larger grid cell of $3° \times 5°$ distribution for $N$ events in this cell and also from the respective 8 surrounding grid cells of the same size, yielding nine times the required number of events. To mimic
the grouping of potential hail events in severe days, the same day is then attributed to blocks of $N^{1/3}$. The exponent of 1/3 was chosen to best fit the observations (1 would have all events on one day, 0 would keep the original date for each event). Finally, the day is retained for $N$ of these events at random. The spatial smoothing technique described above effectively introduces averaging on a scale of $9° \times 15°$, which approximately represents the scale of synoptic processes and flow patterns governing the spatial (and temporal) clustering of SCS, for example, by specific weather regimes such as Baltic blocking (e.g., Mohr et al.,
2019, 2020). This procedure requires only one tuning parameter and has been found empirically to approximate the observed space-time distribution of days in a satisfactory manner.

The hour of day is determined in a similar way from the distribution on the larger $3° \times 5°$ grid. Times are drawn from this distribution for the $N$ potential hail events considering again a larger region of $10° \times 6°$ around the grid cells center and retained with a chance of 1/4. Again, the grid cell dimensions may appear arbitrary, but have been carefully chosen to capture
observed spatio-temporal variability. In fact, the time of day is spatially correlated at a smaller scale because in a series of potential hail events the later ones are spatially shifted with respect to the earlier ones.

Figure 9 shows the daily and seasonal variation of the (a) observed potential hail events and (b, same duration – first 14 years) the modeled events. Seasonal hail activity estimates based on OT activity (2004–2018) and passive microwave hail retrievals (1998–2018) are presented in Fig. B2 in the Appendix. A clear maximum of the events is found in austral summer (December
and January) in the afternoon around 15 UTC. Also note the secondary local maximum in the historic OT event set at around day of the year 140 and at 06 UTC (discerned as an area of orange shades in that region in Fig. 9 a), which is also represented in the model. For the Highveld region, Smith et al. (1998) report a somewhat earlier maximum of hail events in November and December. Indeed, we find the convective season peaks around this time in the Highveld and KwaZulu-Natal, and in the first half of November over the Gauteng (not shown). Over the Southern Ocean, the peak occurrence is shifted towards fall.
Off-shore events need to be represented in the model as they can extend to the coastal region. An on-shore impact of these far off-shore event is quite unlikely and will be marginal at this distance.

Peak time of day is between 14 to 15 UTC, or 4–5 pm South African Standard Time, slightly earlier than in Smith et al. (1998, 5–6 pm), but consistent with Olivier (1990) (see also Fig. B1 in the Appendix). The diurnal cycle is most pronounced in summer. The temporal climatologies of historic (Fig. 9a) and modeled (Fig. 9b) events match rather well. In the histograms

shown in Fig. 10a and b, most of the modeled events fall within the error bars of the observed events, quantified from seven independent samples over 14 years.

## 3.1 Geometric event properties

**Table 1.** Potential hail event property distributions and parameters (computed with the standard matlab *mle function).*

| variable | distribution | shape parameter | scale parameter | location parameter |
|---|---|---|---|---|
| length $l$ [km] | GEV | 0.57 | 26.68 | 37.33 |
| width $l$ [km] | GEV | 0.90 | 7.92 | 13.00 |
| event-to-storm area ratio $1/f$ | GEV | 0.20 | 1.26 | 2.36 |
| hail size $d$ [cm] | Gamma | 3.70 | 0.83 | |

Both length $l$ and width $w$ distributions, determined from the observed OT events as described in Sect. 2.7, decay rapidly with increasing values (histograms shown in Figs. 10c and d). In contrast to previous model versions, we choose to approximate
both properties with generalized extreme value (GEV) distributions rather than exponential distributions. This improves their overall fit, but particularly at the lower end of the range. The good match represents a use case for the GEV function family beyond extreme value theory. Positive shape parameters $\kappa$ for length, width, and event-to-storm area ratio (Table 1) indicate that the GEV converges to a Frechet distribution, also known as Fisher-Tippett Type II distribution. These distributions exhibit what are called heavy tails, meaning that the PDF decreases rather slowly for large values of $x$ (Wilks, 1995). Consequently,
the GEV tends to give too large values, which is why length and width were truncated at 1.5 times the largest observed values at which events effectively cover the entire country (1 445 km $\times$ 677 km). In addition, low widths are somewhat over-represented, which can be attributed to the design of the event definition procedure for historic potential hail events.

The fraction $f$ of the event area – the area of the ellipse spanned by major and minor axis of lengths $l$ and $w$, $\pi/4\,l\,w$ – covered by OT events is also modeled using the GEV (histogram for the logarithm of the effective event area shown in
Fig. 10e). Here we fitted the GEV to the inverse of that fraction, the event-to-storm area ratio. This function was found to match observations well as the inverse has no upper bound. Furthermore, very OT-sparse events are extremely rare in the event catalogue. For the highest class ($> 10^5$ km$^2$), the fraction in the stochastic set is significantly higher, while the match is otherwise satisfactory. Table 1 lists the distributions and parameters for these event properties.

The orientation, or the direction of the major axis of an event, generally aligns with the direction of propagation. We find that
most frequently events have an orientation of around $100°$, i.e., propagate eastward to southeastward (Fig. 10f) most frequently. This, however, applies only for the whole country, but not for all regions such as the high hail fall region Gauteng, where storms preferably propagate in north-easterly directions. Stochastic modeling was performed using a von-Mises-distribution function (Wilks, 1995).

For the sake of completeness, Fig. 10g shows the distribution of maximum hail diameters $d^{max}$ modelled with a gamma
distribution function. As can be clearly seen, the distribution has an exponential decay for sizes larger than 2 cm. The decrease

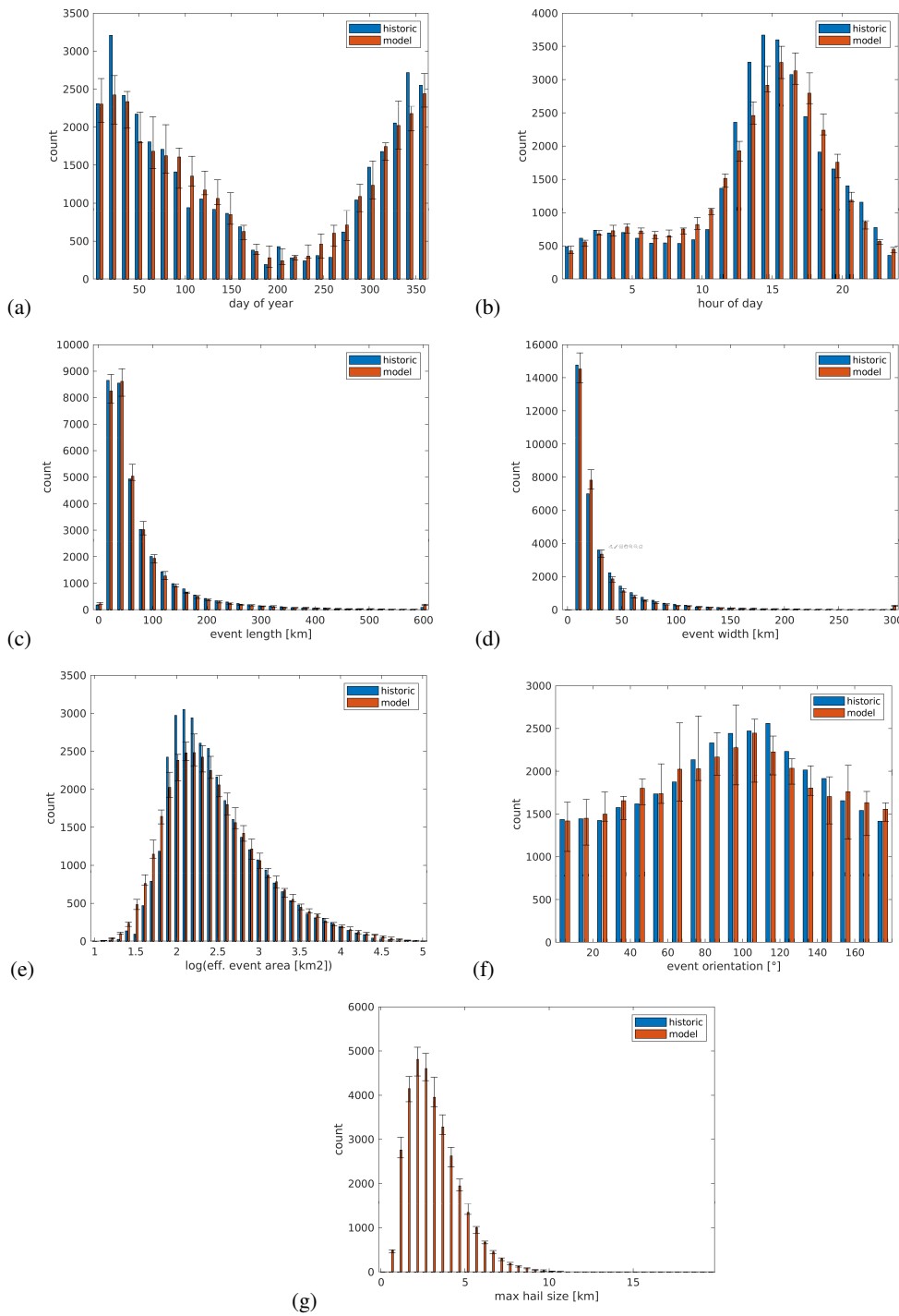

**Figure 10.** Distributions for properties of historic and stochastic events. With modeled events, (error) bars indicate the median (spread) among 7 independent samples of 14 years' length. Each set of bars represents 14 days, 1 hour, 20 km in length, 10 km in width, 0.1 on the log scale of part (e) and 10°in orientation. Since hail severity information is essentially unknown for the historic events, blue bars are missing in (g).

in hail counts for smaller diameters is only due to reporting practices and does not reflect reality. However, small hail is not relevant to hail damage and risk.

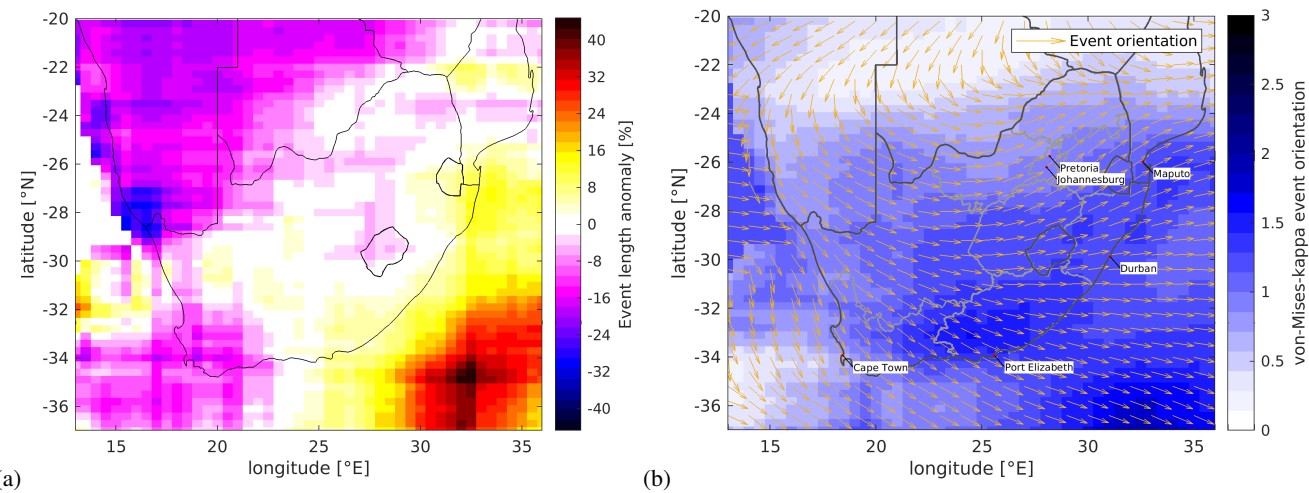

**Figure 11.** (a) Spatial variation of mean event length (%) and (b) spatial variation of event orientation, location parameter (arrows) and the shape parameter $\kappa$ of the von-Mises distribution derived from the observed OT events.

Because event properties vary across the South Africa domain, a box-window average over a 2.5° window is used to estimate these variations and to scale event properties from the historic event set. Fig. 11a shows the spatial variation of event length, estimated using maximum likelihood estimation. Here the objective function is the negative logarithm of the product of the probabilities of each sample length given the assumed (GEV) distribution of lengths. The distribution parameters were then varied to find the maximum of the objective function.

Fig. 11b represents the most common orientation, obtained by fitting a von-Mises-distribution (Mardia and Zemroch, 1975) to the regional events on a national scale. Events occurring offshore, where MCS occur more frequently than over land (Feng et al., 2021), tend to run longer, whereas events are shorter in the western part of the domain, hinting towards less organized forms of convection. Orientation varies from south-eastward in the south-west to north-eastward in the north-east and is most aligned over the Eastern Cape region. Garstang et al. (1987) found winds from the north-westerly sector to prevail at 850 hPa on convective storm days in north-eastern South Africa, but from a much smaller sample. The larger spread in orientation towards the north and west can be explained by (i) the prevalence of storm systems affecting larger regions at the same time, hence grouping multiple parallel storms, as well as by (ii) small, quasi-stationary events whose orientation does not reflect an influence of storm propagation.

## 3.2 Storm severity indicator

At present, SCS intensity cannot be estimated accurately from geostationary satellite measurements alone, even if Marion et al. (2019) suggest that storm's tornadic intensity is related to OT area. In a recent study, Khlopenkov et al. (2021) showed that the difference between OT temperature and the tropopause temperature and the OT temperature and surrounding anvil in general is larger for significant severe ($\geq$ 5 cm) hailstorms than storms with smaller hail reports. But, because of the large uncertainty involved in these relationships, hail sizes estimated from the temperature differences are not further considered in our stochastic modeling approach. To at least separate days with strong convection from less convective days, we determined a severity indicator based on the temperature difference between the overshooting top and surrounding anvil. This extent can be somehow related to the strength of the updraft supporting hailstones growth (Marion et al., 2019; Khlopenkov et al., 2021; Lin and Kumjian, 2022). To quantify the hail severity indicator, first the temperature difference between the OT and the surrounding anvil is computed. The severity index is then assigned to the highest such temperature difference of all OTs comprised within an event (see also Discussion in the Appendix).

## 3.3 Hail size distribution in stochastic modeling

For the estimation of the damage and the risk, the hail model needs as intensity metric the hail size. Because OT data do not allow us to estimate reliable hail sizes (see previous Sect. 3.2), this quantity is only considered and implemented in the stochastic part of the model based on the hail size distributions obtained from severe weather reports. Since hail size observations in South Africa are very rare, we assume that the maximum hail diameters follow the same distribution as large hail diameters recorded in the databases of ESWD and Severe Storms Archive (see Sect. 2.3). Hail sizes for all hailswaths within an event are derived from the attributed largest hail diameter. This means that hailswaths with small hail sizes are more frequent in the stochastic model than the distribution derived from the database of maximum diameters would suggest. That is also the case in reality, as hail with a small diameter is less likely to be recorded in a database than large hail. The hailstone size distribution used in the stochastic model is assumed to be exponential as suggested by various authors using hailpad data (Sioutas et al., 2009; Berthet et al., 2011; Grieser and Hill, 2019). As we do not know the hail spectra for South Africa, this uncertainty remains in the model. However, calibrating the final hail risk model using historical loss events, this uncertainty in the result can be reduced to a large degree.

## 3.4 Length-severity correlation

An important feature of the stochastic hail model is that it conserves the relation of different properties of the historic potential hail events. The most relevant properties that are correlated among each other are event length $l$, event width $w$, event severity, and fraction $f$ of event area covered by potential hail streaks (see also Punge et al., 2014).

The scatter plot of length and width (Fig. 12a) confirms that correlations between these event properties are conserved in the model. As in Punge et al. (2014), the correlations between length and width to the storm severity indicator – represented by minimum OT temperature difference for historic events as described in Section 3.2 – are likewise conserved here. By the same

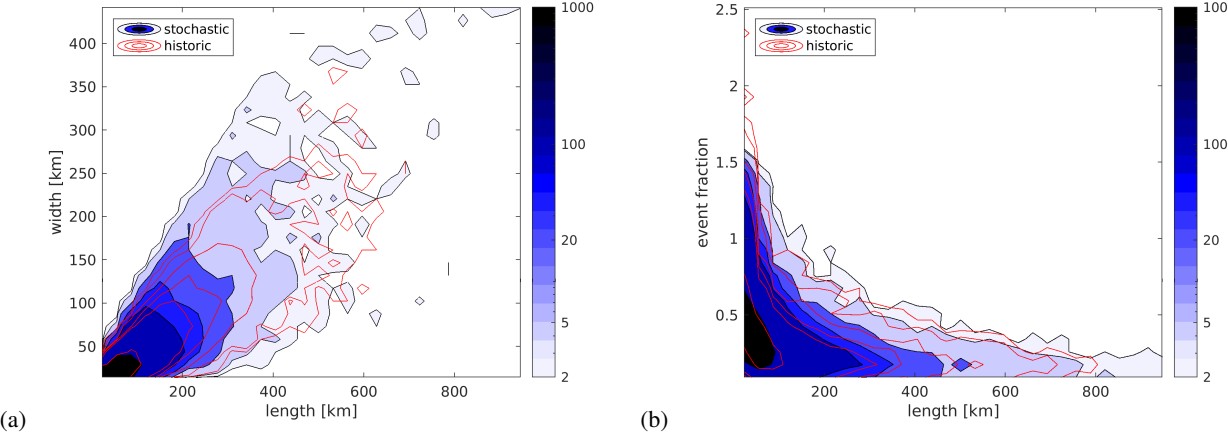

**Figure 12.** Scatter plot of historic and stochastic event properties: (a) length and width, and (b) length and fraction of event covered by potential hail events.

method, the fraction of the event area affected by potential hail events ('effective track area' determined from the ellipse area

spanned by length $l$ and width $w$; see also Sect. 3.1) was also taken into account. This is achieved by first drawing correlated sets of random numbers for each property from a uniform distribution and determine ranks. Then, for each property, we draw values from the actual distribution, sort them, and attribute to events using the pre-determined ranks (Punge et al., 2014).

   Figure 12b shows the relation of event length $l$ and fraction $f$ area, indicating that in large events, a lower fraction of the area is affected. Table 2 summarizes the Spearman rank correlation coefficients of the four variables considered. Accordingly,

longer hail events tend to be wider, have a higher storm intensity, and a lower fraction of the event area covered by hail streaks.

**Table 2.** Spearman rank correlation matrix of historic OT event properties.

| property | set | length | width | severity | event-to-storm area ratio |
|---|---|---|---|---|---|
| length | historic | 1.00 | 0.89 | 0.39 | 0.75 |
|  | model | 1.00 | 0.90 | 0.39 | 0.49 |
| width | historic | 0.89 | 1.00 | 0.41 | 0.61 |
|  | model | 0.90 | 1.00 | 0.40 | 0.43 |
| severity | historic | 0.39 | 0.41 | 1.00 | 0.17 |
|  | model | 0.39 | 0.40 | 1.00 | 0.16 |
| event-to-storm area ratio | historic | 0.75 | 0.61 | 0.17 | 1.00 |
|  | model | 0.49 | 0.43 | 0.16 | 1.00 |

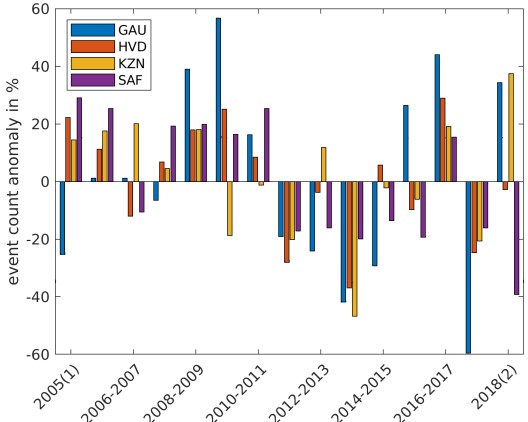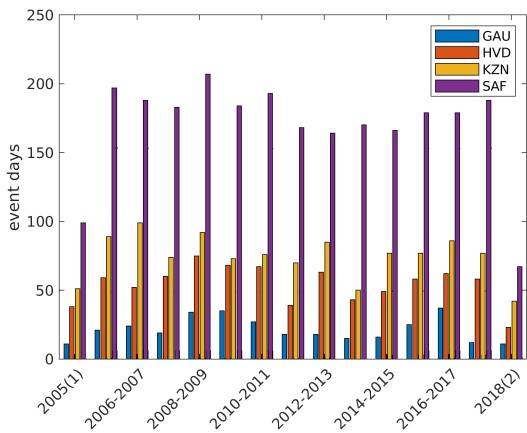

**Figure 13.** Time series of event count anomaly (deviation of the annual sum from the 14-year mean, left) and event days (right) per year (July–June) for regions Gauteng (GAU, regional annual mean event count 35), Highveld (HVD, 152), KwaZulu-Natal (KZN, 181) and South Africa (SAF, 1259). The July–June period is considered more appropriate as the Austral summer season is not split among years. For the 2004/2005 period, only events in 2005 are covered; likewise 2018/2019 covers only the last 6 months of 2018. When computing the event count anomalies for these years, the overall repartition of events to half years was taken into account.

### 3.5  Inter-annual variability

Hailstorm frequency shows considerable year-to-year variation in both the annual number of hail events and hail days (Fig. 13). Even if there is strong correlation between all regions shown in the Figure (i.e., GAU, HVD, and KZN), smaller regions because of the short temporal and small spatial scales of hpotential ail events tend to experience relatively higher variability. This information helps to better understand the year-to-year variability of hail hazard and resulting losses. Note, however, that the year 2013/2014 had the second-lowest event count in the Gauteng region, despite the large damage from the 28 November 2013 event. Figure 14 shows patterns for the first five years of the historic event set (left), along with five years of the stochastic event set for a visual impression of spatio-temporal variability. Overall, the variations in location and intensity of the annual maximum look realistic. The model has fewer events in the north compared to the observed event count as it is based on the scaled OT frequency instead of events, and events in this region contain fewer OTs. The modeled distribution appears overly smoothed at lower rates, but this is unlikely to be a concern.

### 3.6  Intra-annual variability

Quite relevant in practice is the representation of multiple events occurring on a single day, not considered by Punge et al. (2014). In fact, it turns out that only 15 % of the potential hail event days have just one event, whereas on a few occasions, more than 30 events were detected on a single day. Figure 15 shows the number of days with a given number of observed events per day for the South Africa domain (KwaZulu-Natal (KZN), Highveld (HVD), and Gauteng (GAU) regions as displayed in

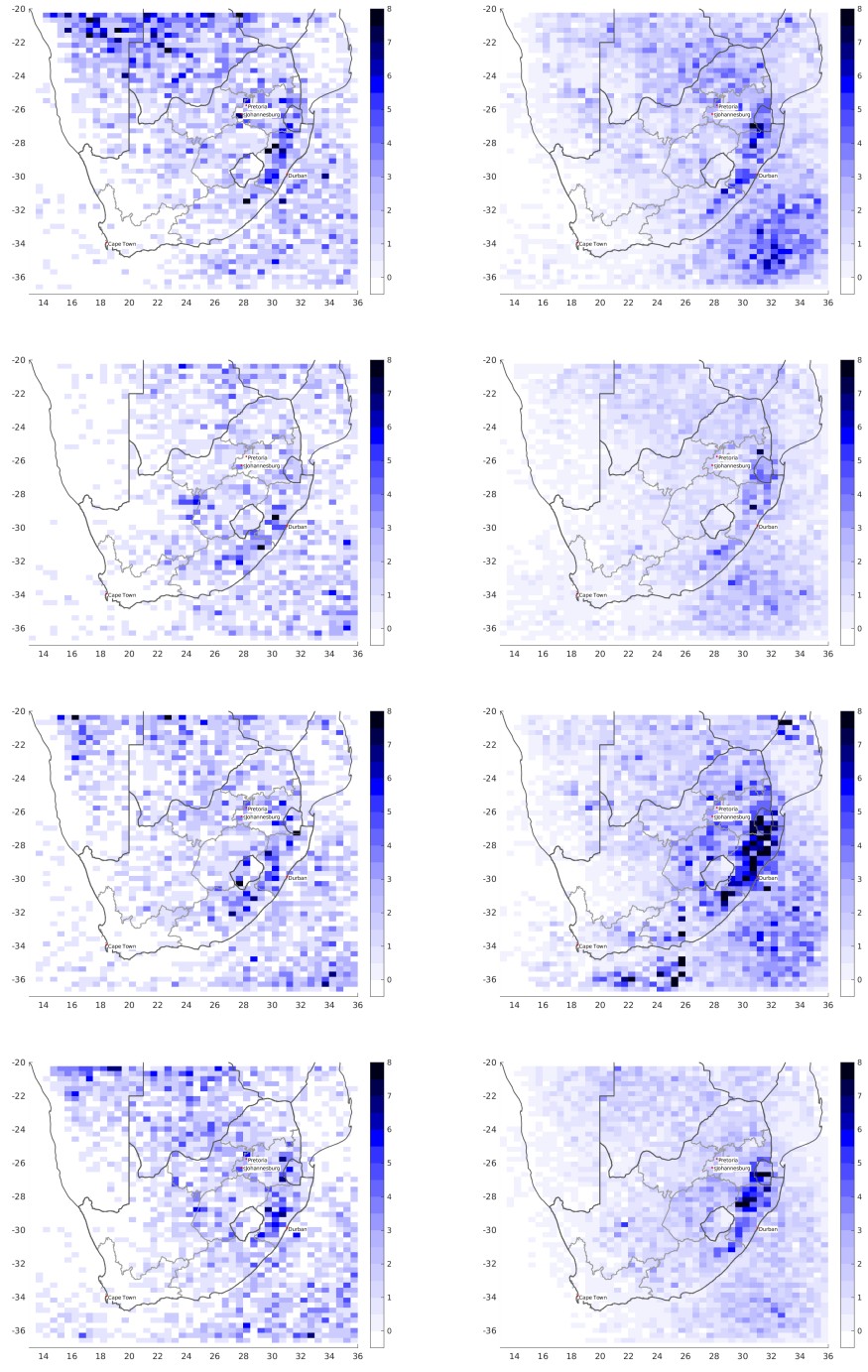

**Figure 14.** Spatial patterns of annual event distribution for years 2005/06 to 2008/09 (left) and in the stochastic event set (right).

Fig. 15). Naturally, the smaller the study area, the lower the respective counts. Over the Highveld, the events are concentrated on a smaller number of days compared to the similar-sized KZN region.

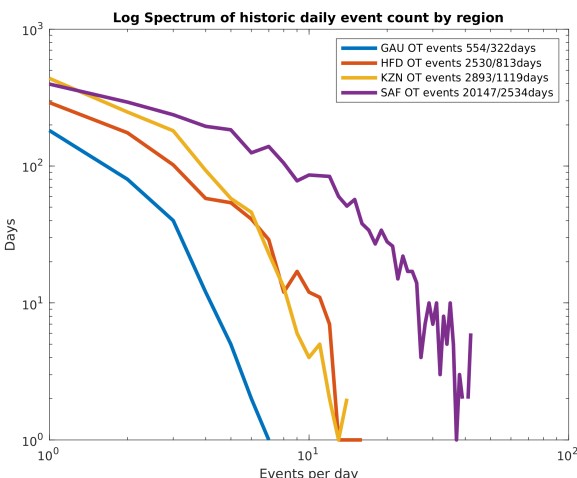

**Figure 15.** Spectrum of number of events per day on historic event days for regions SAF (entire country), HVD (Highveld), KZN (KwaZu-luNatal) and GAU (Gauteng).

The panels in Fig. 16 show the same regional frequency spectra, comparing the historic events to 6 equivalent subsets of the stochastic set. For example, the Highveld region has 2 530 OT events over 813 days. This corresponds to around 180 hail events on 58 days each year. In an equivalent sample of subsets from the stochastic event set, the event count ranges from 2281 to 2 942 (2 523 $\pm$ 232) events on 805 to 893 (838 $\pm$ 32) days. Table 3 summarizes the annual OT and model event statistics for the four regions.

**Table 3.** Annual severe convective storm characteristics for South Africa, estimated from 14-year periods for the Greater South Africa domain (SAF), KwaZulu-Natal (KZN), the Highveld (HVD), and Gauteng (GAU).

| region | OT count | OT event count | Model event count | OT days | OT event days | model hail days |
|---|---|---|---|---|---|---|
| SAF | 2594 | 1439 | 1494 $\pm$ 47 | 196 | 181 | 172 $\pm$ 2 |
| KZN | 359 | 207 | 240 $\pm$ 13 | 96 | 80 | 66 $\pm$ 1 |
| HVD | 339 | 181 | 180 $\pm$ 17 | 74 | 58 | 60 $\pm$ 2 |
| GAU | 71 | 40 | 42 $\pm$ 3 | 32 | 23 | 26 $\pm$ 2 |

The result is satisfactory for the South Africa and Highveld regions, whereas for the KwaZulu-Natal region, events concentrate on slightly fewer days. Given the absence of multiple-event treatment in previous model versions, this new event approach represent an improvement for the estimation of damage on individual days.

In the Highveld region, there were 74 days per year with OT detections, 58 days with OT events, and 60 $\pm$ 2 events in the model event set, whereas in the Gauteng region, these numbers were 32 and 23, respectively (Table 3).

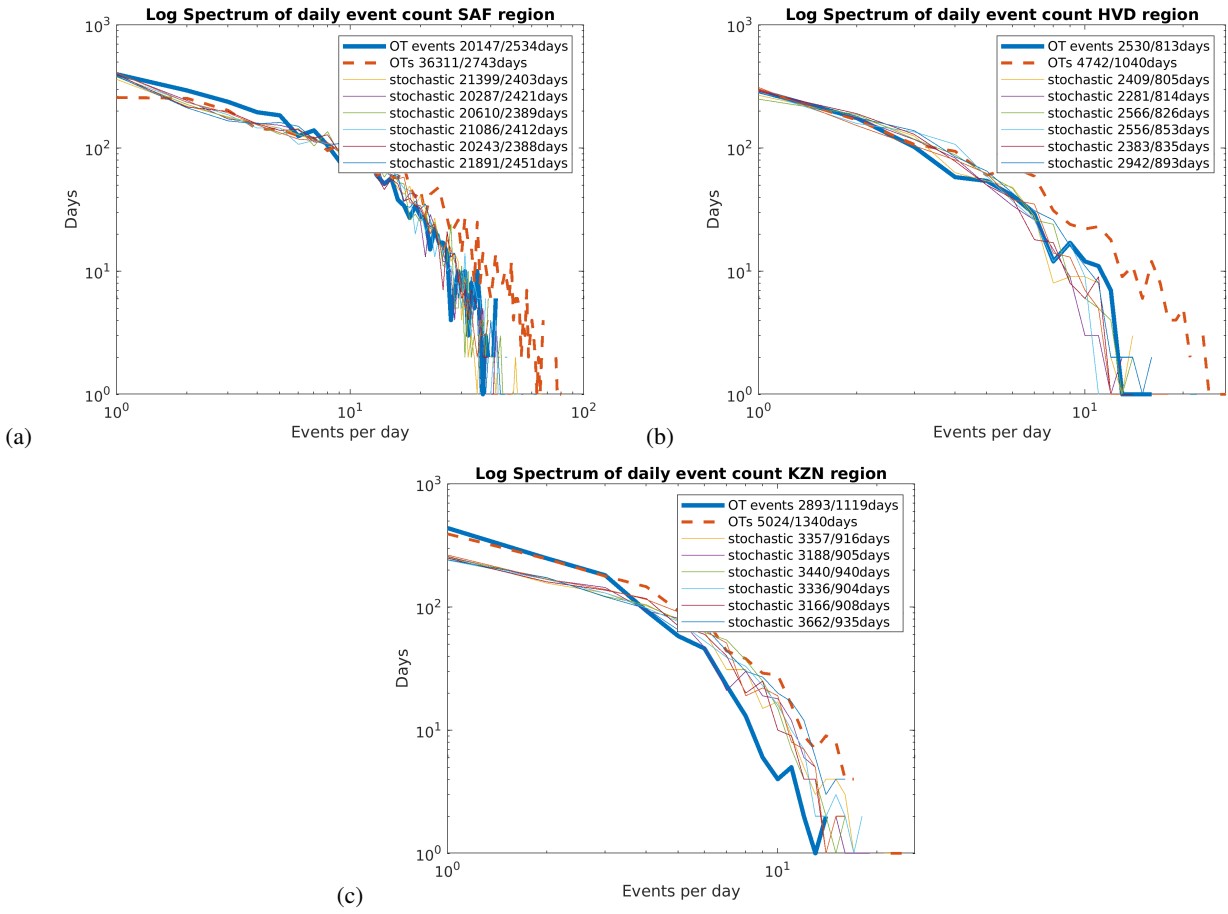

**Figure 16.** As Fig. 15, but comparing subsets of the stochastic event set to the historic OT event set (as well as the set containing events lasting only one time step) for each region. (a) South Africa, (b) Highveld, (c) KwaZulu-Natal. Numbers indicate the total count and total number of days on which these occurred for each set.

In contrast, Smith et al. (1998) found 69 hail days per year from hail reporting from a network of voluntary observers by mail, for a portion of the Gauteng (2 800 km$^2$). The observers reported three size categories: 3–10, 11–30, and >31 mm; information about the number of days for the different classes, however, are missing. Assuming an exponential hail size distribution, one can assume that the given number of hail days is dominated by events with small hail sizes. These are underrepresented in our model as they are not relevant for hail damage. This hypothesis is supported by the fact that Smith et al. (1998) reported about severe hail (>3 cm in diameter) on an average of only 3.3 days per year.

When considering only hail of this size and assuming the hail severity indicator of the stochastic event set would correspond to an actual diameter,the stochastic event set has 14.5 hail days per year in the Gauteng region. Clearly, the frequency of severe events may have changed since the time of Smith's observations, through natural variability or climatic change, but the effect is likely small compared to the uncertainty of both past and present estimates.

## 4 Modeled event footprints

### 4.1 Importance sampling

When applying the stochastic hail risk model to an insurer's portfolio, going through millions of events for – potentially – millions of assets is a time consuming process. While the complete event set is optimal for describing the hail hazard, an intermediate step, called importance sampling, is introduced to make risk calculations more efficient, reducing the event count by a factor of approximately ten. However, the most important events in terms of damage potential are over-represented to allow for adequate statistics (notably, computing damage at higher return periods reliably).

The newly introduced explicit modeling of the date requires that all events occurring on the same day need to be considered together. Consequently, daily aggregated damage potential (here: ellipse area × hail severity indicator) is the relevant quantity to rank event days by overall aggregated severity. The class thresholds correspond to the 50th, 80th, 95th, and 99th percentiles of aggregated severity, splitting the event set into 5 classes, of which 2.5, 7.5, 5, 10, and 100 % were retained. To compensate, the retained events are attributed a higher frequency (default is once per event set period, e.g., 25 000 years), so the total damage potential is conserved. These frequency weights are hence the inverse of the retained fraction per class, i.e., 40, 13.3, 20, 10, 1 for classes 1 to 5.

The large differences in event frequency across South Africa mean that local statistics on, for example, hail damage or probable maximum loss for a 200-year return period, can rely on a much bigger sample in the hail hot spots of the country compared to the less hail-prone regions in the west. It is hence important to retain a minimum number of events in those low-hail regions, in practice at least one per 250-year batch (if one is present in the first place).

Figure 17 illustrates this point: Fig. 17a shows the distribution of events retained in the importance sampling, some areas in the west and far north have less than 50 events in a $0.5° × 0.3°$ box ($\approx 30\,\text{km}$ in extent). As to be expected, when frequency weights are applied (Fig. 17b and c), the distribution corresponds very well to that of the full event set (Fig. 8).

### 4.2 Footprint generation

Finally, the areas affected by hail with corresponding hail sizes need to be determined for the stochastic events, forming a hail "footprint." Given the arbitrary paths of thunderstorms observed for the historic events, we chose to achieve this by a randomized process of allocating ellipsoid hail streaks within the event area. These streaks are aggregated across all events of a day to form the daily hail footprint, which is applied to portfolios in the further stages of the risk model. The footprint catalog gives local information on hail occurrence and severity indicator for each day on a $2 × 2\,\text{km}^2$ grid covering continental South Africa.

The footprint generation algorithm attempts to mimic observed patterns of hailfall in an empirical way. A first streak is located in the center of the event, and its length, orientation and severity indicator match those attributed to the event. Streak widths were chosen to approximate hail streaks in ground- and radar-based studies (Stout et al., 1960; Changnon Jr, 1977; Kleinschroth, 1999; Schmidberger, 2018; Fluck et al., 2021), without strictly following observed distributions. An exponential distribution is assumed, with a mean width of 6 km, and a maximum of 20 km. Hail severity decreases towards the streak's

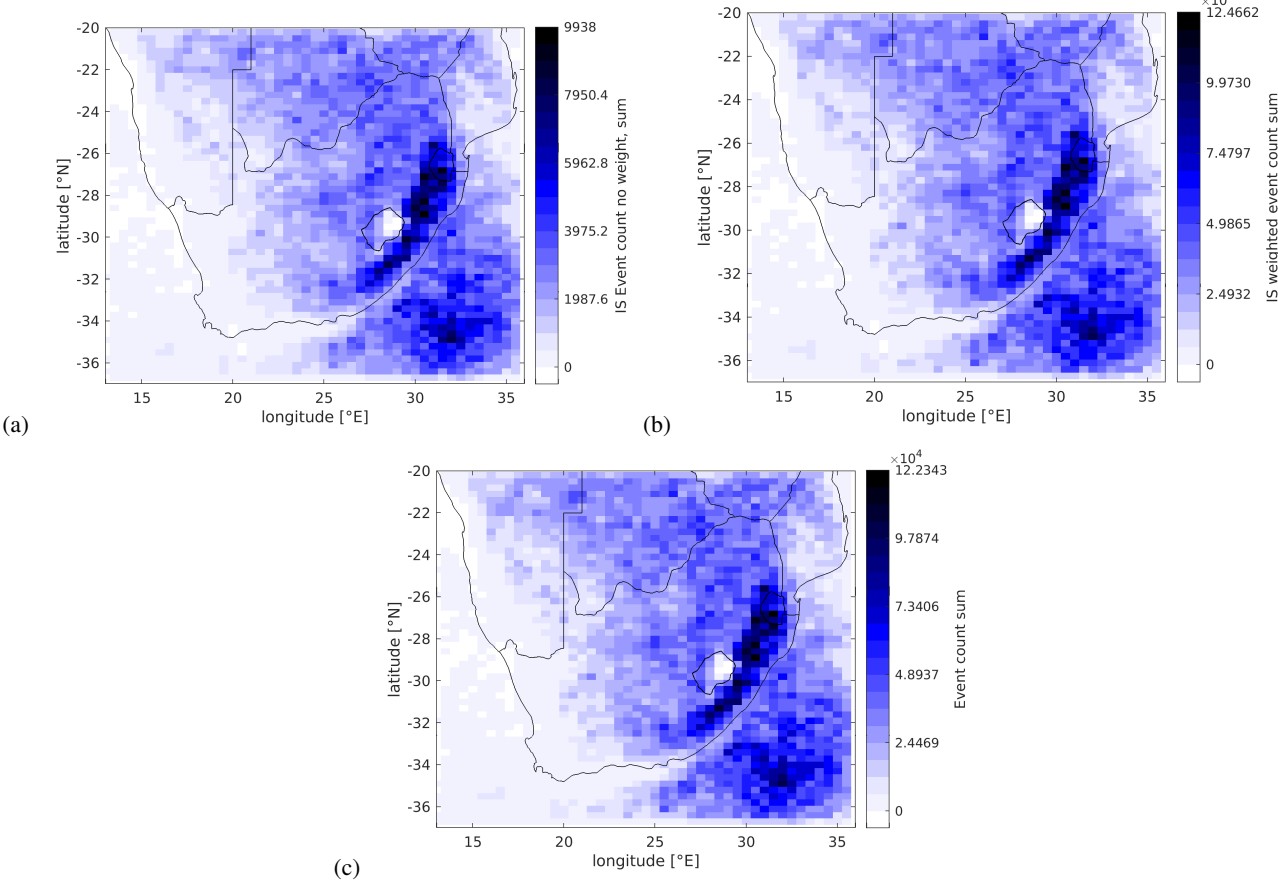

(a)

(b)

(c)

**Figure 17.** (a) Number of events retained by the importance sampling; (b) As (a), but weighed by frequency; (c) event frequency before importance sampling, showing very little difference.

edges in a parabolic way, as proposed by Schmidberger (2018). This is deemed acceptable since actual hail patterns on the ground are largely uncertain across the world. In an iterative process, further streaks are added until the combined streak area covers the prescribed fraction of the event area. They are located randomly within the event area, and the possible event length 460 decreases with each new streak. Streak orientation is varied by +/- 10° around the event orientation to account for both the uncertainty in the tracking of OTs and new cell formations preferably at the downshear flank.

Accumulating events over a time equivalent of 2 500 years, Fig. 18a and b show the number of events hitting each cell for a 10%-sample of the importance sampled event set, for the least and most severe class (1 and 5, respectively). Clearly, the footprint frequency over the Highveld and KwaZulu-Natal increases from class 1 to 5, whereas it decreases in the western half 465 of the country. Consequently, hailstorms are relatively more often severe over the Highveld than other parts of the country, with important consequences on the financial risks associated with the peril. Figure 18c shows the accumulated, frequency-weighed annual sum of hail occurrences in the model. We note that the local frequency-weighed hail count per year is about 2 in the

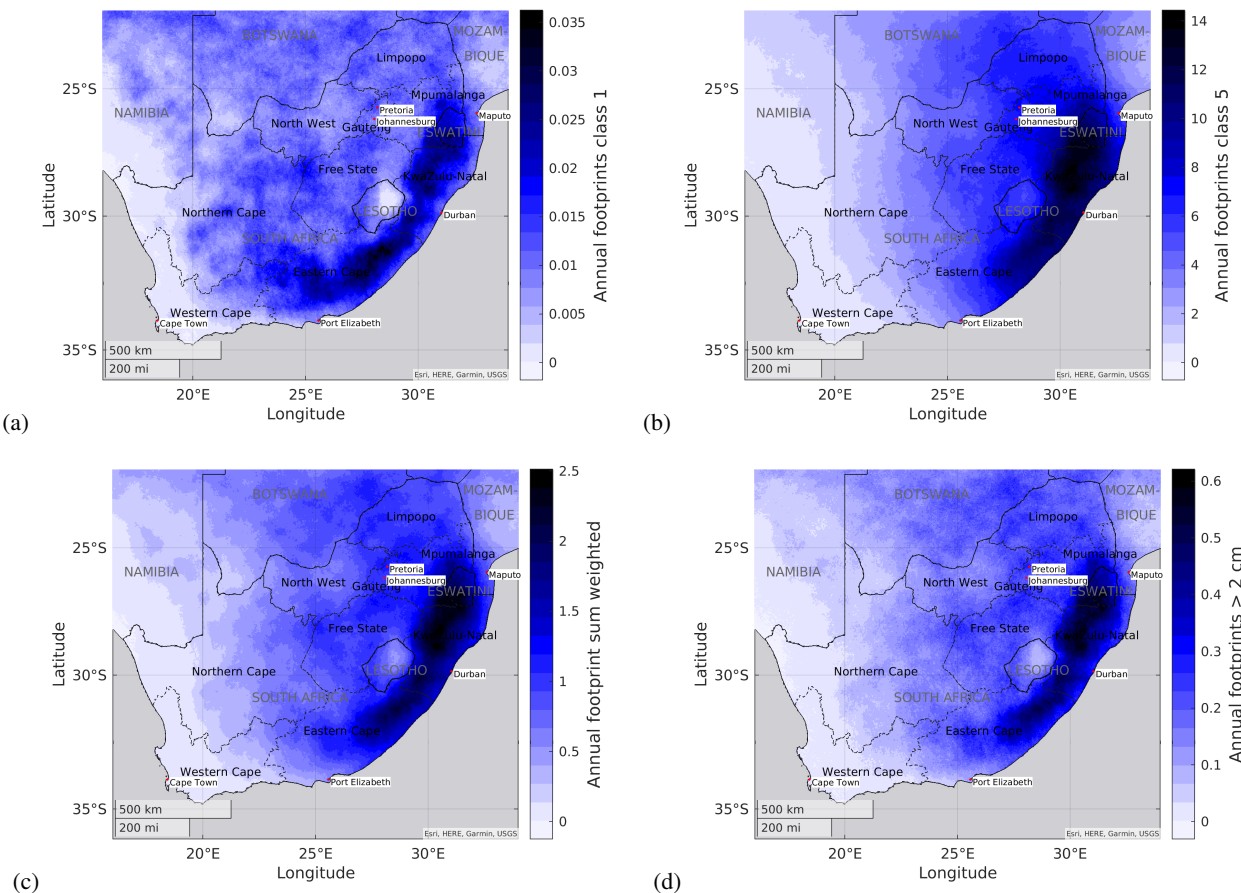

**Figure 18.** Number of events hitting a grid cell per year (a)–(b): for classes 1 and 5, respectively, (c): total, frequency-weighed, (d) as (c), but for hail diameter >2 cm.

KwaZulu-Natal maximum and around 1 in the Highveld and Gauteng regions, roughly in line with 0.81 normalized hail days per year in Held (1974). For a maximum hail severity indicator greater than 2 cm, the overall number of events decreases by
470 about 75%, while the overall spatial distribution remains almost the same (Fig. 18d).

Based on the local hail count and event set length, return periods, i. e. inverse frequencies, can be estimated for given hail severity thresholds. Hail severity for a fixed return period was calculated by linear interpolation between such thresholds fixed at constant intervals. Summarizing the information contained in the event set, and assuming hail severity estimates were corresponding to actual hail sizes, Fig. 19 shows the maximum hail severity that can be expected once per decade at a given
location: In the most affected parts in the East, near Newcastle and Ladysmith, hailstones of around 4 cm were to be expected, followed by 3.5 cm for Pietermaritzburg, Mthatha or Nelspruit, 3.1 cm in Durban, 2.8 cm in the major cities of the Gauteng, but less than 2 cm in the western third of the country. Despite the uncertainty regarding the exact hail size – OT relation, this

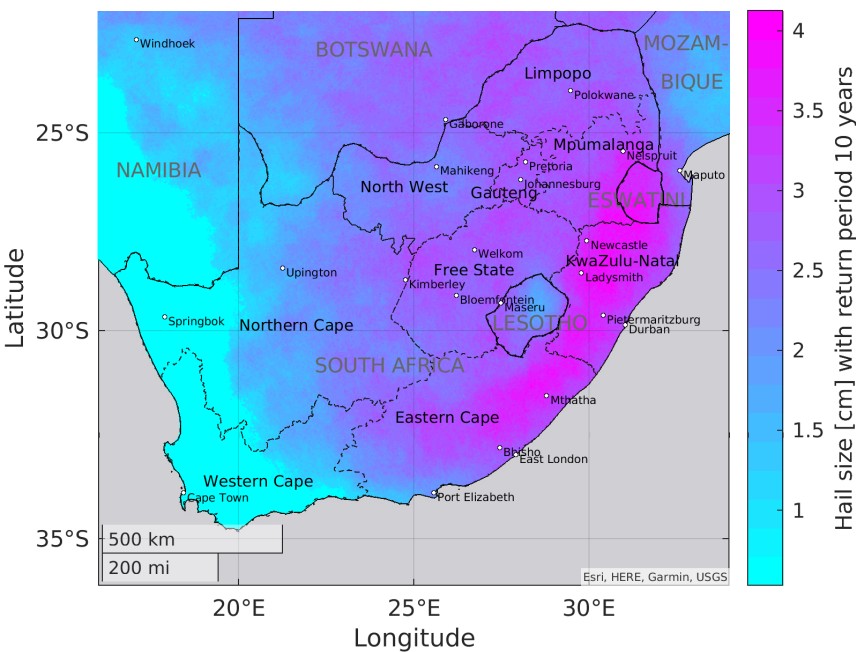

**Figure 19.** Model-estimated maximum hail size occurring once in 10 years for grid cells across South Africa, computed as the lowest hail diameter class with a return period.

information has clear implications on the need for mitigation measures to reduce hail risk, such as roof cover robustness or covered parking of vehicles.

## 5   Conclusions

In our paper, we have presented a method to estimate hail frequency for South Africa, a country frequently affected by large hail, and to generate footprints over a long-term period of 25 000 years, which can be used by the insurance industry to quantify hail risk. Despite of the high exposure to hail, no reliable estimates of hail frequency are available for South Africa to date. The lack of available damage reports or accurate hail observations, for example, by human observers or by hailpad observations, limits the accuracy of hail hazard descriptions (compared to, e.g., Europe, Púčik et al., 2019). The uneven distribution of population and wealth in the country complicates this matter further. For this reason, satellite-based observations of SCS/hailstorms have been used to describe the spatial distribution and nature of intense convection in the country. By stochastic modeling, hail hazard was derived from a large sample of events, which can be used to quantify hail risk for a given portfolio of insured assets. As was shown in the manuscript, the stochastically generated event set matches very well with the historic event set over a 14-year period.

The combination of improved OT detection and advanced spatio-temporal clustering allows the determination of hail hazard zones much more precisely compared to the method used for the Willis European Hail Model (Punge et al., 2014). Regarding storm properties, exponential distributions have been replaced by GEV distributions in most instances, yielding a better fit to observations.

More importantly, the model is now capable of producing realistic spatio-temporal distribution of events, handling multiple events per day as well as their spatial spread and multi-year variability. By explicitly including the date as an event property, it has become possible to represent outbreaks with multiple events on a single day in a realistic way (see Fig. 16). Hence the increased financial risk of clusters with repeating severe storms (e.g., 11 and 28 Nov. 2013 in Gauteng, Dyson et al. (2020); or in Germany on 27/28 July 2013, Kunz et al. (2018); Munich RE (2015)) is accounted for.

In addition, the footprint generation algorithm has been revised to predict hail only in a fraction of the event area, mimicking observed storms. This will assist the calibration of the exposure and vulnerability functions of the risk model and yield more accurate loss estimates. Finally, another addition compared to previous model versions is the time of day, which will allow to reflect daily changes in exposure, for example of parked cars, in the risk model.

Of course, with the OT-based approach, some scattered, short lasting hail episodes forming smaller hailstones may be missed. These events, however, are generally unimportant for the hail threat to insurance businesses, which are mostly concerned about major loss events. Another limitation of the OT approach is the difficulty to distinguish hail-producing from non-hail producing storms or to identify the fraction of an individual storm's track in which it produces hail. While storm environments from reanalysis (e.g., Bedka et al., 2018b; Punge et al., 2017) can help with the first task, they can hardly address the second.

Nonetheless, while the absolute number of hail days per year strongly depends on the minimum size of hail considered, findings are comparable to local hail reporting-based studies, particularly for the Gauteng region, for which some studies of observed hail frequency are available (Carte, 1977; Admirat et al., 1985; Smith et al., 1998). However, there is disagreement with previous studies over the presence of a hail frequency maximum in the Lesotho mountains, absent in our study. Also, there is uncertainty on the occurrence of hail off-shore, albeit not in the focus of this study. Future studies will have to address whether the differences between our study and previous work in the spatial distribution of hail are artifacts of imperfect methodology and assumption or actually represent reality.

## Appendix A: OT intensity

To demonstrate the relationship between IR-based storm intensity metrics and an estimate of hail size, we compared GOES-12 and GOES-13 data with 95th percentile Maximum Expected Size of Hail (MESH) from the U.S. NEXRAD GridRad dataset at hourly intervals from 2007-2017 (Murillo and Homeyer, 2019). MESH cell objects exceeding 2 pixels in area (10 km$^2$) and spaced by at least 28 km are derived using watershed segmentation applied to the hourly 10 mm+ MESH95 climatology (Murillo and Homeyer, 2019) using the open-source Tracking and Object Based Analysis of Clouds (tobac v1.2; Heikenfeld et al., 2019) Python package. Further, following Murillo et al. (2021), we have applied Linear Discriminant Analysis (LDA) using their coefficients to combine reanalysis precipitable water and 0-6-km shear to filter out likely false alarms.

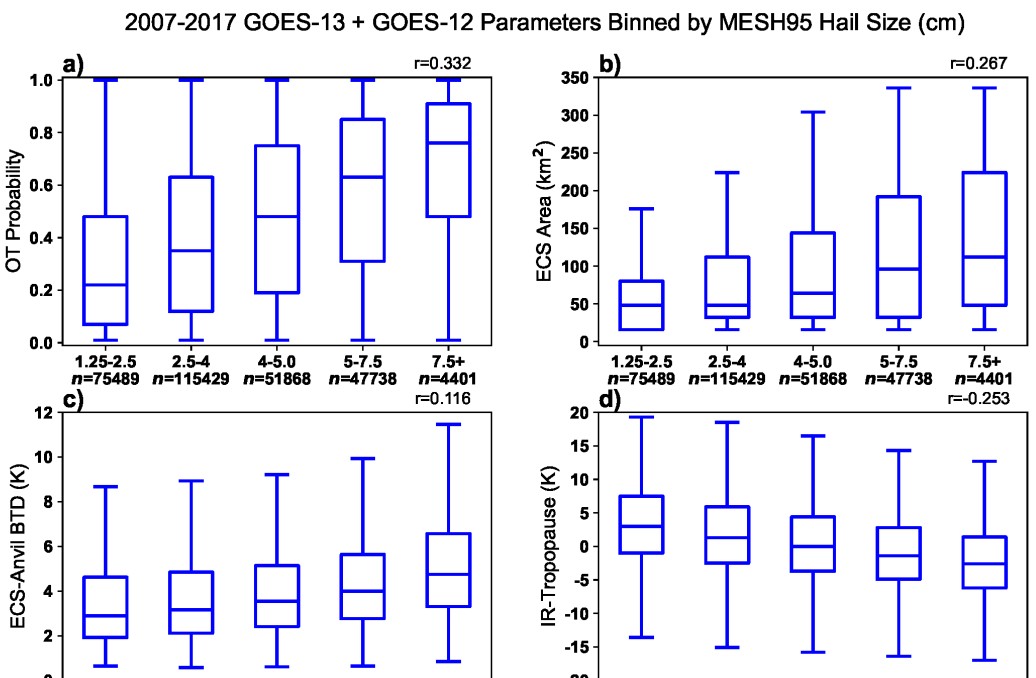

**Figure A1.** GOES-12 and GOES-13 IR-based parameters binned by MESH95 hail sizes. MESH95 exceeding 4 cm is considered a potentially severe storm. Counts in each bin are shown below the $x$-axis, and the Pearson correlation coefficient is shown in the upper right of each panel.

Relationships between GOES-12/13 OT probability, area of the embedded cold spot (ECS) matched with the MESH95 cell,

ECS-Anvil and ECS-tropopause temperature difference, and MESH95 are shown in Fig. A1. Though there is considerable overlap in GOES parameters between the various MESH95 bins, it can be seen that all for parameters representing updraft intensity and area are positively correlated with MESH95. The correlation between GOES-16 data and MESH95 during spring and summer 2017 (not shown) is even greater than GOES-13, due to higher spatial resolution of GOES-16. OT probability is better correlated with MESH95 suggesting that the prominence of an ECS relative to the background anvil combined with its

intensity and area are all contributing to higher OT probability. Therefore, IR-anvil BT difference is a suitable parameter, independent of any reliance on a numerical model, for purposes of modeling the expected hail severity at the ground. Though these results suggest a quasi-linear relationship between MESH and satellite-derived updraft intensity proxies, the true relationship between such proxies and hail size encountered on the ground is unknown, primarily due to known uncertainties with hail size reporting. While direct matching of SEVIRI with MESH cells over South Africa is not possible, the $\approx$3 km nadir pixel

resolution (halfway between GOES-13 and GOES-16) and lower view zenith angles over the South African domain (MSG is positioned at 0°E) are expected to result in similarly robust correlations with hail size diameter.

**Table A1.** Detection counts and fractions of geostationary derived embedded cold spot and OT detections matched within 28 km$^2$ and 15 minutes of hail observations.

| **GOES-13** | count | count with ECS | fraction with ECS | count with OT | fraction with OT |
|---|---|---|---|---|---|
| MESH95 $\geq$ 4 cm | 148 326 | 104 007 | 0.70 | 67 668 | 0.46 |
| MESH95 $\geq$ 2.5 cm | 384 444 | 219 436 | 0.56 | 109 839 | 0.29 |
| SPC hail $\geq$ 2.5 cm | 121 505 | 82 443 | 0.68 | 46 211 | 0.38 |
| MWR P_hail $\geq$ 50 | 4 225 | 3 165 | 0.75 | 2 440 | 0.58 |
| **GOES-16** | count | count with ECS | fraction with ECS | count with OT | fraction with OT |
| MESH95 $\geq$ 4 cm | 10 452 | 8 677 | 0.83 | 5 814 | 0.56 |
| MESH95 $\geq$ 2.5 cm | 28 811 | 19 367 | 0.67 | 10 619 | 0.37 |
| **MSG** | count | count with ECS | fraction with ECS | count with OT | fraction with OT |
| MWR P_hail $\geq$ 50 | 732 | 536 | 0.73 | 458 | 0.63 |

Table A1 further compares the frequency of GOES-13, GOES-16, and MSG embedded cold spot (ECS) detections, e.g., areas that appear distinctly colder than the surrounding anvil and are considered to be OT candidates, and OT detections (OT probability $\geq$0.5) matching various hail detections from radar cells, ground spotter reported hail size (SPC), and MWR hail detections. Requiring OT probability $\geq$0.5 to refine severe hail detections to those we are most confident in, we lose 54% (44%) of the severe hail-producing storms exceeding 4 cm MESH95 maxima for GOES-13 (GOES-16). In other words, many severe hailstorms can look quite "boring" from a satellite infrared perspective, but the boring ones are hard to differentiate from false OT detections in anvils (i.e., detections in cold outflow near to real OTs). Uncertainty between report time or the time a radar scanned a storm vs the time of OT detections may also influence our results. For example, an OT may have been prominent several minutes before the time of a hail detection, but we only have a single GOES snapshot to match. By relaxing the matching criterion to ECS detections, we lose only 30% (17%) of likely severe hail producing cells for GOES-13 (GOES-16).

The frequency of geostationary updraft detections that are co-located with microwave hail detections is comparable to MESH95 and ground spotter severe hail reports despite added uncertainty due to parallax shifts in the storm positions in microwave data, especially those close to the limb of the overpass. Enabled by the global coverage of microwave satellites, Table A1 shows that 2005–2018 MSG SEVIRI ECS and OT detections over South Africa match with likely severe MW hail detections with frequencies similar to GOES-13 over the US. Although the total number of matches is relatively low over South Africa, this suggests that MSG IR-based updraft detections agree with independent hail detections; thus, supporting the use of MSG SEVIRI to detect hail cores over South Africa.

Angular dependence of GOES IR-anvil BTD difference data can be addressed through normalization based on the footprint area of GOES relative to the nadir footprint area (16 km$^2$ for GOES-13) Without removing view angle dependence, the prominence of an OT observed at low VZA (e.g., <40° in the Southeastern US) would be greater than had this same OT occurred at high VZA (e.g., >50° in the Northern Plains) due to differences in the effective pixel resolution.

The formulae to derive the $x$ and $y$ component of pixel resolution for GOES-12 and GOES-13 are:

$$\lambda_d = |-75 - \lambda_c|\frac{\pi}{180}$$

$$\phi_d = |\phi_c|\frac{\pi}{180}$$

$$\Delta_r = \frac{4}{\cos\theta_z}\frac{\pi}{180}$$

$$\Delta x = 4 + 0.263\frac{\Delta_r \sin\lambda_d}{\cos\lambda_d}$$

$$\Delta y = 4 + 0.4849\frac{\Delta_r \sin\phi_d}{\cos\phi_d}$$

$$\text{nf} = \frac{16}{(\Delta x^2 + \Delta y^2)^{0.5}},$$

where $\lambda_c$ and $\phi_c$ are longitude and latitude of pixel center, $\theta_z$ the viewing zenith angle, and nf the norm factor.

This normalization results in improved correlations between ECS-Anvil BTD and MESH95 for GOES-12/13 and GOES-16 shown in the box and whisker plots (Fig. A1).

## Appendix B: Climatology of satellite-derived hail estimates

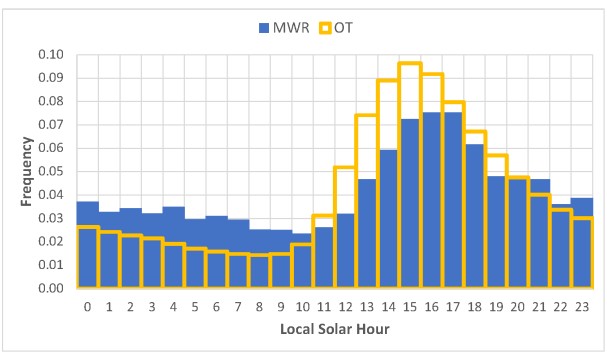

**Figure B1.** Variation of OT activity throughout the day, estimated with the OT method (yellow) and passive microwave detection (blue), normalized by the total number of detections.

The microwave hail detection algorithm (Bang and Cecil, 2019) and the OT-based detection method (Khlopenkov et al., 2021) have independently been designed to represent hail occurrence. A comparison of the spatial and temporal variability of the detected occurrences can thus be used to identify times and locations where the two methods disagree, indicating potential weaknesses or imbalances in one of the approaches.

The daily cycle of OT activity is more pronounced compared to that for the microwave detection (Fig. B1), and its maximum sets in almost an hour earlier. Overall the agreement between the two satellite climatologies, however, is very good, but the OT algorithm may be slightly too sensitive for weaker convection.

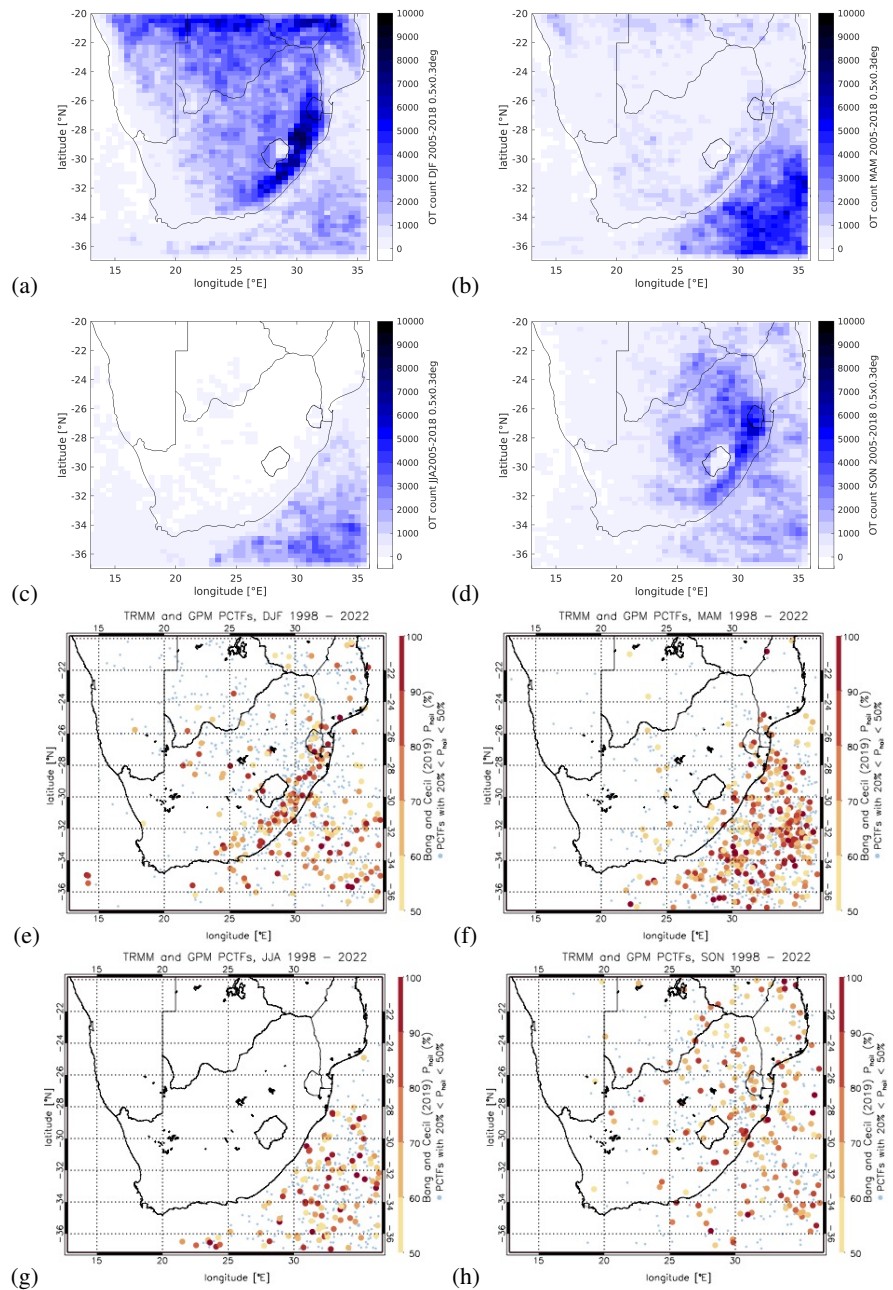

**Figure B2.** Seasonal hail activity estimates based on overshooting top activity (2004-2018, a-d) and passive microwave hail retrievals (1998-2018, e-h) for austral summer (DJF; a, e), autumn (MAM; b, f), winter (JJA; c, g), and spring (SON; d, h).

To that end, Fig. B2 shows the seasonal sums of OT (a-d) and microwave (e-h) activity over the respective observational period. Of all seasons, most OTs are detected over land in Austral summer (DJF; Fig. B2a, e), and the spatial distribution is

similar to that of climatology. In autumn (MAM; Fig. B2b, f), and to a lesser degree in winter (JJA; Fig. B2c, g), both methods indicate persisting hail activity over the eastern oceans and along the coast. Summertime (JJA; Fig. B2c, g) presents the highest activity with both methods, and a clear focus along the Eastern slopes of the Grand Escarpment and some activity over the south-eastern ocean. We note a difference in the North, where the OT algorithm appears to detect intense but, due to high temperatures, non-hail producing convection (Fig. B2c). Both methods indicate widespread hail activity in spring (SON; Fig. B2d, h) over the eastern half of the continent and adjacent oceans, with a somewhat more pronounced concentration around Botswana and the Drakensberge with the OT method. This method could thus be somewhat too sensitive to the frequent but weak convective activity in springtime also found in other regions of the world (e.g. Europe, Punge et al., 2017).

*Author contributions.* HJP, MK, SDB and KMB conceived the stochastic model and this study. KMB designed the OT detection algorithm, provided OT data, and suggested figures. MK supervised the model design and development. SDB developed the microwave hail detection algorithm and provided the corresponding data. KFI evaluated the OT algorithm against radar data. HJP analyzed the data and prepared the manuscript, with all co-authors providing critical feedback and helping shape the analysis and manuscript.

*Competing interests.* The authors declare they have no competing interests.

*Acknowledgements.* The NASA Applied Sciences Disasters program award 18-DISASTER18-0008 supported the collaboration between NASA and KIT. Heinz Jürgen Punge is supported financially by Willis Towers Watson (wtw) through the Willis Research Network. We thank ECMWF for providing ERA-5 reanalysis data and wtw for providing hail damage claims for sample events. We thank the Data Center at the University of Wisconsin-Madison, Space Science and Engineering Center for providing the archived MSG SEVIRI imagery used in this paper. We thank the European Severe Storms Laboratory (ESSL) and the Australian Bureau of Meteorology for providing hail report data. Credit is given to gadm.org und geonames.org for providing geodata used to produce the maps.

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
