# Peer review of "Characteristics of hail hazard in South Africa based on satellite detection of convective storms"

_Natural Hazards and Earth System Sciences, 2021_

## Referee Comment (RC1)

**Characteristics of hail hazard in South Africa based on satellite detection of convective storms**

Heinz Jurgen Punge[1], Kristopher M. Bedka[2], Michael Kunz[1], Sarah D. Bang[3], and Kyle F. Itterly[4]

**Summary.**

This paper reviews and analyses the hail climatology for South Africa and regions within South Africa using satellite detection of convective storms. The paper investigates 14 years of geostationary satellite observations of convective storms and generated a spatiotemporal multivariate stochastic model representing 25000 years. The historic, stochastic and observed insurance exposure and vulnerability data are analysed to identify the expected hail damage for return periods of 200 years.

**General Comments**

Scientifically the authors have done an excellent job considering the literature review, data collection and modelling that have gone into the research described in this article. One slight drawback is that in the paper itself, the authors attempt to address all the whole multiple complex modelling processes in a relatively short and succinct manner. At times, unfortunately, the description of the process followed does not do the modelling process justice and made it difficult to follow what was done. Some examples will be given below.

The format of the paper also made reading the paper very difficult for me. At times, the figures and tables mentioned in the text were not next to or near the text referencing them. This caused a lot of scrolling up and down in pdf (and eventually I just printed out the text in frustration). As times, figures were even placed to appear to be part of a previous section e.g. Fig. 14 seems to form part of the end of Section 3.3 but is part of Section 3.4 that starts underneath Fig. 14. The format used by the authors may be due to format instructions from the journal. If not, please reconsider the placements of figures and tables to be as close as possible to the relevant text to improve the reading flow of the paper.

**Specific comments**

- 25. The reference to Grieser and Hill (2019). Did Grieser and Hill focus on hailpad derived metrics for South Africa, another country or just in general?

- 35. "*hailstorm formation is often related to local and meso-scale processes related to, for example*" Perhaps do not use the word related twice in one sentence.
- Figure 1 "*a title: SRTM*" acronym is not defined. Please check whole text for acronyms.
- 95. "*Based on past experience, only OTs detected with a probability >50% and with a surrounding anvil cloud (green and yellow colors in Fig. 1a; the IR anvil detection index, a rating based on an anvil detection model accounting for viewing situations, greater than 10; see also Scarino et al., 2020) are used in this work.*" Make the sentence in brackets a sentence on its own or add to figure description.
- 110 - 115. "*A uniformly distributed random number between -0.5 and 0.5 115 was added to each reported hail diameter to compensate*" I assume the -0.5 and 0.5 is also in mm?
- 155 "*As 9.5% of the OTs occur at a melting level of less than 2 400 m, but ony 3.5% of the microwave hail detections and and 2.5% of the claims, a lower threshold of 2 400 m was introduced for this parameter.*" . Spelling. Sentence seems incomplete.
- 160. "*The latter feature is due to the minimum freezing level condition and remains to be confirmed by independent observation.*" Independent observation from whom?
- 190: "*complemented with hail size information from reporting.*" Reporting? You mean the insurance reporting?
- 200. "*Following Punge et al. (2014), both annual and daily cycles are modeled with Gaussian distributions. For the day of year, domains of 3° × 5° are considered, and depending on the…*" Why Gaussian distribution? What is the statistical justification for it? Not sure how these grid definitions relates to the previously defined rectangular grids of 0.3 x 0.5 mentioned on page 10.
- 205: "*Days are drawn from the boxes distribution for the…*" It is not clear to what the boxes distributions refer to.
- 205: "*…Finally, the day is retained only for N/9 events at random. This procedure has been found empirically to approximate the observed space-time distribution of days in a satisfactory manner.* " Is this procedure self-developed or taken from somewhere? Why $n^{1/3}$ and N/9 - those specific values? What is the proof of empirically proof behind it?
- 210*: "from a region of 10° × 6° around..*" Why the double grid size? Is this to also represent the 8 neighbouring grid cells?
- Paragraphs 205 and 210 can be extended to make the spatial construction more clear. In the current format is it difficult to follow, and relate back to standard spatial weight matrixes using the queen criterion.
- 215: "*Also note the secondary maximum in fall (around days 100–150, i.e. April and May) during nighttime, represented in the model. * " Does this represent a local maximum? How do you see from the graphs it is in the night?

- *215: " It is shifted towards fall over the Southern Ocean." ???* Are you modelling that far away from the shores of the country as well? And will it have any landfall impact?

- 220: "*Time, slightly earlier than Smith et al. (1998, 5–6 pm) but consistent with Olivier (1990) (Fig. 11b). The daily cycle is most pronounced…*" What is the possibility of there being a shift in these times from the 1990's to now? In that case, would the results be comparable?

- Figure 10: I assume the day of the year for 1 to 365 represents 1 Jan to 31 Dec. Perhaps add that to the title to indirectly show the difference in expected hail occurrences for northern and southern hemisphere?

- Figure 11: I'm struggling with what number of days each bar represents. It seems the number 50 falls on the 4th bar?? This will only work if each bar represents 12.5 days?

- 225: "*The distributions are well approximated by the GEVs*". GEV is an extreme distribution that requires a "limit" (e.g. peaks-over-threshold or block-maxima) in the data over which you are modelling events? What was that limit and how was it obtained?

- 230: " *to give unrealistic large values, which is why length and width have been truncated at 1.5 times the largest observed values,*" Why the specific value of 1.5 times the largest observed value?

- 230: "*In addition, the fraction f of the event area (the area of the ellipse spanned by major and minor axis of lengths l and w,*" Remember to write the last l and w in italics.

- 235: "*Table 1 lists the distributions and parameters for these event properties.*" Which method was used to estimate the parameters of the distributions?

- 240: "*We find that most frequently, events have an orientation of around 100°, i.e., propagate eastward to southeastward (Fig. 11f).*" This is for the whole country. But it may be misleading as this is not the typical orientation for a high hail fall region like Gauteng where storms normally originate in Johannesburg and move north-easterly to Pretoria. As seen from the discussion in the next paragraph.

- 280: *"sets of random numbers for each property from a uniform distribution and determine ranks. Then, for each property, we draw values from the actual distribution, sort them, and attribute to events using the pre-determined ranks. "* How? Does this again refer to a previously defined or described methodology?

- 285: *"could be expected, smaller regions show relatively higher variability, but there is strong correlation between the two. This"* Which 2? Smaller regions and the country as a whole?

- 295: This section describes the South African domain in terms of latitude and longitude degrees, subregions etc. Should this description not be done earlier in the paper to set the scene – perhaps where Figure 9 is defined?

- 305: "*50 hail days per year), while in an 305 equivalent sample of subsets from the stochastic event set, the event count ranges from 1 883 to 2 162 events on 671 to 703 days*" Perhaps add the equivalent hail events and days per year for comparison.
- 305-310: "*In the Highveld region, there were 74 days per year…*" These numbers are averages per year. The averages per year for the years defined by the authors and the years defined by Smith et al 1998 are different and it should be considered that several climate changes occurred in the years in between. This includes periods of severe drought in the several regions in the country especially between 2010 -2020
- 310: Can the numbers given in this paragraph be added in a table for easier reference?
- 310: "*However, severe hail (>31 mm in diameter*" Why 31 mm and not a round number like 30mm?

---

## Referee Comment (RC2)

**Review for NHESS-2021-342**
Characteristics of hail hazard in South Africa based on satellite detection of convective storms
Recommendation: Reject

The authors have assembled a novel methodology for estimating hailfall in South Africa, a region with frequent hailfall but sparse observations. Such a hailfall climatology in South Africa is clearly needed, particularly with the possibility of shifting or increasing hailfall frequency with a changing climate. From this hail event climatology, they have additionally assembled a statistical model that includes estimations of hail size, hail swath shape and orientation, and frequency of occurrence over a much longer period. The creation of all these products is ambitious, but I feel the authors have overreached what is scientifically defensible though the necessary chain of assumptions. I don't doubt that there is a strong operational need for extended hail climatology products like these in this region, but if choosing to publish the work the assumptions must be reasonably defended. In sum, I recommend rejection in the paper's current form, but would welcome reviewing a resubmission on a narrow, better-grounded portion of the work.

Major comments/fatal flaws:
The work performed here was obviously extensive, and I appreciate the effort to scientifically ground an operational product. I've broken down my view of the chain of reasoning presented in the paper, along with my opinion of how well each step is grounded in the article.
1. *Hail occurrence can be estimated using the Khlopenkov et al. (2021) OT detections in GOES data over CONUS.* This step is well-grounded, given Khlopenkov et al. and Cooney et al. (2021) results discussed in the introduction, although a quick sentence or two discussing the skill level of that algorithm with the severe hail report database used in those studies would be useful to add.
2. *The Khlopenkov et al. OT algorithm can be applied to MSG SEVIRI data over S. Africa with similar success as GOES data over CONUS, with the additional environmental filtering applied.* This claim is generally supported by the results in the paper (c.f., Figs. 3 and 4), but needs a fuller explanation. The geographic hotspots are similar in Figs. 3 and 4, but is the frequency of potential hail occurrences reasonable? Comparison of OTs, GPM/TRMM detections, and radar-based detections over CONUS could confirm the relative change in frequency between OT and GPM/TRMM detections over S. Africa is reasonable. Comparisons should also be made to climatologies made over the region from other methods, such as those discussed in the introduction (Admirat et al. 1985; Prein and Holland 2018; Kunz et al. 2020; Dyson et al. 2020).
3. *The hail grouping methodology into events reasonably represents hail swaths from a single storm system.* While the description of the methodology (lines 176-177) is intuitive and simple, the results of the grouping methodology in Fig. 7 don't seem to follow that description. Why are there multiple events occurring at a single place and time? Once the methodology itself is cleaned up, a few example applications of this methodology in an area with radar data would show its value in establishing hail events and their duration and speed. Right now, the results of the methodology are only briefly compared in text to two other radar-based studies of severe convective storms (not limited to hailstorms) in the literature.
4. *The created hail event climatology shows reasonable distributions of hail event frequency by time of year and time of day.* No comparison of these distributions is made to the observational or GPM/TRMM datasets. While they are admittedly sparse, they should at least be able to confirm general seasonality. Comparisons should also be made to the other climatology datasets mentioned in point 2 above.
5. *The statistical method established in lines 202-213 can be used to produce similar hail event daily and seasonal hail event variations established by points 1-4 above (assuming points 1-4 are successful at representing actual hailfall).* The annual and daily

distributions produced by the model do appear similar – I'd prefer a difference plot instead of a side-by-side comparison, given the relatively large magnitudes involved. However, the description of the statistical method is not clear, and only one reference is sited. How common are methods like these? The steps involved in its description are very specific, making one wonder if the model is being over-fit to its underlying dataset. How similar is the methodology used here to Punze et al. (2014, unfortunately behind a paywall), what changes were made, and why?

6. *The statistical method in lines 225-238 can be used to produce similar hail event length, width, area, and orientation as the event climatology produced in point 3 above (again, assuming point 3 is valid).* These results do seem reasonable as presented in Fig. 11, but no point of comparison is provided. How well do other statistical methods perform? What is expected behavior?

7. *Hail size can be estimated using the OT climatology product produced in point 2 (I don't think the event climatology from point 4 is being used here, but text isn't clear).* This claim is (currently) indefensible.
   - Marion et al. (2019) suggested a relationship between OT *area*, not strength, with updraft width and hence potential tornadic intensity. That's a not insignificant difference. Hail size, particularly as one reaches larger hail sizes, is more related to updraft width than updraft strength (e.g., Nelson 1983, Foote 1984; Kumjian et al. 2021). I am concerned that by relating hail size to an updraft strength metric, an erroneous hail size distribution will be produced.
   - Khlopenkov et al. (2021) connected OT detection probability with hail *occurrence* and did not try to distinguish among hail sizes.
   - Figure 2 appears to represent original work from the authors (sentence is oddly phrased, making it seem like it is sourced from Murillo and Homeyer 2019). While I do appreciate the correlation shown, I am concerned the MESH95 dataset is being used, and not actual hail reports. Per Murillo and Homeyer, the MESH95 dataset has a significant large bias, with 40 mm being most skillful at determining 25 mm hail, and 64 mm being most skillful at determining 50 mm hail. That bias does not appear to be accounted for in Fig. 2. Further, while Murillo and Homeyer (2019) did not specifically examine the skill of tropospheric-OT temperature difference in differentiating among hail sizes, they did examine the distribution of minimum GOES IR Brightness and GOES OT Area (see their Figs. 6a, b, 8a, b), and did *not* find a strong relationship between those fields and observed hail size.
   - In my opinion, this claim cannot be supported given the current literature, and hail sizes should be removed from the database (or only provided to customers with a strong caution about their use, and not published in the literature).

Given these issues above, I cannot recommend the article for acceptance. I would be happy to review an article focusing on points 1-4 above, after addressing the issues I've described. A companion paper focusing on points 5-6, after points 1-4 are successfully established, would also be interesting. I cannot support an article including point 7 at the current time.

Other major comments:
- Section 3 would be much easier to follow if the observations were presented first (i.e., annual, daily cycles) with comparison to other parts of the world, and only then the development of the statistical method discussed. In my opinion the main scientific point of the article should be the new climatology and establishing those results should be given more weight than the statistical developments (which rely on the new climatology being accurate). I would prefer statistic results being shifted to a new paper or at the least a new section.
- Event grouping:

- Lines 173-174: This definition seems straightforward and reasonable. However, Fig. 7 doesn't use that definition, instead identifying multiple events that exist within a larger event. For the climatology to be useful (e.g., for Fig. 8 to convey valuable information),
explicitly restricting each OT to a single event is necessary.
- Line 197-198: Why not an equirectangular grid? Why this resolution? Lines 197-198: Shouldn't this just be calculated as the number of events total in each box, divided by the total number of years? Introducing the number of overshooting tops seems unnecessarily complicated. Are the authors retaining their criteria that OTs must last longer than a single time step (lines 181-182)?
- Line 200: Wouldn't using number of events instead of OTs better address this issue? To further constrain the issue of multiple events in the same location, events could be capped at one per day.
- Temporal distributions
  - Lines 203- 209: What made the authors think a Gaussian distribution was best? Why was this large a grid needed- can the authors discuss additional sizes that were tried?
  - Lines 205- 209: Can the authors point to other studies or methods even in another field, that use a similar technique? These values appear very tuned to these specific observations/year groups and hence potentially not broadly applicable. Finally, the explanation itself is confusing. "The boxes distribution in the N events in this box and 8 surrounding boxes" - do you mean a distribution in time or in space? All blocks of $N^{1/3}$ (not clear what that means) are then randomly assigned to the same event?

Minor comments:
- Lines 33-35: Given this article is about S. Africa, it would be worthwhile to point out that HAILCAST originated there with Poolman (1992).
- Lines 33-35: I would also note retrospective dynamical downscaling with NWP models can be used to generate climatologies that include the local and mesoscale processes listed here, particularly if convective permitting resolution is used.
- Line 40: The results of the Ayob (2019) study should be included for comparison with results from this study.
- Figs. 1 and 2 need to be switched to the correct order.
- Fig. 1: Given that, from my understanding, only the IR 10.8 brightness temperature is used in OT detection, it isn't clear how those OTs in Fig. 1a are identified as they don't align with local minima. Are they supposed to be northwest of every cold cloud peak?
- Line 96: Is Fig. 1a also supposed to be showing the IR anvil detection index?
- Fig. 6: With the two figures several pages apart, it is hard to see the difference between Figs. 6 and 3. Could they be combined into one figure with two side-by-side sub figures?
- Lines 111-113: What will the European and Australia reports be used for? Or are they reports of hail in S. Africa? Confusing.
- Line 111: Prein and Holland (2018) was not a study of observed data.

---

## Author Comment (AC1)

**Author comment to reviewer comments RC1:**

Review for NHESS-2021-342

*Characteristics of hail hazard in South Africa based on satellite*
*detection of convective storms*
*Heinz Jurgen Punge1, Kristopher M. Bedka2, Michael Kunz1, Sarah D. Bang3, and*
*Kyle F.*
*Itterly4*

*Summary.*
*This paper reviews and analyses the hail climatology for South Africa and regions*
*within South Africa using satellite detection of convective storms. The paper*
*investigates 14 years of geostationary satellite observations of convective storms*
*and generated a spatiotemporal multivariate stochastic model representing 25000*
*years. The historic, stochastic and observed insurance exposure and vulnerability*
*data are analysed to identify the expected hail damage for return periods of 200*
*years.*

*General Comments*
*Scientifically the authors have done an excellent job considering the literature*
*review, data collection and modelling that have gone into the research described in*
*this article. One slight drawback is that in the paper itself, the authors attempt to*
*address all the whole multiple complex modelling processes in a relatively short and*
*succinct manner. At times, unfortunately, the description of the process followed*
*does not do the modelling process justice and made it difficult to follow what was*
*done. Some examples will be given below.*
*The format of the paper also made reading the paper very difficult for me. At times,*
*the figures and tables mentioned in the text were not next to or near the text*
*referencing them. This caused a lot of scrolling up and down in pdf (and eventually I*
*just printed out the text in frustration). As times, figures were even placed to appear*
*to be part of a previous section e.g. Fig. 14 seems to form part of the end of Section*
*3.3 but is part of Section 3.4 that starts underneath Fig. 14. The format used by the*
*authors may be due to format instructions fromthe journal. If not, please reconsider*
*the placements of figures and tables to be as close as possible to the relevant text*
*to improve the reading flow of the paper.*

The authors thank the reviewer for the helpful and extensive comments. We will make an
effort to clarify the description of the modelling process and revise the arrangement of
figures and tables.

*Specific comments*
*25. The reference to Grieser and Hill (2019). Did Grieser and Hill focus on hailpad*
*derived metrics for South Africa, another country or just in general?*

The study of Grieser and Hilll uses data from CoCoRaHS, a volunteer-based network of
weather observations in the United States (Doesken and Reges 2011; Reges et al. 2016).
Data of this kind has been collected and published only for very few locations, and no

recent study was found for South Africa.We will specify the data origin in the revised version of the manuscript.

> 35. "hailstorm formation is often related to local and meso-scale processes related to, for example" Perhaps do not use the word related twice in one sentence.

The sentence will be rephrased.

> Figure 1 "a title: SRTM" acronym is not defined. Please check whole text for acronyms.

We will double check to make sure all acronyms are defined in the revised version.

> 95. "Based on past experience, only OTs detected with a probability >50% and with a surrounding anvil cloud (green and yellow colors in Fig. 1a; the IR anvil detection index, a rating based on an anvil detection model accounting for viewing situations, greater than 10; see also Scarino et al., 2020) are used in this work." Make the sentence in brackets a sentence on its own or add to figure description.

The sentence in the brackets will be moved to the figure caption.

> 110 - 115. "A uniformly distributed random number between -0.5 and 0.5 115 was added to each reported hail diameter to compensate" I assume the -0.5 and 0.5 is also in mm?

The article will be revised to use only cm as a unit for hail diameters. In this case, a random value between -0.5 and +0.5 cm has been added, accounting for the coarse classification of hail size in the databases.

> 155 "As 9.5% of the OTs occur at a melting level of less than 2 400 m, but ony 3.5% of the microwave hail detections and and 2.5% of the claims, a lower threshold of 2 400 m was introduced for this parameter." . Spelling. Sentence seems incomplete.

The spelling and grammar will be corrected.

> 160. "The latter feature is due to the minimum freezing level condition and remains to be confirmed by independent observation." Independent observation from whom?

Very few information on hail occurrence was found for that region; the most suitable method of ground verification would be a network of hail pads or sensors covering multiple regions of South Africa.

> 190: "complemented with hail size information from reporting." Reporting? You mean the insurance reporting?

In this case hail reports such as those registered in the hail databases are meant. This will be clarified in the text.

> 200. "Following Punge et al. (2014), both annual and daily cycles are modeled with Gaussian distributions. For the day of year, domains of 3° × 5° are considered, and depending on the..." Why Gaussian distribution? What is the statistical justification for it?

The occurrence of hail is linked to conditions on a set of variables that need to be fulfilled and can therefore be described as a convolution of the distributions of these variables. As such distributions, e.g. of insolation, are themselves usually continuous and often normal, their convolution can be expected to be normal as well.

*Not sure how these grid definitions relates to the previously defined rectangular grids of 0.3 x 0.5 mentioned on page 10.*

The grids of 3 x 5 group 10 x 10 of the smaller grids, with the intention of increasing the number of observations in each cell, large enough to derive characteristics of the distribution. This will be clarified.

*205: "Days are drawn from the boxes distribution for the..." It is not clear to what the boxes distributions refer to.*

It refers to the 3° x 5° boxes; this will be specified in the revised version.

*205: "...Finally, the day is retained only for N/9 events at random. This procedure has been found empirically to approximate the observed space-time distribution of days in a satisfactory manner. " Is this procedure self-developed or taken from somewhere? Why n^(1/3) and N/9 - those specific values? What is the proof of empirically proof behind it?*

The method is self-developed. We realize the description was somewhat imprecise. A division by nine is required as we draw nine times the required number of events: for the box concerned and the eight surrounding ones (queen criterion). Hail events cluster on this scale (15°x9°) due to synoptic processes. The empirical proof is that the distributions in Fig. 17 represent the observed distributions quite well with a single tuning parameter. We will improve the description to make this more clear.

*210: "from a region of 10° × 6° around.." Why the double grid size? Is this to also represent the 8 neighbouring grid cells? Paragraphs 205 and 210 can be extended to make the spatial construction more clear. In the current format is it difficult to follow, and relate back to standard spatial weight matrixes using the queen criterion.*

No, 10°x 6° region turned out to be a good choice for this parameter. The time of day is correlated on a smaller scale spatially as in a series of events, the later ones are shifted spatially with respect to the earlier ones. We will add this explanation.

*215: "Also note the secondary maximum in fall (around days 100–150, i.e. April and May) during nighttime, represented in the model. " Does this represent a local maximum? How do you see from the graphs it is in the night?*

This maximum is a local maximum in the historic events and ocurs at around day of year 140 and 6 UTC and can be discerned as an area of orange shades in that region of the plot. We will add this explanation to the text.

*215: " It is shifted towards fall over the Southern Ocean." ??? Are you modelling that far away from the shores of the country as well? And will it have any landfall impact?*

Off-shore events need to be represented as they can extend to the onshore coastal region. An impact of a far off-shore event is quite unlikely and will be marginal at this distance, but has been included for completeness.

> *220: "Time, slightly earlier than Smith et al. (1998, 5–6 pm) but consistent with Olivier (1990) (Fig. 11b). The daily cycle is most pronounced..." What is the possibility of there being a shift in these times from the 1990's to now? In that case, would the results be comparable?*

A shift in the diurnal distribution of severe convective storms cannot be excluded or proven with the data at hand and has sometimes been discussed in the context of climate change. A more likely explanation would be a different sensitivity to hail size in the two methodologies. Larger hail has a tendency to peak later in the day than small hail and Olivier may miss some of the smaller hail events, but at this point this is only speculative.

> *Figure 10: I assume the day of the year for 1 to 365 represents 1 Jan to 31 Dec. Perhaps add that to the title to indirectly show the difference in expected hail occurrences for northern and southern hemisphere?*

The assumption is correct and will be added to the figure caption.

> *Figure 11: I'm struggling with what number of days each bar represents. It seems the number 50 falls on the 4th bar?? This will only work if each bar represents 12.5 days?*

The days of year have been grouped in classes of 14 days in this figure. "50" hence roughly corresponds to weeks 7 and 8. An explanation will be added.

> *225: "The distributions are well approximated by the GEVs". GEV is an extreme distribution that requires a "limit" (e.g. peaks-over-threshold or block-maxima) in the data over which you are modelling events? What was that limit and how was it obtained?*

There is a possible misunderstanding here in that we are not applying extreme value theory here. Instead, we use GEV to approximate the distribution of *all* events, not only the most extreme ones. The text will be clarified in this respect.

> *230: " to give unrealistic large values, which is why length and width have been truncated at 1.5 times the largest observed values," Why the specific value of 1.5 times the largest observed value?*

*It turns out that these distances occur in events that are of a size comparable to South Africa, the domain of interest. Hence the cutoff has little practical implications. Other than that, there is no specific reason to choose this particular value.*

> *230: "In addition, the fraction f of the event area (the area of the ellipse spanned by major and minor axis of lengths l and w," Remember to write the last l and w in italics.*

This will be corrected in the revised version.

> *235: "Table 1 lists the distributions and parameters for these event properties." Which method was used to estimate the parameters of the distributions?*

We used the standard matlab mle function to obtain the maximum likelihood estimate of the distribution parameters.

> *240: "We find that most frequently, events have an orientation of around 100°, i.e., propagate eastward to southeastward (Fig. 11f)." This is for the whole country. But it may be misleading as this is not the typical orientation for a high hail fall region like Gauteng where storms normally originate in Johannesburg and move north-easterly to Pretoria. As seen from the discussion in the next paragraph.*

This is right, we will rephrase to avoid this confusion.

> *280: "sets of random numbers for each property from a uniform distribution and determine ranks. Then, for each property, we draw values from the actual distribution, sort them, and attribute to events using the pre-determined ranks. " How? Does this again refer to a previously defined or described methodology?*

The methodology has been specifically designed for this application and has been described by Punge et al 2014 regarding the European hail event set.

> *285: "could be expected, smaller regions show relatively higher variability, but there is strong correlation between the two. This" Which 2? Smaller regions and the country as a whole?*

Yes; we will include that there is a strong correlation between all regions.

> *295: This section describes the South African domain in terms of latitude and longitude degrees, subregions etc. Should this description not be done earlier in the paper to set the scene – perhaps where Figure 9 is defined*

We will add a paragraph near Fig. 9 to discuss the importance of these regions for hail hazard.

> *305: "50 hail days per year), while in an equivalent sample of subsets from the stochastic event set, the event count ranges from 1 883 to 2 162 events on 671 to 703 days" Perhaps add the equivalent hail events and days per year for comparison.*

The numbers will be added.

> *305-310: " In the Highveld region, there were 74 days per year..." These numbers are averages per year. The averages per year for the years defined by the authors and the years defined by Smith et al 1998 are different and it should be considered that several climate changes occurred in the years in between. This includes periods of severe drought in the several regions in the country especially between 2010 -2020*

We will add a comment highlighting the difference in methodology and the possible impact of climate change and variability.

> *310: Can the numbers given in this paragraph be added in a table for easier reference?*

A table will be added to cover the numbers.

*310: "However, severe hail (>31 mm in diameter" Why 31 mm and not a round number like 30mm?*

The size threshold was chosen to match that of Smith et al. But we realize they use >= 31mm and hailstone diameters are given in 1mm classes. We will use >30mm instead, which is indeed more intuitive and should not differ too much from Smith's approach.

---

## Author Comment (AC2)

**Author comment to reviewer comments RC2:**

Review for NHESS-2021-342 Characteristics of hail hazard in South Africa based on satellite detection of convective storms Recommendation: Reject

The authors have assembled a novel methodology for estimating hailfall in South Africa, a region with frequent hailfall but sparse observations. Such a hailfall climatology in South Africa is clearly needed, particularly with the possibility of shifting or increasing hailfall frequency with a changing climate. From this hail event climatology, they have additionally assembled a statistical model that includes estimations of hail size, hail swath shape and orientation, and frequency of occurrence over a much longer period. The creation of all these products is ambitious, but I feel the authors have overreached what is scientifically defensible though the necessary chain of assumptions. I don't doubt that there is a strong operational need for extended hail climatology products like these in this region, but if choosing to publish the work the assumptions must be reasonably defended. In sum, I recommend rejection in the paper's current form, but would welcome reviewing a resubmission on a narrow, better-grounded portion of the work.

The authors appreciate the reviewer thorough assessment of our manuscript. As the reviewer points out, to cover the entire modeling process from satellite detection of storms to the stochastic footprints can appear ambitious. However, for each of the steps, we can build on existing publications where similar assumptions had to be made, and focus on improving the methodology in the best possible way on the basis of the available data. Presenting all those steps in a succint way and in one article will benefit other authors pursuing similar objectives or trying to test the accuracy of our results and therefore benefit scientific exchange as a whole. Naturally, the supporting evidence is clearer and the assumptions to be made are weaker in the portions of the work directly dealing with observations compared to the modelling part; this is however a common situation in atmospheric science. To address the reviewers concerns, during the revision of our work for final publication, we will particularly stress the assumptions made and caution needed in interpreting the results. As an example, the use of hail diameters in the model portion may be susceptible to over-interpretation, which is why we will instead center on the aspect of hail severity and avoid the use of diameters.

**Major comments/fatal flaws:**

The work performed here was obviously extensive, and I appreciate the effort to scientifically ground an operational product. I've broken down my view of the chain of reasoning presented in the paper, along with my opinion of how well each step is grounded in the article.

1. Hail occurrence can be estimated using the Khlopenkov et al. (2021) OT detections in GOES data over CONUS. This step is well-grounded, given Khlopenkov et al. and Cooney et al. (2021) results discussed in the introduction, although a quick sentence or two discussing the skill level of that algorithm with the severe hail report database used in those studies would be useful to add.

The skill level of the algorithm has been assessed against severe hail reports, MESH radar and satellite products and the respective information will be added to the publication. Table 1 summarizes the results, which we suggest to put in the appendix. We can detect updrafts near to or above the tropopause with about a 60% success rate using data from GOES-13 a proxy for Meteosat (Cooney et al. 2021). This shows that through the use of OTprob > 0.5 to refine detections to those we are most confident in, we lose about 45% of the probably hail-producing storms. In other words, many severe hail storms can look quite "boring" from a satellite infrared perspective, but the boring ones are hard to differentiate from false OT detections in anvils (i.e. detections in cold outflow near to real OTs). The fact that there can be some time uncertainty between report time or the time a radar scanned a storm vs the time of OT detections may also influence our results. For example, an OT may have been really prominent 3 min before the time of a report/MESH, but we only have just the one GOES image to match. We see the agreement with microwave may be lower than with MESH or reports because of parallax shifts in the storm positions in microwave data, especially those close to the limb of the overpass. We will put a comment on that in the revised manuscript.

Table 1: Detection counts and fractions of CONUS 2013-2016 GOES-13 derived parameters matched within 28 km2 and 15 minutes of hail reports with various GOES OT Probability conditions applied.

|            | Count | Count with | Fraction with | Count with | Fraction with |
|------------|-------|------------|---------------|------------|---------------|
|            |       | otProb     | otProb        | otProb>=50 | otProb>=50    |
| MESH95     | 72048 | 49866      | 0.69          | 27268      | 0.38          |
| >=40 cm    |       |            |               |            |               |
| MESH75     | 92066 | 61340      | 0.67          | 32222      | 0.35          |
| >=25 cm    |       |            |               |            |               |
| SPC Hail   | 35049 | 24478      | 0.70          | 14055      | 0.40          |
| >= 25 cm   |       |            |               |            |               |
| MWR P_hail | 1740  | 892        | 0.51          | 535        | 0.31          |
| >=50       |       |            |               |            |               |

2. The Khlopenkov et al. OT algorithm can be applied to MSG SEVIRI data over S. Africa with similar success as GOES data over CONUS, with the additional environmental filtering applied. This claim is generally supported by the results in the paper (c.f., Figs. 3 and 4), but needs a fuller explanation. The geographic hotspots are similar in Figs. 3 and 4, but is the frequency of potential hail occurrences reasonable? Comparison of OTs, GPM/TRMM detections, and radarbased detections over CONUS could confirm the relative change in frequency between OT and GPM/TRMM detections over S. Africa is reasonable. Comparisons should also be made to climatologies made over the region from other methods, such as those discussed in the introduction (Admirat et al. 1985; Prein and Holland 2018; Kunz et al. 2020; Dyson et al. 2020).

There is an inherent difficulty in comparing hail frequency estimates based on different methods, in particular when it is not possible to directly compare each occurrence of hail (see, e.g., the review of Punge and Kunz (2016) on hail frequency estimates in Europe). The nature of the model-based studies cited here and the satellite-based approaches presented in this study are such that such a direct comparison is hardly possible for South Africa.

As TRMM and GPM were/are satellites in inclined orbits, their sampling is not continuous, like a geostationary satellite. It is therefore not reasonable to use the absolute counts from these satellites as indication of the true frequency of hailstorms. When assessing the gridded climatologies as in Bang and Cecil (2019; their figure 7), the values are scaled to account

for the sampling. Comparing the CONUS hail events/year to the radar methodology of Cintineo et al. (2012; their figure 9), we see a GPM climatology over the central US that ranges from ~6-13 events per year, while the radar-derived climatology estimates about ~4-12 hail days (note events versus days). A US climatology using MESH to estimate the presence of hail from Murillo et al. (2021) shows a frequency of 3-7 hail days per year over the central US. This substantiates our confidence in the passive-microwave hail retrievals. We will add to the discussion of differences with other publications on hail in the region.

3. The hail grouping methodology into events reasonably represents hail swaths from a single storm system. While the description of the methodology (lines 176-177) is intuitive and simple, the results of the grouping methodology in Fig. 7 don't seem to follow that description. Why are there multiple events occurring at a single place and time? Once the methodology itself is cleaned up, a few example applications of this methodology in an area with radar data would show its value in establishing hail events and their duration and speed. Right now, the results of the methodology are only briefly compared in text to two other radar-based studies of severe convective storms (not limited to hailstorms) in the literature.

In the given example, there are indeed overlapping events. Those simply are too distanced in time and space (in particular time) to be grouped into the same event. The algorithm is designed to follow storms related to a common conditions or trigger, such as a propagating front triggering storms along its way. Hence, the temporal aspect will receive stronger emphasis in the revised article. We will comment on this in the text and may add an example with radar for the US to the appendix

4. The created hail event climatology shows reasonable distributions of hail event frequency by time of year and time of day. No comparison of these distributions is

made to the observational or GPM/TRMM datasets. While they are admittedly sparse, they should at least be able to confirm general seasonality. Comparisons should also be made to the other climatology datasets mentioned in point 2 above.

Below there are the four seasonal frequencies of hailstorms as well as the diurnal cycle as seen by TRMM and GPM combined.

We will provide a direct comparison of these seasonal climatologies to the OT-based estimate in the appendix.

---

## Author Response (AR1)

The authors thank the two reviewers for their helpful and detailed comments. We have considered all the points they listed. We went through the entire paper again (several times), trying to improve clarity and readability. We have deleted two of the original figures that we felt were not really necessary, but have added additional figures in a new appendix.
The added text in the revised manuscript with track highlighting is marked in red.

**Author comments to reviewer comments RC1:**

Review for NHESS-2021-342

*Characteristics of hail hazard in South Africa based on satellite detection of convective storms*
*Heinz Jurgen Punge1, Kristopher M. Bedka2, Michael Kunz1, Sarah D. Bang3, and Kyle F.*
*Itterly4*

*Summary.*
*This paper reviews and analyses the hail climatology for South Africa and regions within South Africa using satellite detection of convective storms. The paper investigates 14 years of geostationary satellite observations of convective storms and generated a spatiotemporal multivariate stochastic model representing 25000 years. The historic, stochastic and observed insurance exposure and vulnerability data are analysed to identify the expected hail damage for return periods of 200 years.*

*General Comments*
*Scientifically the authors have done an excellent job considering the literature review, data collection and modelling that have gone into the research described in this article. One slight drawback is that in the paper itself, the authors attempt to address all the whole multiple complex modelling processes in a relatively short and succinct manner. At times, unfortunately, the description of the process followed does not do the modelling process justice and made it difficult to follow what was done. Some examples will be given below.*

*The format of the paper also made reading the paper very difficult for me. At times, the figures and tables mentioned in the text were not next to or near the text referencing them. This caused a lot of scrolling up and down in pdf (and eventually I just printed out the text in frustration). As times, figures were even placed to appear to be part of a previous section e.g. Fig. 14 seems to form part of the end of Section 3.3 but is part of Section 3.4 that starts underneath Fig. 14. The format used by the authors may be due to format instructions from the journal. If not, please reconsider the placements of figures and tables to be as close as possible to the relevant text to improve the reading flow of the paper.*

The authors thank the reviewer for the helpful and extensive comments. We considered all points listed below in the revised version of the manuscript (replies in blue).

As suggested, we improved the description of the modelling process. We are sorry for the wrong placement of the figures and Tables. We rearranged their appearance in order to improve readability of the paper.

*Specific comments*
*25. The reference to Grieser and Hill (2019). Did Grieser and Hill focus on hailpad derived metrics for South Africa, another country or just in general?*

The study of Grieser and Hilll uses data from CoCoRaHS, a volunteer-based network of weather observations in the United States (Doesken and Reges 2011; Reges et al. 2016). Data of this kind has been collected and published only for very few locations, and no recent study was found for South Africa. We have specified the data origin in the revised version of the manuscript.

*35. "hailstorm formation is often related to local and meso-scale processes related to, for example" Perhaps do not use the word related twice in one sentence.*

Sentence rephrased ("...hailstorm often form by…").

*Figure 1 "a title: SRTM" acronym is not defined. Please check whole text for acronyms.*

We have defined SRTM, but deleted the acronym as it is only used here. In addition, we double checked the manuscript to make sure all acronyms are defined in the revised version.

*95. "Based on past experience, only OTs detected with a probability >50% and with a surrounding anvil cloud (green and yellow colors in Fig. 1a; the IR anvil detection index, a rating based on an anvil detection model accounting for viewing situations, greater than 10; see also Scarino et al., 2020) are used in this work." Make the sentence in brackets a sentence on its own or add to figure description.*

As suggested, we moved the sentence in the brackets to the figure caption.

*110 - 115. "A uniformly distributed random number between -0.5 and 0.5 was added to each reported hail diameter to compensate" I assume the -0.5 and 0.5 is also in mm?*

Sorry that we did not included the unit; the random value accounting for the coarse classification of hail sizes in the database is actually between -0.5 and +0.5 cm. We have corrected this.

*155 "As 9.5% of the OTs occur at a melting level of less than 2 400 m, but ony 3.5% of the microwave hail detections and and 2.5% of the claims, a lower threshold of 2 400 m was introduced for this parameter." . Spelling. Sentence seems incomplete.*

We corrected both the spelling and grammar.

*160. "The latter feature is due to the minimum freezing level condition and remains to be confirmed by independent observation." Independent observation from whom?*

Very few information on hail occurrence was found for that region; the most suitable method of ground verification would be a network of hail pads or sensors covering multiple regions of South Africa. This was added to the sentence.

*190: "complemented with hail size information from reporting." Reporting? You mean the insurance reporting?*

In this case hail reports such as those registered in the hail databases are meant. This is now clarified in the text.

*200. "Following Punge et al. (2014), both annual and daily cycles are modeled with Gaussian distributions. For the day of year, domains of 3° × 5° are considered, and depending on the..." Why Gaussian distribution? What is the statistical justification for it?*

The occurrence of hail is linked to conditions on a set of variables that need to be fulfilled and can therefore be described as a convolution of the distributions of these variables. As such distributions, e.g. of insolation, are themselves usually continuous and often normal, their convolution can be expected to be normal as well.
According to literature, both the diurnal and annual cycles can be well approximated by a normal distribution. We added an explanation with references to the manuscript.

*Not sure how these grid definitions relates to the previously defined rectangular grids of 0.3 x 0.5 mentioned on page 10.*

The grids of 3 x 5 group 10 x 10 of the smaller grids, with the intention of increasing the number of observations in each cell, large enough to derive characteristics of the distribution. This is now explained in the manuscript.

*205: "Days are drawn from the boxes distribution for the..." It is not clear to what the boxes distributions refer to.*

It refers to the 3° x 5° boxes; this is specified in the revised version.

*205: "...Finally, the day is retained only for N/9 events at random. This procedure has been found empirically to approximate the observed space-time distribution of days in a satisfactory manner. " Is this procedure self-developed or taken from somewhere? Why n^(1/3) and N/9 - those specific values? What is the proof of empirically proof behind it?*

The method is self-developed. We realise the description was somewhat imprecise. A division by nine is required as we draw nine times the required number of events: for the box concerned and the eight surrounding ones (queen criterion). SCS/hail events preferably cluster on this scale (15°x 9°) due to synoptic processes. The empirical proof is that the distributions in Fig. 17 represent the observed distributions quite well with a single tuning parameter. We improved the description to make this more clear and added two references on the relation between synoptic processes and SCS clusters.

*210: "from a region of 10° × 6° around.." Why the double grid size? Is this to also represent the 8 neighbouring grid cells? Paragraphs 205 and 210 can be extended to make the spatial construction more clear. In the current format is it difficult to follow, and relate back to standard spatial weight matrixes using the queen criterion.*

The 10°x 6° region turned out to be a good choice for this parameter. The time of day is correlated on a smaller scale spatially as in a series of events, the later ones are shifted spatially with respect to the earlier ones. We have added this explanation.

> 215: "Also note the secondary maximum in fall (around days 100–150, i.e. April and May) during nighttime, represented in the model. " Does this represent a local maximum? How do you see from the graphs it is in the night?

This maximum is a local maximum in the historic events and occurs at around day of year 140 and 6 UTC and can be discerned as an area of orange shades in that region of the plot. We have added this explanation to the text.

> 215: " It is shifted towards fall over the Southern Ocean." ??? Are you modelling that far away from the shores of the country as well? And will it have any landfall impact?

Off-shore events need to be represented as they can extend to the onshore coastal region. An impact of a far off-shore event is quite unlikely and will be marginal at this distance, but has been included for completeness. We have added an explanation.

> 220: "Time, slightly earlier than Smith et al. (1998, 5–6 pm) but consistent with Olivier (1990) (Fig. 11b). The daily cycle is most pronounced..." What is the possibility of there being a shift in these times from the 1990's to now? In that case, would the results be comparable?

A shift in the diurnal distribution of severe convective storms cannot be excluded or proven with the data at hand and has sometimes been discussed in the context of climate change. A more likely explanation would be a different sensitivity to hail size in the two methodologies. Larger hail has a tendency to peak later in the day than small hail and Olivier (1990) may miss some of the smaller hail events. As this explanation is speculative, we haven't included it in the manuscript.

*Figure 10: I assume the day of the year for 1 to 365 represents 1 Jan to 31 Dec. Perhaps add that to the title to indirectly show the difference in expected hail occurrences for northern and southern hemisphere?*

The assumption is correct and was added to the figure caption.

> Figure 11: I'm struggling with what number of days each bar represents. It seems the number 50 falls on the 4th bar?? This will only work if each bar represents 12.5 days?

The days of year have been grouped in classes of 14 days in this figure. "50" hence roughly corresponds to weeks 7 and 8. We added an explanation to the caption.

> 225: "The distributions are well approximated by the GEVs". GEV is an extreme distribution that requires a "limit" (e.g. peaks-over-threshold or block-maxima) in the data over which you are modelling events? What was that limit and how was it obtained?

There is a possible misunderstanding here in that we are not applying extreme value theory here. Instead, we use the GEV to approximate the distribution of *all* events, not only the most extreme ones. The is now clarified in this respect.

> 230: " to give unrealistic large values, which is why length and width have been truncated at 1.5 times the largest observed values," Why the specific value of 1.5 times the largest observed value?

*It turns out that these distances occur in events that are of a size comparable to South Africa, the domain of interest. Hence the cutoff has little practical implications. Other than that, there is no specific reason to choose this particular value. We added an explanation on this.*

> 230: "In addition, the fraction f of the event area (the area of the ellipse spanned by major and minor axis of lengths l and w," Remember to write the last l and w in italics.

This is now corrected.

> 235: "Table 1 lists the distributions and parameters for these event properties." Which method was used to estimate the parameters of the distributions?

We used the standard matlab mle function to obtain the maximum likelihood estimate of the distribution parameters and included an explanation in the caption.

> 240: "We find that most frequently, events have an orientation of around 100°, i.e., propagate eastward to southeastward (Fig. 11f)." This is for the whole country. But it may be misleading as this is not the typical orientation for a high hail fall region like Gauteng where storms normally originate in Johannesburg and move north-easterly to Pretoria. As seen from the discussion in the next paragraph.

This is right; we have added this specification to avoid any confusion.

> 280: "sets of random numbers for each property from a uniform distribution and determine ranks. Then, for each property, we draw values from the actual distribution, sort them, and attribute to events using the pre-determined ranks. " How? Does this again refer to a previously defined or described methodology?

The methodology has been specifically designed for this application and has been described by Punge et al (2014) regarding the European hail event set; we added the reference to the text.

> 285: "could be expected, smaller regions show relatively higher variability, but there is strong correlation between the two. This" Which 2? Smaller regions and the country as a whole?

Yes; we have included a statement that there is a strong correlation between all regions.

> 295: This section describes the South African domain in terms of latitude and longitude degrees, subregions etc. Should this description not be done earlier in the paper to set the scene – perhaps where Figure 9 is defined

We have added a paragraph near Fig. 9 to discuss the importance of these regions for hail hazard.

> 305: "50 hail days per year), while in an equivalent sample of subsets from the stochastic event set, the event count ranges from 1 883 to 2 162 events on 671 to 703 days" Perhaps add the equivalent hail events and days per year for comparison.

The numbers were added.

> 305-310: " In the Highveld region, there were 74 days per year..." These numbers are averages per year. The averages per year for the years defined by the authors and the years defined by Smith et al 1998 are different and it should be considered that several climate changes occurred in the years in between. This includes periods of severe drought in the several regions in the country especially between 2010 -2020

We have added a comment highlighting the difference in methodology and the possible impact of climate change and variability.

> 310: Can the numbers given in this paragraph be added in a table for easier reference?

We included a new table (Table 3) to cover the numbers.

> 310: "However, severe hail (>31 mm in diameter" Why 31 mm and not a round number like 30mm?

The size threshold was chosen to match that of Smith et al. But we realise they use >= 31mm and hailstone diameters are given in 1mm classes. We therefore changed to >30mm instead, which is indeed more intuitive and should not differ too much from Smith's approach.

**Author comments to reviewer comments RC2:**

*Review for NHESS-2021-342*
*Characteristics of hail hazard in South Africa based on satellite detection of convective storms*
*Recommendation: Reject*

*The authors have assembled a novel methodology for estimating hailfall in South Africa, a region with frequent hailfall but sparse observations. Such a hailfall climatology in South Africa is clearly needed, particularly with the possibility of shifting or increasing hailfall frequency with a changing climate. From this hail event climatology, they have additionally assembled a statistical model that includes estimations of hail size, hail swath shape and orientation, and frequency of occurrence over a much longer period. The creation of all these products is ambitious, but I feel the authors have overreached what is scientifically defensible though the necessary chain of assumptions. I don't doubt that there is a strong operational need for extended hail climatology products like these in this region, but if choosing to publish the work the assumptions must be reasonably defended. In sum, I recommend rejection in the paper's current form, but would welcome reviewing a resubmission on a narrow, better-grounded portion of the work.*

The authors appreciate the reviewer's thorough assessment of our manuscript. As the reviewer points out, to cover the entire modeling process from satellite detection of storms to the stochastic footprints can appear ambitious. However, for each of the steps, we can build on existing publications where similar assumptions had to be made, and focus on improving the methodology in the best possible way on the basis of the available data. Presenting all those steps in a succinct way and in one article will benefit other authors pursuing similar objectives or trying to test the accuracy of our results and therefore benefit scientific exchange as a whole. Naturally, the supporting evidence is clearer and the assumptions to be made are weaker in the portions of the work directly dealing with observations compared to the modelling part; this is however a common situation in atmospheric science. To address the reviewers concerns, during the revision of our work for final publication, we will particularly stress the assumptions made and caution needed in interpreting the results. As an example, the use of hail diameters in the model portion may be susceptible to over-interpretation, which is why we will instead center on the aspect of hail severity and avoid the use of hail diameters in the hazard part of the model.

However, hail size as a measure of intensity is needed in the stochastic model to estimate hail risk for a given portfolio. These sizes do not derived from OT intensity estimates, but are modelled stochastically based on observed hail size spectra. This approach is the only way to provide hail risk assessments. It is also implemented more or less in all catastrophe (risk) models.

**Major comments/fatal flaws:** *The work performed here was obviously extensive, and I appreciate the effort to scientifically ground an operational product. I've broken down my view of the chain of reasoning presented in the paper, along with my opinion of how well each step is grounded in the article.*

1. *Hail occurrence can be estimated using the Khlopenkov et al. (2021) OT detections in GOES data over CONUS. This step is well-grounded, given Khlopenkov et al. and Cooney et al. (2021) results discussed in the introduction, although a quick sentence or two discussing the skill level of that algorithm with the severe hail report database used in those studies would be useful to add.*

The skill level of the algorithm has been assessed against severe hail reports, MESH radar and satellite products. We have summarized the most important findings in Section 2.1, but most of the respective information and details were added to the appendix (including a Figure and the Table shown below).

"To better emulate the present study methodology, MESH cell objects exceeding 2 pixels in area (10 km$^2$) and spaced by at least 28-km are derived using watershed segmentation applied to the hourly 10 mm+ MESH95 climatology (Bowman and Homeyer, 2017) over CONUS between 2013-2017 using the open-source Tracking and Object Based Analysis of Clouds (tobac v1.2; Heikenfeld et al. 2019) Python package. Further, following Murillo et al. (2021), we have applied Linear Discriminant Analysis (LDA) using their coefficients to combine precipitable water and 0-6-km shear to filter out likely false alarms.

Cooney et al. (2021) showed that we can detect updrafts near to or above the tropopause with about a 60% success rate using data from GOES-13 a proxy for Meteosat (Cooney et al. 2021). Table 1 compares the frequency of GOES-13, GOES-16, and MSG embedded cold spot (ECS) detections, e.g. areas that appear distinctly colder than the surrounding anvil and are considered to be OT candidates, and OT detections (OT probability $>= 0.5$) matching various hail detections from radar cells, ground spotter reported hail size, and MWR hail detections. Requiring OT probability >= 0.5 to refine severe hail detections to those we are most confident in, we lose 56% (46%) of the severe hail-producing storms exceeding 40 cm MESH95 maxima for GOES-13 (GOES-16). In other words, many severe hailstorms can look quite "boring" from a satellite infrared perspective, but the boring ones are hard to differentiate from false OT detections in anvils (i.e. detections in cold outflow near to real OTs). Uncertainty between report time or the time a radar scanned a storm vs the time of OT detections may also influence our results. For example, an OT may have been prominent several minutes before the time of a hail detection, but we only have a single GOES snapshot to match. By relaxing the matching criterion to ECS detections, we lose only 30 (17%) of likely severe hail producing cells for GOES-13 (GOES-16).

The frequency of geostationary updraft detections that are co-located with microwave hail detections is comparable to MESH95 and ground spotter severe hail reports despite added uncertainty due to parallax shifts in the storm positions in microwave data, especially those close to the limb of the overpass. Enabled by the global coverage of MWRs, Table 1 shows that 2005-2018 MSG SEVIRI ECS and OT detections over South Africa match with likely severe MW hail detections with frequencies similar to GOES-13 over CONUS. Although the total number of matches is relatively low over South Africa, this suggests that MSG IR-based updraft detections agree with independent hail detections; thus, supporting the use of MSG SEVIRI to detect hail cores over South Africa."

*2. The Khlopenkov et al. OT algorithm can be applied to MSG SEVIRI data over S. Africa with similar success as GOES data over CONUS, with the additional environmental filtering applied. This claim is generally supported by the results in the paper (c.f., Figs. 3 and 4), but needs a fuller explanation. The geographic hotspots are similar in Figs. 3 and 4, but is the frequency of potential hail occurrences reasonable? Comparison of OTs, GPM/TRMM detections, and radar-based detections over CONUS could confirm the relative change in frequency between OT and GPM/TRMM detections over S. Africa is reasonable. Comparisons should also be made to climatologies made over the region from other methods, such as those discussed in the introduction (Admirat et al. 1985; Prein and Holland 2018; Kunz et al. 2020; Dyson et al. 2020).*

There is an inherent difficulty in comparing hail frequency estimates based on different methods, in particular when it is not possible to directly compare each occurrence of hail (see, e.g., the review of Punge and Kunz (2016) on hail frequency estimates in Europe). The nature of the model-based studies cited here and the satellite-based approaches presented in this study are such that such a direct comparison is hardly possible for South Africa.

As TRMM and GPM were/are satellites in inclined orbits, their sampling is not continuous, like a geostationary satellite. It is therefore not reasonable to use the absolute counts from these satellites as indication of the true frequency of hailstorms. When assessing the gridded climatologies as in Bang and Cecil (2019; their figure 7), the values are scaled to account for the sampling. Comparing the CONUS hail events/year to the radar methodology of Cintineo et al. (2012; their figure 9), we see a GPM climatology over the central US that ranges from ~6-13 events per year, while the radar-derived climatology estimates about ~4-12 hail days (note events versus days). A US climatology using MESH to estimate the presence of hail from Murillo et al. (2021) shows a frequency of 3-7 hail days per year over the central US. This substantiates our confidence in the passive-microwave hail retrievals. We have added a statement about that to the discussion of differences with other publications on hail in the region.

> *3. The hail grouping methodology into events reasonably represents hail swaths from a single storm system. While the description of the methodology (lines 176-177) is intuitive and simple, the results of the grouping methodology in Fig. 7 don't seem to follow that description. Why are there multiple events occurring at a single place and time? Once the methodology itself is cleaned up, a few example applications of this methodology in an area with radar data would show its value in establishing hail events and their duration and speed. Right now, the results of the methodology are only briefly compared in text to two other radar-based studies of severe convective storms (not limited to hailstorms) in the literature.*

In the given example, there are indeed overlapping events. Those simply are too distanced in time and space (in particular time) to be grouped into the same event. The algorithm is designed to follow storms related to a common conditions or trigger, such as a propagating front triggering storms along its way. Hence, the temporal aspect will receive stronger emphasis in the revised article. We have commented on this in the text.

> *4. The created hail event climatology shows reasonable distributions of hail event frequency by time of year and time of day. No comparison of these distributions is made to the observational or GPM/TRMM datasets. While they are admittedly sparse, they should at least be able to confirm general seasonality. Comparisons should also be made to the other climatology datasets mentioned in point 2 above.*

In the Appendix, we have included four seasonal frequencies of hailstorm detections based on overshooting top activity (2004-2018) and passive microwave hail retrievals (1998-2018) for austral summer, autumn, winter, and spring. Additionally, we have included two Figures showing the diurnal cycle from OTs and as seen by TRMM and GPM combined.

It turned out that daily cycle of OT activity is more pronounced compared to that for the microwave detection, and its maximum sets in almost an hour earlier. Overall the agreement between the two satellite climatologies, however, is very good, but the OT algorithm may be slightly too sensitive for weaker convection. Concerning seasonality, it turned out that both methods indicate widespread hail activity quite well, with some discrepancies depending on the season. All these points are briefly discussed in the Appendix.

*5. The statistical method established in lines 202-213 can be used to produce similar hail event daily and seasonal hail event variations established by points 1-4 above (assuming points 1-4 are successful at representing actual hailfall). The annual and daily distributions produced by the model do appear similar – I'd prefer a difference plot instead of a side-by-side comparison, given the relatively large magnitudes involved. However, the description of the statistical method is not clear, and only one reference is cited. How common are methods like these? The steps involved in its description are very specific, making one wonder if the model is being over-fit to its underlying dataset.*
*How similar is the methodology used here to Punge et al. (2014, unfortunately behind a paywall), what changes were made, and why?*

A difference plot daily and seasonal hail event variation is included, now Figure 10c. In addition, the presentation of the methodology was revised. The methods are a recoded version of the 2014 article. In the redesign, the use of a von-Mises distribution was considered for modeling the periodic variables, at the cost of losing information on the shape of the distributions.

*6. The statistical method in lines 225-238 can be used to produce similar hail event length, width, area, and orientation as the event climatology produced in point 3 above (again, assuming point 3 is valid). These results do seem reasonable as presented in Fig. 11, but no point of comparison is provided. How well do other statistical methods perform? What is expected behavior?*

Indeed, a host of different options exist to model such relationships, including machine learning models, which may perform better, but the metrics and parameters will have to be chosen carefully to consider all event properties adequately and avoid overfitting and other issues. We instead opted to build on the methodology developed for the Punge et al. 2014 article, which is explainable, reproducible, and uses a rather small set of parameters.

*7. Hail size can be estimated using the OT climatology product produced in point 2 (I don't think the event climatology from point 4 is being used here, but text isn't clear). This claim is (currently) indefensible.*

To avoid a possible misunderstanding, we clarified the use of the OT product as a proxy for hail intensity (but not size). We do appreciate that a significant amount of uncertainty remains on the exact relation of OT strength and maximum reported hail size.
For the sake of modelling, we do not require this relation, and just need to assume that the size-extent relation holds on average, thus the strongest storms in terms of updraft occur in the largest systems as determined per event detection procedure. Maximum hail sizes in the stochastic event set and the model's event catalogue are drawn from hail size distributions in reports (ESWD or other). Relations between event properties are used only for the problem of matching those hail sizes to the other event characteristics in the stochastic events. We have included some statements to make that more clear.

*Marion et al. (2019) suggested a relationship between OT area, not strength, with updraft width and hence potential tornadic intensity. That's a not insignificant difference. Hail size, particularly as one reaches larger hail sizes, is more related*

*to updraft width than updraft strength (e.g., Nelson 1983, Foote 1984; Kumjian et al. 2021). I am concerned that by relating hail size to an updraft strength metric, an erroneous hail size distribution will be produced.*

We agree that the focus and findings of Marion et al. are different from our study. While tornadic activity may indeed rather be related to the size of a storm system, it is still a reasonable assumption that hail size is related to updraft strength rather than size. There is no doubt that besides updraft strength, other factors like the buoyancy in the UTLS region and timing of the imagery relative to peak intensity also impact the measured cloud top temperature differences. In the manuscript, we have deleted statements about hail sizes derived from OT detections. But we briefly present in the appendix a relation between hail size parameters based on radar – where the spatial match can be expected to be better than with hail reports – and the OT- anvil temperature. The inherent assumption is thus that other factors will average out when the sample size is big enough, which we are confident is the case.

*Khlopenkov et al. (2021) connected OT detection probability with hail occurrence and did not try to distinguish among hail sizes.*

Actually Figure 11 of Khlopenkov et al. (2021) showed that increasing embedded cold spot (ECS) IR-anvil brightness temperature difference and decreasing IR-tropopause temperature difference, both metrics of storm intensity, were more extreme for 2+ inch hail reports than hail < 2 inch in diameter. But again, we have deleted the part of hail size estimation using OT data.

*Figure 2 appears to represent original work from the authors (sentence is oddly phrased, making it seem like it is sourced from Murillo and Homeyer 2019). While I do appreciate the correlation shown, I am concerned the MESH95 dataset is being used, and not actual hail reports. Per Murillo and Homeyer, the MESH95 dataset has a significant large bias, with 40 mm being most skillful at determining 25 mm hail, and 64 mm being most skillful at determining 50 mm hail. That bias does not appear to be accounted for in Fig. 2. Further, while Murillo and Homeyer (2019) did not specifically examine the skill of tropospheric-OT temperature difference in differentiating among hail sizes, they did examine the distribution of minimum GOES IR Brightness and GOES OT Area (see their Figs. 6a, b, 8a, b), and did not find a strong relationship between those fields and observed hail size.*

As indicated above, we do appreciate the uncertainty in relating of maximum reported hail size, average size of hailstones (which may be a better proxy for the damage), and hail metrics based on radar or satellite sources. There is significant need for further research in this area for improved and more reliable modeling, this aspect is stressed in the revised discussion. Further, we have deleted the original Figure 2, but provide another Figure about the relationship in the Appendix.

After the initial submission, we have made several methodology tweaks to the radar and satellite matching to remove view angle dependencies from the geostationary measurements and to ensure highly confident matches by requiring larger cell objects, which better emulates the present study's methodology. Specifically, ECS-Anvil BTD is normalized by the effective grid resolution degradation compared to nadir and MESH cells under 10 km² are excluded. The positively biased MESH95 distribution does not change the strength of the correlation among the GOES parameters compared to the more

realistic values of MESH75, which is the primary takeaway from Figure 2. We have noted the positive MESH95 bias in the text to aid the interpretability of the estimated hail size bins and included multiple thresholds of MESH95 (25 mm and 40 mm) in Table 1 to illustrate this bias relative to the ground reports.

Without removing view angle dependence, the prominence of an OT observed at low VZA (e.g., <40° in the Southeastern US) would be greater than had this same OT occurred at high VZA (e.g., >50° in the Northern Plains) due to differences in the effective pixel resolution. We derive the normalization factor based on the effective footprint area of GOES relative to the nadir footprint area (16 km2 for GOES-13) to account for the imager's reduced ability to observe the prominence of a colder pixel relative to the background anvil at higher view angles. The formulae to derive the x and y component of pixel resolution for are shown in the Appendix for GOES-13.
The normalization results in improved correlations between ECS-Anvil BTD and MESH95 for GOES-12/13 and GOES-16 shown in box and whisker plots (Fig. A1). We have summarized this discussion in the paper's appendix section to justify use of a satellite metric of updraft intensity to estimate storm severity.

We feel that Murillo and Homeyer did not find a correlation between satellite parameters and reported hail size because of uncertainties in hail size reporting. Though MESH has uncertainties as well, it is spatially and temporally consistent, unlike reporting and the relationships described above indicating a correlation between storm intensity and hail size are robust.

> *In my opinion, this claim cannot be supported given the current literature, and hail sizes should be removed from the database (or only provided to customers with a strong caution about their use, and not published in the literature).*

It is clear that improved data and methodology may yield more accurate results in terms of estimated hail size distributions, and the ones presented in our model may be proven wrong. Still, our work may serve as a reference to such studies. To account for the large uncertainty, we have rephrased all text mentioning hail size into hail intensity estimate. The uncertainties have always been communicated quite clearly with the model users, and results are generally not used directly for pricing insurance premiums but as a general indicator of risk. There are additional uncertainties in the exposure and vulnerability models that users have to deal with.

> *Given these issues above, I cannot recommend the article for acceptance. I would be happy to review an article focusing on points 1-4 above, after addressing the issues I've described. A companion paper focusing on points 5-6, after points 1-4 are successfully established, would also be interesting. I cannot support an article including point 7 at the current time.*

We encourage the reviewer to revise his/her decision in light of the further explanations and material presented and would welcome her/his continued guidance in the review process.

---

## Referee Report (RR1)

**Review of**

**Characteristics of hail hazard in South Africa based on satellite detection of convection storms**

**By Punge et al**

**Summary**

This article is the second revision of the paper with the same title. In this paper, the authors investigate the formation of hail events and hail hazard in South Africa as guided by 14 years of geostationary satellite observations storms. A multivariate stochastic model was built simulating event properties spanning 25 000 years of hail occurrence using overshooting cloud top detection to describe the spatio-temporal extent of potential hail events. Hail footprints were generated that could be used by the insurance industry for risk analyses.

The revised paper shows where revision was done and the authors took care to answer the questions from the first review. The suggestions made by the reviewer are minor and will help with the clarification of a few minor points.

Below are some questions and suggested corrections:

1. Page 1: "to estimate risk for the insurance sector (Punge et al., 2014; Radler et al.)." The date for the reference is missing. Please double-check check rest of the references for a similar problem.
2. Page 13-15

The description here is a bit fuzzy for me. I understand the differences in groupings of grid sizes for hail events and for time. However, the manner in which it is described is not clear for me.

For the number of events, the $0.3 \times 0.5°$ is combined into new grids of $3 \times 5°$ by grouping $10 \times 10$ of the smaller grid cells (I assume the $0.3 \times 0.5°$ grid cells) together. Then days are drawn from a newly formed $3 \times 5°$ grid and the surrounding 8 boxes. What are the sizes of these boxes ($0.3 \times 0.5°$ or $3 \times 5°$)?

Are these 8 boxes chosen in any particular manner?

The switch between the terminology of "grid", "domain" and "boxes" becomes confusing.

For time, is the centre point taken in the newly formed $3 \times 5°$ grid and then taking a new grid over the centre point of the size $10 \times 6°$

3. **Section 3.1**

Event lengths and widths are approximated with GEV over the exponential distribution due to the better fit. Its stated that the GEV fits well over the bottom tail of the distribution. But it does not fit well for low widths that are over-represented (over-estimated) and it does not fit well in the upper tails in that it gives unrealistic large values.

a. Which GEV function was used?
b. From the description, it is my understanding that the good fit of GEV is only for the bottom tail of the distribution for length of the storm. And not really anywhere else on the distributions? Should a different GEV function be applied to get a better fit? What statistical tests were performed to see the goodness-of-fit of the tested distributions?

    c.   How was the value 1.5xlargest observed value chosen as the point/place where to truncate the events? It appears this was done for the whole country – but what this checked to hold true for the whole country?

    d.   Figures 11 a and b and g are not discussed in the text

**4. Section 3.2**

The authors state in this section that due to the large uncertainty the hail size estimated were not considered for the modelling approach using geostationary satellite measurements alone. And that a severity index was created as a substitute.

The section, however, does not elaborate on how the severity index was set up in terms of the range of the scale. Or some descriptives on how this scale looks like or work in terms of the available data for South Africa. It is discussed throughout the rest of the paper but it does leave this section feeling unfinished.

Section 3.3 in terms discusses how the hail size can be calculated from the stochastic modelling using data from the ESWD and Severe Storms Archive. It makes the assumption that the largest hail size distributions over the continent will the same for South Africa. The authors can discuss the level of uncertainty (although not modelled explicitly) that this assumption can bring into the modelling process.

**5. Section 3.5**

- Page 21: "Even if there is a strong correlation between all regions, smaller regions tend to experience relatively higher variability"
  What is the definition of the regions in this context? And what is considered the larger vs the smaller regions?

- Page 24: The comparison between the modelled number of hail days for Gauteng (26 days) against that of Smith et al of 69 seems like a big difference. A description follows from how the results from table 3 can change when events larger than different event sizes are viewed. But it is not related back to the 69 events of Smith et al. From Table 3 it is not clear for what events sizes (>= to what cm size) the days are valid.

**6. Section 4.1**

Bottom page 24 discusses applying frequency-weights for Figure 17b and c but not which frequency weights are used and where it was obtained from.

- Figures 17, 18 – it is not clear if the number of events referred to are that of the observed geostationary data or from the modelled dataset

**7. Section 4.2**

- Page 27: "while Fig 18d presents the same occurrence for maximum hail severity indicator greater than 2." Should this be 2cm?
- "We lso note that the local hail count per year is about 2 in KwaZulu-Natal maximum and ` in the Highveld and Gauteng region so…"
- Does this sentence refer to the number of hail events per year, hail events per year over a certain hail size/severity index? This seems like a very low value per year.
- Bottom of page 27:
- From line 465 – the event sized discussed – are these the maximum event sizes expected or the average event sizes expected per 10 year period? Where are these values compared with actually observed hail sizes as seen from newspaper/twitter reports?

Publish with minor revision that the editor can check.

---

## Referee Report (RR2)

**NHESS-2021-342: Characteristics of hail hazard in South Africa based on satellite detection of convective storms**

The authors clearly expended a lot of effort addressing many of my and the other reviewers' concerns. I appreciate their attention to detail. The new phrasing makes it clear an explicit OT temperatures – hail size relationship is not being established herein, which was my main concern in my previous review. Added text, figures, and appendices help clarify the process and provides verification context. I still have some concerns remaining, mainly requests for additional clarification and requests for reader cautioning, but they are not insurmountable to address.

**Note**: all line numbers herein refer to the author tracked changes document.

**Major comments:**

My major comments fall into two categories: requests for additional clarification, and points to make to caution the future reader.

Additional clarification:

- Lines 187- 202: What time ERA5 file is used for the insurance claims, considering the insurance data doesn't have a time of occurrence? Give the odd distribution of parameters for the claims data in Figs. 5a, 5b, it would seem a possibly unrepresentative time was chosen.

- Lines 225-228: This additional explanation helps, thank you. That being said, Fig. 7 is still not very clear. Why not separate it into ~3 subfigures over different time intervals, so the three events can be separately shown?

- Lines 241 - 250: I appreciate the added text, but a bit more clarification is still required. Line 243 introduces the phrase "potential hail events", an excellent addition, but further sentences don't use the phase. "Historic events" here could refer to historic hail detections from TRMM/GPM, S. African claims data, or the Australian/European reports.

  I understand that "potential historic hail events" is a mouthful to repeat multiple times; possibly "historic OT events" could be used instead. (And is used, in subsequent sections.)

- Lines 261-264: A Gaussian distribution is fit to each 3° by 5° box, correct? I'm having trouble following what the phrases in parentheses ("mean number of events " and "summer peak) mean.

- Lines 266-270: While I appreciate the extra text, I'm still having trouble following this explanation. I understand a random drawing of the day of year occurrence of N events in a grid box, and from each of the surrounding boxes. But are not all of these events retained? What are "blocks of N ⅓"? Most importantly, what N is chosen (and why?)

- Line 276: No blocks of N⅓ here?

- Lines 316- 317: While I appreciate the addition, the application of the nomenclature to this specific case is still a bit murky. What does the objective function predict? The sample data probabilities of.... a specific length and width?

- Lines 349- 375: While this section is clearer, some additional clarification can be provided. State up front at line 361 what statistical correlation relationships are preserved both in the historical dataset and the model. From Punge et al (2014, P14 hereafter) it seems there is a first step that moves from correlations between length-width-hail size to length-width-OT temp difference before the historical dataset can be constructed, is that correct?

  This explanation will also keep the reader from jumping to conclusions that some sort of OT temp difference and hail size scaling equation exists.

  A very quick recap (or reference to specific section of P14) for how track area is determined would also be helpful.

  I appreciate the authors uploading P14 to Researchgate so I (and future readers) can review it for answers to these questions.

  Finally, why are some of the correlations so different between the historical and modeled length/width and event to storm area ratio? Are these values still within the realm of reasonability?

Cautioning the reader:

- Lines 177-180: This phrasing makes it sound like the extensive calibration of the OT detection algorithm has been for improving its severe weather detection capabilities, but in my opinion has actually been for improving OT detection compared to human ID. The studies cited here found OTs to correlated with severe weather but were not explicitly looking at hail. Punge et al. (2014, 2017) and Jurkovic et al (2015) would be better hail-OT connection citations. It should also be acknowledged here that these sources found OTs near only about 50% of hail events, and Bedka et al (2018) noted that large percentages of OTs do not produce hail. I understand use of OTs is the best option the authors have, but all appropriate caveats need to be acknowledged up front.

- Appendix A: Fantastic addition to the article. It addresses many of the concerns I noted earlier about the relationship between MESH and OT probability.

  I would ask the authors to provide a few cautioning statements for the reader. Fig. A1, while convincingly establishing a link between increased MESH-estimated hail size and increased OT probability, does not establish a relationship with observed hail size at the ground. MESH is not observing hail fall but is instead a proxy for hailfall based on a storm's ability to loft condensate - essentially, updraft strength, much like OT temperature difference. The relationship between a storm's updraft strength and hail size produced at the surface is not linear, and at larger updraft speeds may in fact be inversely proportional (see Fig. 6 of Lin and Kumjian 2022). Readers should be cautioned against assuming Fig. A1 implies a similar connection with observed surface hail size.

- Lines 329- 340: While I appreciate the change in some of the phraseology, the text here still is connecting increased updraft speed with the ability to produce larger hail. While this could be true for smaller hail and/or weaker updrafts, this relationship doesn't hold for stronger updrafts, as Lin and Kumjian (2022) makes clear. (Marion et al. was about tornadoes so is not relevant here.)

  Please caution the reader that updraft strength has been shown to not be directly related with increases in hail size, and for stronger updrafts in particular the relationship potentially reverses. However, given the lack of other available data sauces, OT temperature difference here will be used as an estimate of storm severity, and will be connected to hail size via the reports databases., etc. etc. (I would avoid the term "updraft intensity", as it isn't clear if it means strength, area, or both.)

**Minor comments:**
- Line 10: Damage is not limited to large hail. Large quantities of small hail can be equally problematic, as can almost any size of windblown hail.
- Line 25: Cf is used for comparison, but only one figure is listed - perhaps e.g. was meant instead? Also note the reference is to their figure.
- Line 25-26: What methods did Smith et al. use to derive their frequency estimate?
- Lines 28- 32: Oddly phrased. What problems did Grieser and Hill (2019) face that leads the authors to conclude that hail pad and hail report data aren't sufficient? I'm assuming the difference is quantity of data in South Africa vs. the U.S., but phrasing could be improved.
- Lines 45- 46: Again, an odd transition. Based on _just_ these sentences, a radar data climatology In S. Africa seems possible. Perhaps adding "but is not available over other large portions of the country" at the end of these sentences.
- Line 47: "for hail" → "for hail detection"
- Line 57-58: would rephrase to"... an appropriate proxy to assess individual severe convective storms (SCSs) and large-scale outbreaks for the potential of hail production. Large-scale hail-producing outbreaks can cause by far..."
- Line 99, Fig. 2b: Determining where the green colors start in Fig. 2b is difficult. Adding a black outline showing anvil detection would be helpful. Also, adding a sentence pointing out the Great Escarpment and the Drakensberg in the topography map would be useful for later references in the text.
- Lines 121- 122: I'd keep the mention of the Sandmæl algorithm but note that South Africa is not continuously covered by visible imagery.
- Line 124: " ... scanned for hailstorms..." → " .... scanned for OTs..."
- Line 72: Typically, 20% is used as a threshold probability in hail detection (e.g., Bang and Cecil 2019, 2021). Why the change here?
- Lines 145- 146: Prein and Holland (2018) focused on comparison of the distribution of hail environments across the globe and hail detection, not hail size

spectra (and they weren't particularly successful in global application.) What publications have focused specifically on *observed* hail size distributions across the globe?

Even if the answer is none, I think arguing from scarcity is reasonable enough, it just should be presented with the necessary caveats.

- Lines 147- 149: Any reason not to include reports from the US? With inclusion of MPing and COCORAHS sources, hail smaller than 2 cm could be included in the spectra calculation.

- Lines 182-184: Is the filter based on environments associated with the insurance claims? These sentences and lines 79-80 make it seem like that is the case, but such connections aren't described in this section.

Because Punge et al. (2017) is behind a paywall (and Bedka et al (2018) mainly just cites Punge et al.) please provide a brief recap here.

- Lines 203-205, Figs. 5c-e: Adding thicker black lines where the filter threshold were chosen would be helpful.

- Figs. 3,4, 6: I'd prefer grouping these figures all in one figure, to allow for easier, direct comparison.

- Lines 538-539: Which criteria are used in the method described herein, ECS or OT? If OT, why, given that it seems like it misses a lot of hail - producing storms? (Perhaps because of false alarms, which could be indicated in another table column in Table A1.)

- Lines 219-220: What's the temporal resolution of the MSG data used for OT detection? That fact should probably be included in section 2.1.

- Lines 238-239: Probably should note a filter for these erroneous groupings is being developed, in future work. 🙂

- Line 241: What model?

- Line 251: frequency of filtered OTs, correct?

- Line 278: "observed and modeled.... OT events"

- Fig 10: These distributions are for the entire domain shown in Fig. 9, correct? Please note in caption.

- Lines 292-293: But both the "historic" and "modeled" events set here are of (filtered) OT detections, correct? So the chance of missing hail reports on the ground is, in this specific context, irrelevant.

- Line 305: "covered by OTs/hail streaks" → covered by individual OTs"

- Line 306: While I understand your meaning here, since this ratio has an inherent upper bound of 1 the phrasing could be better.

- Line 307: Isn't this product f²?

- Line 308: " > 105 km², *not shown*"

- Line 328: "Storm's severity" → "tornadic intensity". "Storm severity" is too nebulous a term, since it could be interpreted as meaning "updraft strength", which is not necessarily correlated with severe impacts on the ground.

- Lines 381- 384: Why was this figure removed? A comparison of the frequency of occurrence of hail events (not including size) across the country in the historical vs. stochastic datasets seems of prime importance. If the figure is not retained, then discussion about it should be eliminated.

- Section 3.6: Much improved, and an interesting result when compared to the Smith et al. study. If I am reading correctly, the stochastic event set underestimates the occurrence of hail days in the region, but potentially overestimates severe hail. It's possible the break down in the updraft strength - large hail relationship is causing these large biased "severe" hail numbers (also possible large hail is underreported, as you note.) Any idea what could be causing the overestimation of hail in general?

- Lines 505- 507: While I agree with these statements, I would shift them earlier in the conclusions as they are awkwardly placed here.

- Fig. B1: Great addition. Can these plots be normalized by total number of detections and plotted in the same plot for easier comparison?

- Lines 570- 580: Excellent addition. I'd rearrange the text (or figure) so the subfigures are referenced in order.

**Grammatical:**
- Line 47: "this" → "these"
- Line 131:" ... east of..." → " .. in east..."
- Line 206: Add "Fig. 6" after Fig. 3, so the comparison has an object.
- Line 510: "95." → "95$^{th}$"
- line 221: "of the event, *or grouped*, OTs"
- Line 267: "of" → "from"
- Fig 11: years → years'
- Line 417: "hail hazard" → "the hail hazard"

---

## Author Response (AR2)

**Review "Characteristics of hail hazard in South Africa based on satellite detection of convection storms" by Punge et al**

**Reviewer 1**

The authors thank the reviewer again for the comments and suggestions. All points raised by the reviewer have been taken into account in the revised version of the manuscript. Our answers below are shown in blue.

**Summary**
This article is the second revision of the paper with the same title. In this paper, the authors investigate the formation of hail events and hail hazard in South Africa as guided by 14 years of geostationary satellite observations storms. A multivariate stochastic model was built simulating event properties spanning 25 000 years of hail occurrence using overshooting cloud top detection to describe the spatio-temporal extent of potential hail events. Hail footprints were generated that could be used by the insurance industry for risk analyses.

The revised paper shows where revision was done and the authors took care to answer the questions from the first review. The suggestions made by the reviewer are minor and will help with the clarification of a few minor points.

Below are some questions and suggested corrections:

**1. Page 1**
"to estimate risk for the insurance sector (Punge et al., 2014; Radler et al.)." The date for the reference is missing. Please double-check check rest of the references for a similar problem.
An entry was missing from the bibtex file. We checked all the references but found only this incorrect entry.

**2. Page 13-15**
The description here is a bit fuzzy for me. I understand the differences in groupings of grid sizes for hail events and for time. However, the manner in which it is described is not clear for me.

For the number of events, the 0.3°x0.5° is combined into new grids of 3°x5° by grouping 10x10 of the smaller grid cells (I assume the 0.3°x0.5° grid cells) together. Then days are drawn from a newly formed 3°x5° grid and the surrounding 8 boxes. What are the sizes of these boxes (0.3°x0.5° or 3°x5°)?
The surrounding boxes are also the large boxes of 3°x5°. This is included in the text.

Are these 8 boxes chosen in any particular manner?
Yes, as explained below, with this method we obtain regions that "represent the scale of synoptic processes and flow patterns governing the spatial (and temporal) clustering of SCS, for example, by specific weather regimes such as Baltic blocking". To avoid confusion, we changed the word "The process" to "The spatial smoothing technique described above"

The switch between the terminology of "grid", "domain" and "boxes" becomes confusing.

Right. Domain in the revised version only refers to the entire study domain in South Africa; we changed that here and in the entire manuscript. Boxes mean grid cell; we changed accordingly.

For time, is the centre point taken in the newly formed 3°x5° grid and then taking a new grid over the centre point of the size 10°x6°
Correct; we have slightly reworded this sentence for clarity.

**3. Section 3.1**

Event lengths and widths are approximated with GEV over the exponential distribution due to the better fit. Its stated that the GEV fits well over the bottom tail of the distribution. But it does not fit well for low widths that are over-represented (over-estimated) and it does not fit well in the upper tails in that it gives unrealistic large values.
    a.   Which GEV function was used?
        Table 1 shows all three parameters of the GEV; for all three parameters, $\kappa > 0$, thus the GEV is the Frechet (Fisher-Tippett I) distribution; we included that in the text.
    b.   From the description, it is my understanding that the good fit of GEV is only for the bottom tail of the distribution for length of the storm. And not really anywhere else on the distributions? Should a different GEV function be applied to get a better fit? What statistical tests were performed to see the goodness-of-fit of the tested distributions?
        This cannot be said. As shown in Figure 11 (Figure 11c for event length), most parameters are reproduced very reliably by the model using the GEV across all dimensions. In almost all cases, the model results are within the error bars of the historical event set - particularly for length and width, which are the most important for damage. We added a comment on this.
    c.   How was the value 1.5 x largest observed value chosen as the point/place where to truncate the events? It appears this was done for the whole country – but what this checked to hold true for the whole country?
        As written in the manuscript, "length and width were truncated at 1.5 times the largest observed values at which events effectively cover the entire country (1 445 km x 677 km)". This was checked for the whole country.
    d.   Figures 11 a and b and g are not discussed in the text
        We included a brief discussion of all Figures from the panel, including a, b, and g.

**4. Section 3.2**

The authors state in this section that due to the large uncertainty the hail size estimated were not considered for the modelling approach using geostationary satellite measurements alone. And that a severity index was created as a substitute.
Yes, this was the major point of the 2[nd] reviewer to exclude hail size estimation from the satellite data, which we followed in previously revised manuscript (see the large deleted parts).

The section, however, does not elaborate on how the severity index was set up in terms of the range of the scale. Or some descriptives on how this scale looks like or work in terms of

the available data for South Africa. It is discussed throughout the rest of the paper but it does leave this section feeling unfinished.

The severity index is largely discussed in the Appendix. Both the reliability of the methods and the uncertainty are assessed using the Maximum Expected Size of Hail (MESH). We think this discussion is sufficient, also since the severity index is not central in the paper and the risk model.

Section 3.3 in terms discusses how the hail size can be calculated from the stochastic modelling using data from the ESWD and Severe Storms Archive. It makes the assumption that the largest hail size distributions over the continent will the same for South Africa. The authors can discuss the level of uncertainty (although not modelled explicitly) that this assumption can bring into the modelling process.

It's very difficult to estimate the resulting uncertainty in the model. Nor does a discussion of what we do not know really help the reader. However, we have included a statement on this, explaining that the resulting uncertainty can be reduced by calibrating the model using past loss events. By the way, cat models usually require calibration.

**5. Section 3.5**

- Page 21: "Even if there is a strong correlation between all regions, smaller regions tend to experience relatively higher variability"
  What is the definition of the regions in this context? And what is considered the larger vs the smaller regions?
  This statement refers to the regions shown in Fig. 14, i.e. KwaZulu-Natal (KZN), Highveld (HVD) and Gauteng (GAU) (added in the manuscript). We have not systematically examined event variability for smaller regions (smaller than these three); however, due to the short temporal and small spatial scales of hail events, variability is generally greater for small areas compared to larger areas (also included a brief explanation).

- Page 24: The comparison between the modelled number of hail days for Gauteng (26 days) against that of Smith et al of 69 seems like a big difference. A description follows from how the results from table 3 can change when events larger than different event sizes are viewed. But it is not related back to the 69 events of Smith et al. From Table 3 it is not clear for what events sizes (>= to what cm size) the days are valid.
  Unfortunately, Smith et al. (1998) do not provide information on the number of events within the three different categories. However, assuming an exponential hail size distribution, it can be assumed that events in the lowest class (3-10 mm, which includes sleet/graupel) dominate the statistics. Small diameter events are not included in our stochastic event set as they are not relevant to damage. We have added a few lines on this to explain the discrepancies. This insertion hopefully makes the transition to severe hail days clearer.

**6. Section 4.1**

- Bottom page 24 discusses applying frequency-weights for Figure 17b and c but not which frequency weights are used and where it was obtained from.

We added the frequency weights and an explanation for that: "These frequency weights are hence the inverse of the retained fraction per class, i.e. 40, 13.3, 20, 10, 1 for classes 1 to 5."

- Figures 17, 18 – it is not clear if the number of events referred to are that of the observed geostationary data or from the modelled dataset
  All Figures show the results of the stochastically generated event set. To make this clear, we changed the heading into "Modeled event footprints"

**7. Section 4.2**

- Page 27: "while Fig 18d presents the same occurrence for maximum hail severity indicator greater than 2." Should this be 2cm?
  Corrected
- "We also note that the local hail count per year is about 2 in KwaZulu-Natal maximum and 1 in the Highveld and Gauteng region so…"
  Does this sentence refer to the number of hail events per year, hail events per year over a certain hail size/severity index? This seems like a very low value per year.
  This applies to the frequency-weighed hail count per year; we included that in the text.
- Bottom of page 27: From line 465 – the event sized discussed – are these the maximum event sizes expected or the average event sizes expected per 10 year period? Where are these values compared with actually observed hail sizes as seen from newspaper/twitter reports?
  The figure shows the maximum hail size occurring once in 10 years (word maximum included in the text). We are not aware of any study or report that can be used for comparison (i.e. relating frequency to diameter). However, based on the few reports available, our estimates appear to be reliable.

Publish with minor revision that the editor can check.

Reviewer 2

The authors thank the reviewer again for his/her extensive work and the additional comments and suggestions. All points raised by the reviewer have been taken into account in the revised version of the manuscript. Our answers below are shown in blue.

The authors clearly expended a lot of effort addressing many of my and the other reviewers' concerns. I appreciate their attention to detail. The new phrasing makes it clear an explicit OT temperatures – hail size relationship is not being established herein, which was my main concern in my previous review. Added text, figures, and appendices help clarify the process and provides verification context. I still have some concerns remaining, mainly requests for additional clarification and requests for reader cautioning, but they are not insurmountable to address.

**Note**: all line numbers herein refer to the author tracked changes document.

**Major comments:**

My major comments fall into two categories: requests for additional clarification, and points to make to caution the future reader.

**Additional clarification:**

- Lines 187- 202: What time ERA5 file is used for the insurance claims, considering the insurance data doesn't have a time of occurrence? Give the odd distribution of parameters for the claims data in Figs. 5a, 5b, it would seem a possibly unrepresentative time was chosen.
  ERA5 were chosen at 12 UTC. While CAPE is dependent on the time of day, sensitivity tests have shown that shear and melt level are not very sensitive. The somewhat odd distributions are due to heavily population biased sampling locations. We have included an explanation in the text.

- Lines 225-228: This additional explanation helps, thank you. That being said, Fig. 7 is still not very clear. Why not separate it into ~3 subfigures over different time intervals, so the three events can be separately shown?
  As the insurance claims have no time, they cannot be separated and aligned to the different OTs/events. In the event definition and also in the stochastic event set, we do not model single streaks, but several streaks on a given day. Separating Fig. 7 into the three different events would imply that each streak is stochastically modelled and not the number of streaks per day.

- Lines 241 - 250: I appreciate the added text, but a bit more clarification is still required. Line 243 introduces the phrase "potential hail events", an excellent addition, but further sentences don't use the phase. "Historic events" here could refer to historic hail detections from TRMM/GPM, S. African claims data, or the Australian/European reports.
  We went through the whole manuscript and changed OT events to "potential hail events" when referring to the historical event set. Btw, the word "OT detection" is now used only when referring to single OTs without event clustering or stochastic modeling.

- I understand that "potential historic hail events" is a mouthful to repeat multiple times; possibly "historic OT events" could be used instead. (And is used, in subsequent sections.)
  In that formulation, "historic" is not required (see explanation above).

- Lines 261-264: A Gaussian distribution is fit to each 3° by 5° box, correct? I'm having trouble following what the phrases in parentheses ("mean number of events " and "summer peak) mean.

  We reformulated the sentence "Depending on the mean number of OT events at a certain location…" and deleted the summer peak

- Lines 266-270: While I appreciate the extra text, I'm still having trouble following this explanation. I understand a random drawing of the day of year occurrence of N events in a grid box, and from each of the surrounding boxes. But are not all of these events retained? What are "blocks of N ⅓"? Most importantly, what N is chosen (and why?)

  N is the grid cell's number of events in the 250 year batch derived from the modelled event frequency. Indeed, only 1 out of 9 events is retained. The exponent (N^(1/3) is tuned to fit observations. 1 would have all events on one day, 0 would retain the original date for each event. We have included an explanation in the text.

- Line 276: No blocks of N⅓ here?

  No, because events don't tend to cluster on hours as much as on days, and sample sizes per day would be quite small per day for reasonable tuning.

- Lines 316- 317: While I appreciate the addition, the application of the nomenclature to this specific case is still a bit murky. What does the objective function predict? The sample data probabilities of.... a specific length and width?

  The objective function is -ln of the product of the probabilities of each sample length given the assumed (GEV) distribution of lengths. Then the distribution parameters were varied to find the maximum of the objective function. We replaced this extended explanation with the former one.

- Lines 349- 375: While this section is clearer, some additional clarification can be provided. State up front at line 361 what statistical correlation relationships are preserved both in the historical dataset and the model. From Punge et al (2014, P14 hereafter) it seems there is a first step that moves from correlations between length-width-hail size to length-width-OT temp difference before the historical dataset can be constructed, is that correct? This explanation will also keep the reader from jumping to conclusions that some sort of OT temp difference and hail size scaling equation exists.

  The correlations refer to all four parameters length, width, area, and severity; correlatios are calculated for each combination (i.e., 16 pairs). We reformulated the sentence accordingly.

- A very quick recap (or reference to specific section of P14) for how track area is determined would also be helpful.

  We added a recap that this refers to the ellipse area determined by length and width and included a reference to Sect. 3.1, where also the formula is given.

- I appreciate the authors uploading P14 to Researchgate so I (and future readers) can review it for answers to these questions.

  You're welcome

- Finally, why are some of the correlations so different between the historical and modeled length/width and event to storm area ratio? Are these values still within the realm of reasonability?

  Larger differences are more or less limited for larger event lengths. However, the number of those events decreases almost exponentially. Thus, small differences have a larger impact on the results. For event lengths below about 200 km, the agreement between historical and modelled events is good, and this is the case for the majority of events (note the logarithmic scale of the color bar).

**Cautioning the reader:**

- Lines 177-180: This phrasing makes it sound like the extensive calibration of the OT detection algorithm has been for improving its severe weather detection capabilities, but in my opinion has actually been for improving OT detection compared to human ID. The studies cited here found OTs to correlated with severe weather but were not explicitly looking at hail. Punge et al. (2014, 2017) and Jurkovic et al (2015) would be better hail-OT connection citations. It should also be acknowledged here that these sources found OTs near only about 50% of hail events, and Bedka et al (2018) noted that large percentages of OTs do not produce hail. I understand use of OTs is the best option the authors have, but all appropriate caveats need to be acknowledged up front.

  We agree that this issue has to be discussed in more detail. However, as the reviewer also mentioned, it need to be acknowledged up front. Therefore, we extended the discussion of the caveats of OT detections in the introduction and deleted the few sentences in Sect. 2.5. "Still, in some cases, OT features may have been misdetected or may not have produced hail on the ground, for example, due to melting of hailstones during fall through a deep column of warm air. This is acknowledged in the studies of Punge et al. (2014, 2017) and by Bedka et al. (2018) who noted the large percentages of OTs without hail on the ground. However, in addition to the hazard modeling purpose, the focus of our study is on the identification of larger spatial SCS clusters with an increased potential of hail production during the lifetime of the event, rather than detecting each individual storm with enhanced hail potential. These large-scale hail-producing outbreaks can cause by far the largest part of the damage registered by insurers, and can induce solvency issues when the risk was not properly estimated."

- Appendix A: Fantastic addition to the article. It addresses many of the concerns I noted earlier about the relationship between MESH and OT probability. I would ask the authors to provide a few cautioning statements for the reader. Fig. A1, while convincingly establishing a link between increased MESH-estimated hail size and increased OT probability, does not establish a relationship with observed hail size at the ground. MESH is not observing hail fall but is instead a proxy for hailfall based on a storm's ability to loft condensate - essentially, updraft strength, much like OT temperature difference. The relationship between a storm's updraft strength and hail size produced at the surface is not linear, and at larger updraft speeds may in fact be inversely proportional (see Fig. 6 of Lin and Kumjian 2022). Readers should be cautioned against assuming Fig. A1 implies a similar connection with observed surface hail size.

  An upcoming provisionally-accepted paper (Minor revisions, Scarino et al. Artificial Intelligence for the Earth Systems, 2023) expands upon the analyses presented in this section, and shows high uncertainty associated with hail size reports on the ground over the U.S. This is a known issue in the hail research world, was it a shilling, or nickel, "hen egg", "teacup", "golfball", "cricket ball", "softball", etc.. size, and what do these objects correlate to in true physical dimension? The Scarino et al study shows is almost no correlation with any NWP or satellite parameter with observed hail size, whereas there is notable correlation of many of these parameters with MESH. The attached graphic shows this general lack of correlation between these parameters and reported hail size (SPC Hail Size in graphic). The lack of correlation of satellite parameters with observed size is also reflected in the results of Murillo and Homeyer (JAMC, 2019).

[Figure]

2007-2017 GOES-13 + GOES-12 Parameters Binned by MESH95 Hail Size (cm)

We appreciate your concern about a reader inferring that our analysis might suggest that there is a linear relationship between satellite-inferred intensity and hail size.

To address this, we have included the following text in Appendix A: "Therefore, IR-anvil BT difference is a suitable parameter, independent of any reliance on a numerical model, for purposes of modeling the expected hail severity at the ground. Though these results suggest a quasi-linear relationship between MESH and satellite-derived updraft intensity proxies, the true relationship between such proxies and hail size encountered on the ground is unknown, primarily due to known uncertainties with hail size reporting."

We also updated Figure A1 with the one shown above.

- Lines 329- 340: While I appreciate the change in some of the phraseology, the text here still is connecting increased updraft speed with the ability to produce larger hail. While this could be true for smaller hail and/or weaker updrafts, this relationship doesn't hold for stronger updrafts, as Lin and Kumjian (2022) makes clear. (Marion et al. was about tornadoes so is not relevant here.)

  Please caution the reader that updraft strength has been shown to not be directly related with increases in hail size, and for stronger updrafts in particular the relationship potentially reverses. However, given the lack of other available data sauces, OT temperature difference here will be used as an estimate of storm severity, and will be connected to hail size via the reports databases., etc. etc. (I would avoid the term "updraft intensity", as it isn't clear if it means strength, area, or both.)

  As suggested, we modified the statement of the Marion et al. (2019) paper and omitted the term updraft by relying on temperature difference solely. However, the reviewer has to be aware that the intensity measure does not enter our stochastic risk model.

**Minor comments:**

- Line 10: Damage is not limited to large hail. Large quantities of small hail can be equally problematic, as can almost any size of windblown hail.

  That's right. But we refer in this sentence not to damage at all, but to the significant contribution to natural hazards. In the very detailed insurance loss data we got for a specific

region we see that small hail only cause light damage (< 1% of the insurance losses). Therefore, it makes sense to refer here to large hail.

- Line 25: Cf is used for comparison, but only one figure is listed - perhaps e.g. was meant instead? Also note the reference is to their figure.
  We changed into "see"

- Line 25-26: What methods did Smith et al. use to derive their frequency estimate?
  Same methods and data as Amirat et al. (1985). We included that here, but also give some more details about the reports in the previous sentence.

- Lines 28- 32: Oddly phrased. What problems did Grieser and Hill (2019) face that leads the authors to conclude that hail pad and hail report data aren't sufficient? I'm assuming the difference is quantity of data in South Africa vs. the U.S., but phrasing could be improved.
  Hailpad data and hail reports are always insufficient to quantify hail risk for insurance purposes, because that requires to estimate the damage of a 1 in 200 year event (therefore requires stochastic modeling as later explained). We completely reformulated the two sentence and moved it to the previous paragraph because it does not refer to South Africa.
  *"Grieser and Hill (2019) used volunteer-collected hail observations in the United States to model the rate of hailstones hitting the ground per unit area, time, and hailstone size bin during the passage of a hailstorm. Based on that data, they set up a model to calculate the vulnerability of subjects at risk as a function of the diameter of the largest hailstone, which can be transferred to other regions."*

- Lines 45- 46: Again, an odd transition. Based on just these sentences, a radar data climatology In S. Africa seems possible. Perhaps adding "but is not available over other large portions of the country" at the end of these sentences.
  To improve the transition between the two phrases, we included further – and no unimportant – information: "*However, the South African radar network does not cover the entire country.*"

- Line 47: "for hail" → "for hail detection"
  We changed into *"proxy for hailstorm detection"*

- Line 57-58: would rephrase to"... an appropriate proxy to assess individual severe convective storms (SCSs) and large-scale outbreaks for the potential of hail production. Large-scale hail-producing outbreaks can cause by far..."
  Changed as suggested

- Line 99, Fig. 2b: Determining where the green colors start in Fig. 2b is difficult. Adding a black outline showing anvil detection would be helpful. Also, adding a sentence pointing out the Great Escarpment and the Drakensberg in the topography map would be useful for later references in the text.
  Very good point. We even thought that it might be helpful for the reader to add a short subsection (now new subsection 2.1) where both the climate and the specific topographic situation of South Africa is described. Here we also explain the Great Escarpment and the Drakensberg.
  Fig. 2b is just a snapshot as an example of an SCS and the spatial extent; we think that details are unimportant for the reader.

- Lines 121- 122: I'd keep the mention of the Sandmæl algorithm but note that South Africa is not continuously covered by visible imagery.
  The point here is that the new Khlopenkov et al. (2021) algorithm goes beyond the Sandmæl algorithm. For this reason and not because of the problem with visible imagery we deleted this reference here.

- Line 124: " ... scanned for hailstorms..." → " .... scanned for OTs..."
  changed

- Line 72: Typically, 20% is used as a threshold probability in hail detection (e.g., Bang and Cecil 2019, 2021). Why the change here?
  We assume this comment refers to line 172 and not to 72 (Introduction). The cited studies were designed to represent significant hail (and threshold tuned to match radar MESH), but our OT methodology also covers smaller hail diameters. We included a statement on that around line 172.
- Lines 145- 146: Prein and Holland (2018) focused on comparison of the distribution of hail environments across the globe and hail detection, not hail size spectra (and they weren't particularly successful in global application.) What publications have focused specifically on observed hail size distributions across the globe?
  Even if the answer is none, I think arguing from scarcity is reasonable enough, it just should be presented with the necessary caveats.
  We added some references that estimated hail size spectra in different regions / continents and deleted the Prein and Holland (2018) reference.
- Lines 147- 149: Any reason not to include reports from the US? With inclusion of MPing and COCORAHS sources, hail smaller than 2 cm could be included in the spectra calculation.
  There is no particular reason why we did not include the data from the USA. However, since our sample already includes about 30,000 reports, we believe that including them further does not really change the results - especially since we only used the data for stochastic hail modeling.
- Lines 182-184: Is the filter based on environments associated with the insurance claims? These sentences and lines 79-80 make it seem like that is the case, but such connections aren't described in this section.
  Yes, we used insurance claims but also microwave detections. We added that to the sentence.
  Because Punge et al. (2017) is behind a paywall (and Bedka et al (2018) mainly just cites Punge et al.) please provide a brief recap here.
  As suggested, we included a brief recap of the principals of the filter algorithm.
- Lines 203-205, Figs. 5c-e: Adding thicker black lines where the filter threshold were chosen would be helpful.
  Added
- Figs. 3,4, 6: I'd prefer grouping these figures all in one figure, to allow for easier, direct comparison.
  We have combined Figs. 3 and 6 into one panel, but not Fig. 4, so as not to give the impression that microwave detection could be used to derive hail frequency.
- Lines 538-539: Which criteria are used in the method described herein, ECS or OT? If OT, why, given that it seems like it misses a lot of hail - producing storms? (Perhaps because of false alarms, which could be indicated in another table column in Table A1.)
  The purpose of this analysis is to demonstrate how detectable radar-observed hail cores can be with satellite cloud top signals. Cooney et al. (JGR, 2021) shows that there can be many ECS' in infrared (IR) imagery around a true updraft core. This is routinely evident in every severe thunderstorm, where an updraft core ejects cold outflow that could appear like a weak OT feature in the absence of other spatial context. The OT detection algorithm has been validated with human OT identifications and OT detections using precip radar echo top by Cooney et al and Khlopenkov et al (JGR, 2021). While some weak OT features can be missed, or a weak cold spot can be mis-interpreted to be a true OT updraft core, we feel that OT detections using the OT probability >= 0.5 is our most reliable way for depicting cell tracks that are necessary for the South Africa hail modeling described in this paper. You are correct in noting that we more frequently have ECS detections near MESH events than OT

detections. What this says is that some typically weaker hailstorms can have very little temperature perturbations in their tops. Using ECS's as a proxy for hailstorms though would result in many anvil regions being identified that are not truly updrafts which would be adverse to our model development. It is unclear to us how a "false alarm" would be characterized in Table A1; would any OT that doesn't have a severe MESH value be considered a false alarm?  That would not be very informative because it is well known that many thunderstorms throughout the world routinely produce OTs but not hail.  This is why we begin with what we feel are reliable OT updraft core detections, then filter detections to eliminate updrafts in environments not at all supportive of hail, then spatially cluster updrafts to form storm tracks, and then use the satellite-derived storm intensity as a proxy for maximum hail size for our modeling purposes.

- Lines 219-220: What's the temporal resolution of the MSG data used for OT detection? That fact should probably be included in section 2.1.
  Temporal resolution is 15 min. This was already mentioned in Sect. 2.1 (former line 124-125).

- Lines 238-239: Probably should note a filter for these erroneous groupings is being developed, in future work. ☺
  I refrain from mentioning such a filter as this is not really a major issue. However, it's simple to exclude such event types. We have included a short sentence on that.

- Line 241: What model?
  Hail hazard model included

- Line 251: frequency of filtered OTs, correct?
  right, included

- Line 278: "observed and modeled.... OT events"
  No, here we considered events, i.e., grouped OTs. To avoid any confusion, we made this clear at the beginning of Sect. 4 by including: *"This section refers to events of grouped OTs according to the event definition in Sect. 2.4"*

- Fig 10: These distributions are for the entire domain shown in Fig. 9, correct? Please note in caption.
  Correct and included

- Lines 292-293: But both the "historic" and "modeled" events set here are of (filtered) OT detections, correct? So the chance of missing hail reports on the ground is, in this specific context, irrelevant.
  Yes, all analyses and modeling tasks are based on the filtered OTs. To avoid any conclusions, we added in Sect. 2.3 the statement that *"All subsequent analyses presented in the next sections are based on the filtered OT dataset."*
  And yes, we did not further considered missing hail reports. This effect cannot be quantified because of the large number of unreported hail events (we checked that in previous studies for other countries where we got geo-referenced hail damage data).

- Line 305: "covered by OTs/hail streaks" → covered by individual OTs"
  This refers not to individual, but to grouped OTs (i.e., events). We changed that to "OT events". Further, we hope this is now clear with the explanation added above.

- Line 306: While I understand your meaning here, since this ratio has an inherent upper bound of 1 the phrasing could be better.
  Yes, this sentence is a bit strange. We split that into two sentences.

- Line 307: Isn't this product $f^2$?
  No, f refers to the area and not length and width separately.
  We cannot follow why this should be proportional to the square of f?

- Line 308: " > 105 km², not shown"
  It's the logarithm in the former Fig. 11e, thus $10^5$ is correct.

- Line 328: "Storm's severity" → "tornadic intensity". "Storm severity" is too nebulous a term, since it could be interpreted as meaning "updraft strength", which is not necessarily correlated with severe impacts on the ground.
  Changed accordingly
- Lines 381- 384: Why was this figure removed? A comparison of the frequency of occurrence of hail events (not including size) across the country in the historical vs. stochastic datasets seems of prime importance. If the figure is not retained, then discussion about it should be eliminated.
  This Figure was removed to save space, but if that is not a prime concern - happy to include it again based on the reviewer feedback.
- Section 3.6: Much improved, and an interesting result when compared to the Smith et al. study. If I am reading correctly, the stochastic event set underestimates the occurrence of hail days in the region, but potentially overestimates severe hail. It's possible the break down in the updraft strength - large hail relationship is causing these large biased "severe" hail numbers (also possible large hail is underreported, as you note.) Any idea what could be causing the overestimation of hail in general?
- Indeed, while we didn't mention it explicitly, in the model every other hail event has 3 or more cm, and that would mean more than 7 such severe hail days per year for the Gauteng region, much higher than Smith's result. Besides underreporting of large hail in the Smith study, we also need to consider underreporting of small hail in the model's severity data, which could explain an overall overestimation of large hail.
- Lines 505- 507: While I agree with these statements, I would shift them earlier in the conclusions as they are awkwardly placed here.
  We have moved the two sentences to the beginning of the Conclusion section.
- Fig. B1: Great addition. Can these plots be normalized by total number of detections and plotted in the same plot for easier comparison?
  We exchanged Fig. B1 with a normalized graph.
- Lines 570- 580: Excellent addition. I'd rearrange the text (or figure) so the subfigures are referenced in order.
  We rearranged the text as suggested, but also added a short statement for Fig. B1a and e.

**Grammatical:**

All suggestions / corrections were considered

- Line 47: "this" → "these"
- Line 131:" ... east of..." → " .. in east..."
- Line 206: Add "Fig. 6" after Fig. 3, so the comparison has an object.
- Line 510: "95." → "95$^{th}$"
- line 221: "of the event, or grouped, OTs"
- Line 267: "of" → "from"
- Fig 11: years → years'
- Line 417: "hail hazard" → "the hail hazard"